# Hierarchical fluctuation shapes a dynamic flow linked to states of consciousness

Ang Li [1,14] ✉, Haiyang Liu[2,3,14], Xu Lei [4,5,14], Yini He[6], Qian Wu [6,7], Yan Yan [8], Xin Zhou[6], Xiaohan Tian[6], Yingjie Peng[1], Shangzheng Huang[1], Kaixin Li[1], Meng Wang[6], Yuqing Sun[6], Hao Yan [9,10], Cheng Zhang[11], Sheng He[1], Ruquan Han[2] ✉, Xiaoqun Wang[1,6,7,12] ✉ & Bing Liu [6,7,13] ✉

Consciousness arises from the spatiotemporal neural dynamics, however, its relationship with neural flexibility and regional specialization remains elusive. We identified a consciousness-related signature marked by shifting spontaneous fluctuations along a unimodal-transmodal cortical axis. This simple signature is sensitive to altered states of consciousness in single individuals, exhibiting abnormal elevation under psychedelics and in psychosis. The hierarchical dynamic reflects brain state changes in global integration and connectome diversity under task-free conditions. Quasi-periodic pattern detection revealed that hierarchical heterogeneity as spatiotemporally propagating waves linking to arousal. A similar pattern can be observed in macaque electrocorticography. Furthermore, the spatial distribution of principal cortical gradient preferentially recapitulated the genetic transcription levels of the histaminergic system and that of the functional connectome mapping of the tuberomammillary nucleus, which promotes wakefulness. Combining behavioral, neuroimaging, electrophysiological, and transcriptomic evidence, we propose that global consciousness is supported by efficient hierarchical processing constrained along a low-dimensional macroscale gradient.

The stream of consciousness, as delineated by William James, is an ever-flowing mental continuity of subjective experiences. A conscious brain is capable of adapting, learning, and guiding for future actions in a constantly evolving environment. During natural sleep or pharmacological anesthesia, consciousness seems to be attenuated or absent; under psychedelics or psychosis, distortions can occur.

Revealing the complex but orchestrated brain organization underlying these global brain states (i.e., different levels of

[1]State Key Lab of Brain and Cognitive Science, Institute of Biophysics, Chinese Academy of Sciences, Beijing 100101, China. [2]Department of Anesthesiology, Beijing Tiantan Hospital, Capital Medical University, Beijing 100101, China. [3]Department of Anesthesiology, Qinghai Provincial Traffic Hospital, Xining 810001, China. [4]Sleep and Neuroimaging Center, Faculty of Psychology, Southwest University, Chongqing 400715, China. [5]Key Laboratory of Cognition and Personality (Southwest University), Ministry of Education, Chongqing 400715, China. [6]State Key Laboratory of Cognitive Neuroscience and Learning, Beijing Normal University, Beijing 100875, China. [7]IDG/McGovern Institute for Brain Research, Beijing Normal University, Beijing 100875, China. [8]Shenzhen Institutes of Advanced Technology, Chinese Academy of Sciences, Shenzhen, Guangdong 518055, China. [9]Peking University Sixth Hospital/Institute of Mental Health, Beijing 100191, China. [10]NHC Key Laboratory of Mental Health (Peking University), National Clinical Research Center for Mental Disorders (Peking University Sixth Hospital), Beijing 100191, China. [11]The Department of Respiratory and Critical Care Medicine, Peking University First Hospital, Beijing 100034, China. [12]New Cornerstone Science Laboratory, Beijing Normal University, Beijing 100875, China. [13]Chinese Institute for Brain Research, Beijing 102206, China. [14]These authors contributed equally: Ang Li, Haiyang Liu, Xu Lei. ✉e-mail: al@ibp.ac.cn; ruquan.han@gmail.com; xiaoqunwang@bnu.edu.cn; bing.liu@b-nu.edu.cn

consciousness) is essential to understanding the neural mechanisms of consciousness[1]. The spatiotemporal organization properties of spontaneous brain activity, which is considered to constantly provide top-down predictive models for future interactions during perception[2,3], may offer essential insights into consciousness across different conditions (e.g., during anesthesia and sleep) and species. Resting-state fMRI (rsfMRI), as the widely used technique in humans to map the spatiotemporal patterns of spontaneous brain signals, has detected highly reproducible intrinsic functional brain networks[4], which largely reflect anatomical organization, individual-specific[5] and task-evoked[6] information. Breakdown in consciousness are accompanied by intricate changes in various aspects of the intrinsic functional organization, such as long-distance interactions[7–9], anti-correlated structures[10], and patterns of brain coordination[11,12]. Therefore, uncovering the temporal and spatial aspects of spontaneous brain activity in different global brain states are crucial for understanding the unified brain functional organization of consciousness levels.

Temporal variability/flexibility, which quantifies the dynamic range of ongoing brain activities, is increasingly being recognized as a beneficial factor for the adaptability of neural systems[13–15]. For instance, greater variability of BOLD signals has been reliably observed in younger healthy individuals[16] and associated with more efficient performance in cognitive tasks[13,14,17]. According to computational models[18], the variability of spontaneous activity arises from the dynamic system's noise-induced transitions between multistable attractors[19] or fluctuation around a critical line proximal to instability[20]. Generally, this variability reflects and supports the exploration of a broader brain dynamical repertoire. Conversely, deterministic task stimuli limits the flexibility and quenches signal variability[21,22]. In the spatial domain, resting-state BOLD variability is not randomly distributed but recapitulates the relative expression of cell markers for input-modulating somatostatin and output-modulating parvalbumin interneurons[23], which plays an important role in mediating cortico-cortical communication[24]. Taking these into account, we consider that neural variability organized in an intrinsic spatial arrangement, i.e., the integration of space and time, would be optimal for conscious processing and be sensitive to changes of brain states. Consequently, we hypothesize that a topographically organized neural variability pattern may orchestrate the rise and fall in global states of consciousness. Importantly, this hypothesis prompted us to focus on the search for a temporal-spatial nested signature rather than a specific neuroanatomical location as a determinant of consciousness.

To test this, we incorporated several different fMRI paradigms that capture altered states of consciousness, from deep sleep or under anesthesia to alert wakefulness. First, we systematically investigated how spatiotemporally embedded variability across the neocortex changes in three conditions–dexmedetomidine-induced sedation, normal sleep, and resting-state scanning, in which a proportion of individuals tend to naturally fall asleep in minutes. The results revealed a common hierarchical shift in spontaneous cortical activity. Second, to validate the signature, we collapsed the spatiotemporal dynamics into a simple, low-dimensional index to delineate the hierarchical fluctuation at the level of minutes. The hierarchical index was further tested for different energetic states (i.e., caffeine or fasting) of a single densely sampled subject, volunteers who had been administered a psychedelic drug, and individuals with neuropsychiatric disorders. Subsequently, using fMRI data from the Human Connectome Project[25] (HCP), we showed that the hierarchical fluctuation covaried with the global integration topology, connectome complexity, and an infra-slow propagation wave previously implicated in arousal modulation[26,27]. Similar hierarchical signatures can also be observed in macaques based on ECoG recordings. In addition, spatial analysis using the genetic transcriptome from the Allen Human Brain Atlas[28] and high-resolution HCP 7T fMRI data suggested a contribution from the hypothalamic TMN region that regulates wakefulness.

## Results

### Hierarchical cortex-wide fluctuations reflect ongoing states of consciousness

We examined whether there is a consciousness-related topological pattern of neural variability at the group level using three independent fMRI datasets, each of which engaged a distinct task-free paradigm: anesthesia, sleep, and drowsiness during resting-state scans. We operationalized neural variability as the standard deviation of low-frequency BOLD signal for each voxel/vertex across time. The cortical map was z-normalized to emphasize the spatial heterogeneity.

In Dataset 1 (Fig. 1a; "Methods"), we compared the cortical neural variability from 21 healthy volunteers across three conscious states: wakefulness, dexmedetomidine-induced sedation, and recovery (paired $t$-statistic contrast shown in Fig. 1b and Supplementary Fig. 1). Dataset 2 comprised the resting-state fMRI data from a large sample of healthy subjects from the HCP project. Previous research has demonstrated that individuals would generally exhibit a decrease in alertness over the course of the scan at the population level[29,30]. Therefore, we divided each resting-state run from HCP data into nonoverlapping time windows (50 volumes, 24 windows), and two different time windows were paired to form 276 (24 × (24 − 1)/2) different combinations, with the time interval ranging from 1 to 23 intervals. The larger interval of two windows was inferred to have a higher possibility of a decrease in vigilance. To measure the group-level pattern of alertness dropping, Spearman's rank r was calculated between the normalized low-frequency BOLD variability and the time interval across 1,084,128 (276 combinations × 4 runs × 982 individuals) pairs of states/time windows (Fig. 1d–e, "Methods", Supplementary Fig. 2). In Dataset 3, we analyzed simultaneous EEG–fMRI data from 6 healthy volunteers over a period of sleep around 2 h, to investigate how changes in the low-frequency BOLD variability pattern (Fig. 1g, h) correlate with manually labeled sleep stages. Specifically, the awake, N1, N2 and slow-wave sleep states were encoded as four distinct levels in order to quantify the rank correlation with the fMRI maps (Methods). As a result, we observed a consistent pattern across these three experiments (Fig. 1j), suggesting that there is a common cortex-wide signature of consciousness. At the network level, this pattern manifests as an allocation of neural variability varying from the high-order[31] (Fig. 1k–m, default mode, control, and limbic networks) to low-level networks (Fig. 1k–m, visual and sensorimotor networks).

Previous studies[32,33] have well-documented the existence of a principal functional gradient in the human brain. This gradient explains the greatest variance in the functional connectome and captures the cortical processing hierarchy, spanning from primary sensory to transmodal areas. In addition, T1w/T2w mapping has been shown to provide a non-invasive correlate of the anatomical hierarchy based on the structural profiles[34]. To quantitatively characterize our observations, we performed a cortex-wide spatial comparison between the consciousness-related patterns and the main functional gradient derived from the dense functional connectome data (Fig. 1c, f, i), as well as T1w/T2w mapping (Supplementary Fig. 3). Statistical significance was assessed using 10,000 spatially permutation-based null models; specificity was strengthened by controlling for other functional gradients and within individual Yeo's networks[31] (Supplementary Figs. 3–5).

### Characterizing hierarchical dynamics in single individuals

Next, we aim to test whether the topologically altered spontaneous cortical activities can be compressed to capture the graded changes of the levels of consciousness, elucidating its reproducibility and power at the individual level. Therefore, we defined a univariate 'hierarchical index' as the rank correlation between the spatial distribution of cortical BOLD variability and the group-level principal functional gradient ("Methods"). This hierarchical index can provide a proxy for the topological shift of cortical neural variability, using the group-level

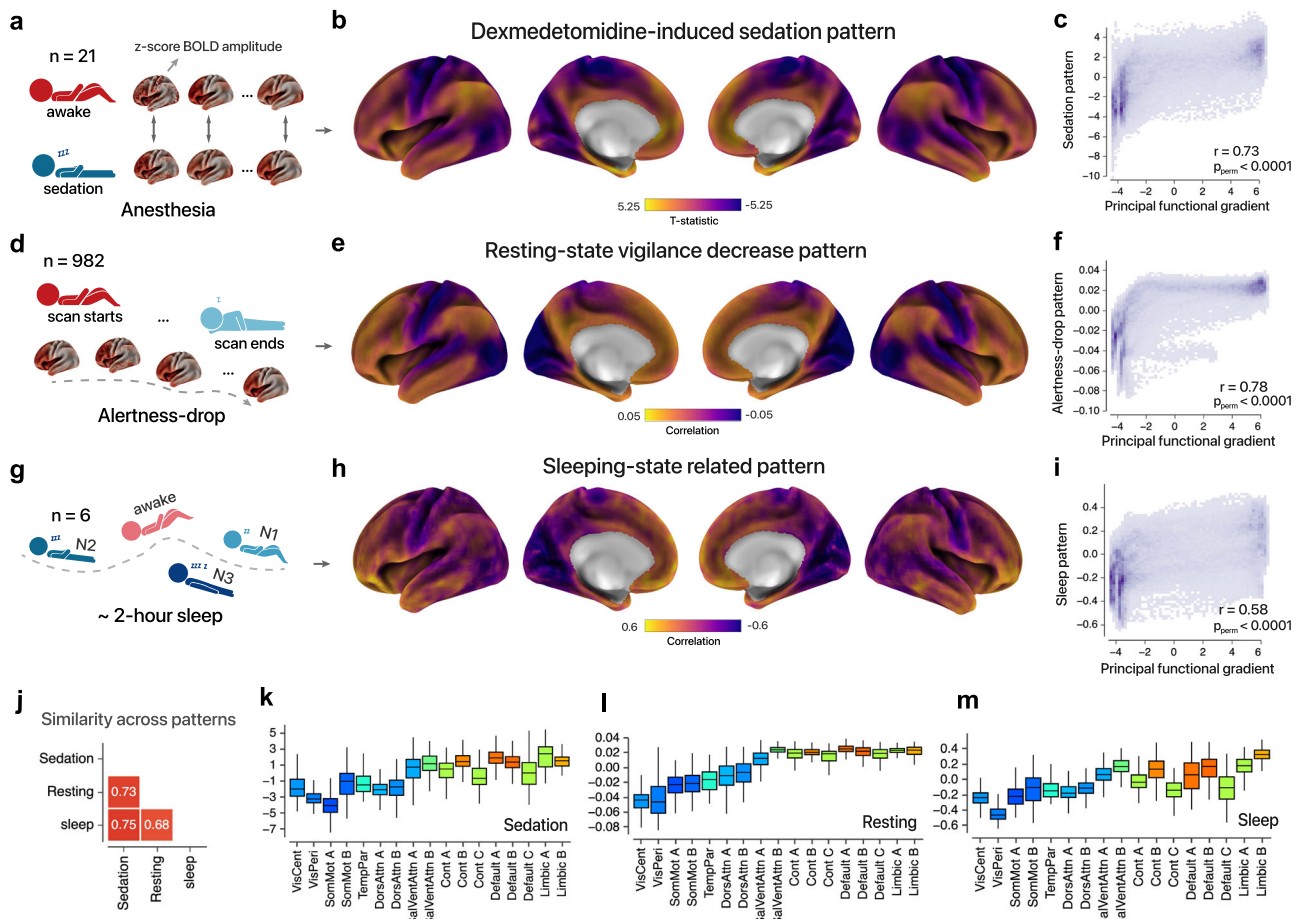

**Fig. 1 | Shared spatial signature of cortex-wide BOLD amplitude relating to anesthesia, sleep, and vigilance. a** Schematic diagram of the dexmedetomidine-induced sedation paradigm; $z$-normalized BOLD amplitude was compared between initial wakefulness and sedation states ($n = 21$ volunteers) using a two-sided paired $t$-test; fMRI was also collected during the recovery states and showed a similar pattern (Supplementary Fig. 1). **b** Cortex-wide, unthresholded $t$-statistical map of dexmedetomidine-induced sedation effect. For the purposes of visualization as well as statistical comparison, the map was projected from the MNI volume into a surface-based CIFTI file format and then smoothed for visualization (59412 vertexes; same for the sleep dataset). **c** Principal functional gradient captures spatial variation in the sedation effect (wakefulness versus sedation: $r = 0.73$, $P_{perm} < 0.0001$, Spearman rank correlation). **d** During the resting-state fMRI acquisition, the level of vigilance is hypothesized to be inversely proportional to the length of scanning in a substantial proportion of the HCP population ($n = 982$ individuals). **e** Cortex-wide unthresholded correlation map between time intervals and $z$-normalized BOLD amplitude; a negative correlation indicates that the signal

became more variable along with scanning time and vice versa. **f** The principal functional gradient is correlated with the vigilance decrease pattern ($r = 0.78$, $P_{perm} < 0.0001$, Spearman rank correlation). **g** Six volunteers participated in a 2-h EEG–fMRI sleep paradigm; the sleep states were manually scored into wakefulness, N1, N2, and slow-wave sleep by two experts. **h** The cortex-wide unthresholded correlation map relating to different sleep stages; a negative correlation corresponds to a larger amplitude during deeper sleep and vice versa. **i** The principal functional gradient is associated with the sleep-related pattern ($r = 0.58$, $P_{perm} < 0.0001$, Spearman rank correlation). **j** Heatmap plot for spatial similarities across sedation, resting-state drowsiness, and sleep pattens. **k–m** Box plots showing consciousness-related maps (**b**–**e**) in 17 Yeo's networks[31]. In each box plot, the midline represents the median, and its lower and upper edges represent the first and third quartiles, and whiskers represent the 1.5 × interquartile range (sample size vary across 17 Yeo's networks, see Supplementary Fig. 3). Each network's color is defined by its average principal gradient, with a jet colorbar employed for visualization.

gradient as an empirical map, which differs from studies focusing on the individual-level perturbation of functional gradients[33,35]. In Dataset 1, we found that the hierarchical index exhibited significant difference across three conscious states, showing a consistent reduction in 20 of 21 participants from wakefulness to sedation (Fig. 2a). In Dataset 2, the hierarchical index was calculated for abovementioned 24 non-overlapped windows in each resting-state run; as a result, significant decrease occurs in 37.4% of the individual runs across time (Fig. 2b, uncorrected $P < 0.05$). Interestingly, the average index across the four runs significantly associated with inter-individual differences relating to vigilance, such as in sleep quantity, impulsivity, response time, and accuracy during task fMRI (Fig. 2b, Supplementary Table 1). In Dataset 3, we compared the trajectories in the hierarchical index and the sleep staging information manually labeled by experts, high synchrony can be observed across all 6 individuals ("Methods", Fig. 2c, rs = 0.75-0.89,

$Ps < 0.0001$); the individual-level correlation can be replicated in an independent simultaneous EEG–fMRI sleep dataset[36] ("Methods", Supplementary Fig. 6). To validate the signature at a longer timescale, we analyzed the resting-state fMRI data from the MyConnectome Project[37] (Dataset 4, "Methods"). This resource provides a longitudinal assessment of fMRI acquisition and self-report states for a single individual more than one year. According to an earlier work[37], 84 sessions were included in our analysis. As a result, the hierarchical indices in the fed/caffeinated session were considerably higher than those in the fasted/uncaffeinated session, enabling a univariate classification accuracy over 80% (Fig. 2d, Mann–Whitney $u$ test; $P < 0.0001$). The hierarchical index also associated with daily fluctuations in fatigue (averaged score of "drowsy", "sleepy", "sluggish", "tired"; Fig. 2e) and heightened attention (averaged score of "attentive", "concentrating", "lively"; Fig. 2f), based on self-report measures ("Methods").

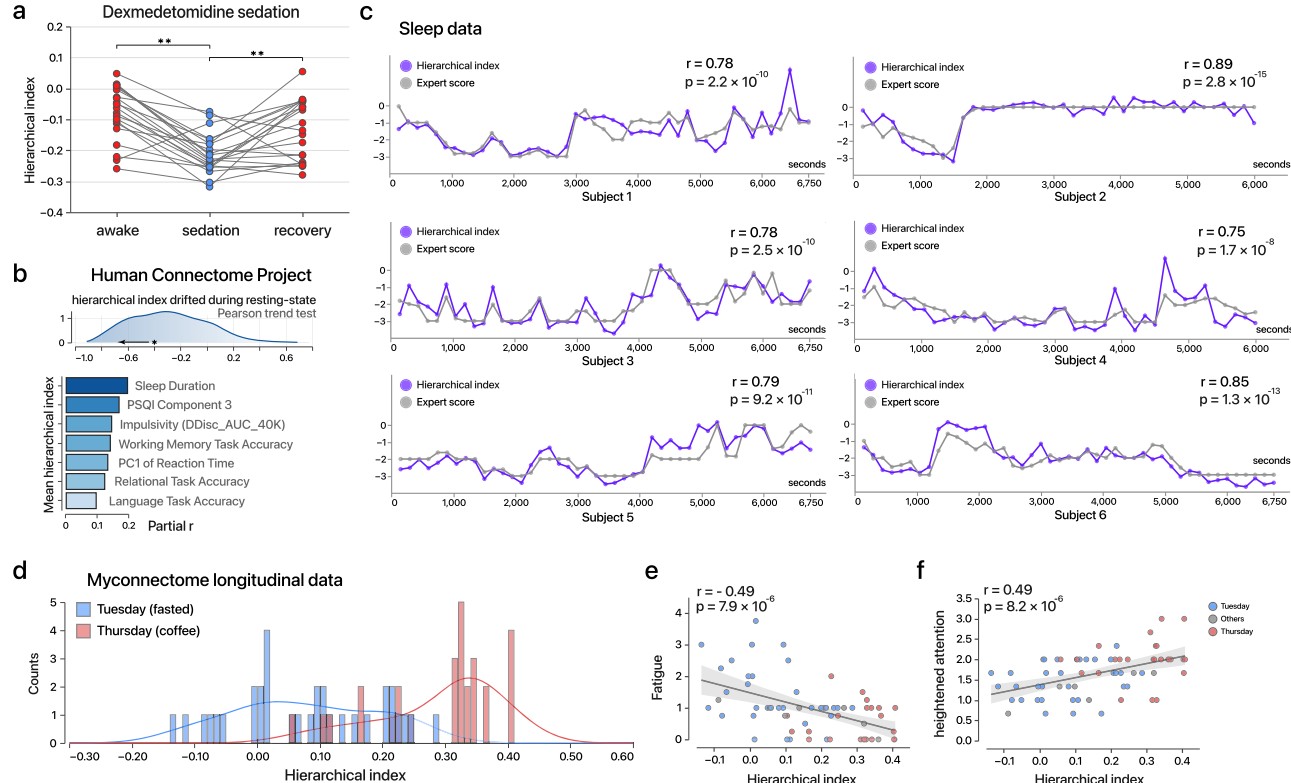

**Fig. 2 | Low-dimensional hierarchical index tracks fluctuations in multiple consciousness-related brain states. a** The hierarchical index distinguished the sedation state from wakefulness/recovery at the individual level (**P < .01, wakefulness versus sedation: $t = 6.96$, unadjusted $P = 6.6 \times 10^{-7}$; recovery versus sedation: $t = 3.19$, unadjusted $P = 0.0046$; no significant difference was observed between wakefulness and recovery; two-sided paired $t$-test; $n = 21$ volunteers, each scanned in three conditions). **b** Top: distribution of the tendency of the hierarchical index to drift during a ~15 min resting-state scanning in HCP data (982 individuals × 4 runs; *$P < 0.05$, unadjusted, Pearson trend test); a negative correlation indicates a decreasing trend during the scanning; bottom: partial correlation (controlling for sex, age, and mean framewise distance) between the hierarchical index (averaged across four runs) and behavioral phenotypes. PC1 of reaction time and PSQI Component 3 were inverted for visualization (larger inter-individual hierarchical index corresponds to less reaction time and healthier sleep quality). **c** The hierarchical index captures the temporal variation in sleep stages in each of six

volunteers (gray line: scores by expert; blue line: hierarchical index; Pearson correlation). The vertical axis represents four sleep stages (wakefulness = 0, N1 = −1, N2 = −2, slow-wave sleep = −3) with time is shown on the horizontal axis (Subject 2 and Subject 4 were recorded for 6000 s; the others summed up to 6750 s); For the visualization, we normalized the hierarchical indices across time and added the average value of the corresponding expert score. **d** Distribution of the hierarchical index in the Myconnectome project. Sessions on Thursdays are shown in red color (potentially high energic states, unfasting / caffeinated) and sessions on Tuesdays in blue (fasting/uncaffeinated). Applying 0.2 as the threshold corresponding to a classification accuracy over 80% (20 of 22 Tuesday sessions surpassed 0.2; 20 in 22 Thursday sessions were of below 0.2) **e–f** The hierarchical index can explain intra-individual variability in energy levels across different days (two-sided unadjusted Spearman correlation). The error band represents the 95% confidence interval. Source data are provided as a Source Data file.

Collectively, the results demonstrate that the hierarchical dynamics of cortical variability can characterize reproducible changes in single individuals under different conditions and timescales over minutes, hours, and days, and potentially across individuals.

## Hierarchical dynamics in psychedelic and psychotic brains

Having studied trajectories of the signature across wakefulness, anesthesia, and sleep, we next aimed to investigate whether it can be sensitive to other altered states of consciousness. Here, we focused on two related situations:[38–40] (i) psychotomimetic effects of drugs which inducing altered subclinical psychotic-like experiences; (ii) individuals with psychiatric disorders whose conscious processing might be impaired. To evaluate this, we used Dataset 5 (see Methods), which included 15 healthy subjects undergoing repeated fMRI scans under the influence of lysergic acid diethylamide (LSD) or placebo in a balanced-order, within-subjects design[41]. Interestingly, we found that the hierarchical indices were significantly higher in the LSD condition than in the placebo condition across 15 subjects (Fig. 3a, $P < 0.01$, Two-way ANOVA). We then associated the hierarchical index with clinical symptoms across the individuals with psychiatric disorders (either schizophrenia, bipolar disorder or attention-deficit/hyperactivity

disorder) within the Consortium for Neuropsychiatric Phenomics dataset[42] (Dataset 6, "Methods"). Correspondingly, we found that the hierarchical index was significantly associated with inter-individual psychotic symptoms, particularly those symptoms which can be induced by psychedelics (such as hallucinations, Fig. 3b). Abnormally higher hierarchical index can be observed during the resting-state in individuals with schizophrenia and bipolar disorder (Fig. 3c), but not in those with attention-deficit/hyperactivity disorder. We replicated the group difference in an independent Chinese cohort which includes individuals with schizophrenia (Dataset 6, Fig. 3c). These results suggest that the deviation of the hierarchical signature is potentially detrimental and characterizes abnormal states of conscious processing.

## Complex brain integration and differentiation

Multiple theories[43–46] suggest that conscious experiences require both functional integration as well as diversity. In light of these earlier theories, we speculate that hierarchical dynamics would accompany a flexible connectome reconfiguration in cortical integration/coordination and diversity during the resting state. Therefore, we employed a modified version of global signal (GS) topology to specifically quantify

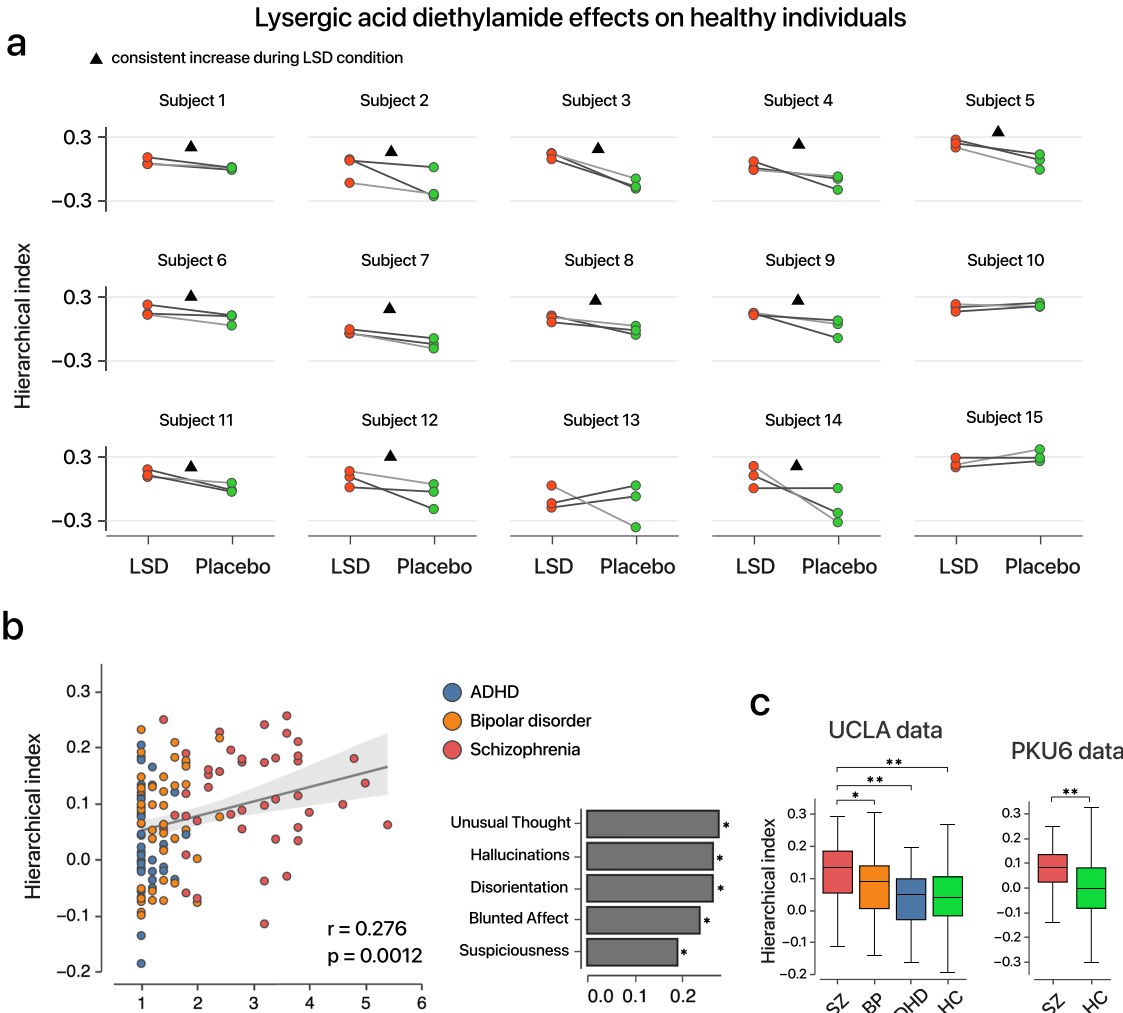

**Fig. 3 | Hierarchical index in psychedelic and psychotic brains. a** LSD effects on the hierarchical index across 15 healthy volunteers. fMRI images were scanned three times for each condition of LSD administration and a placebo. During the first and third scans, the subjects were in an eye-closed resting-state; during the second scan, the subjects were simultaneously exposed to music. A triangle (12 of 15 subjects) indicates that the hierarchical indices were higher across three runs during the LSD administration than in the placebo condition. **b** Left: relationship between the hierarchical index and BPRS positive symptoms across 133 individuals with either ADHD, schizophrenia, or bipolar disorder ($r = 0.276$, $P = 0.0012$, two-sided unadjusted Spearman correlation). The error band represents the 95% confidence interval of the regression estimate. Right: correlation between the hierarchical index and each item in BPRS positive symptoms (*$P < 0.05$, **$P < 0.01$, two-sided unadjusted Spearman correlation; see Source Data for specific $r$ and $P$ values). **c** Left: the hierarchical index across different clinical groups from the UCLA dataset (SZ schizophrenia, $n = 47$; BP bipolar disorder, $n = 45$; ADHD attention-deficit/hyperactivity disorder, $n = 41$; HC healthy control, $n = 117$); right: the hierarchical index across individuals with schizophrenia ($n = 92$) and healthy control ($n = 98$) from the PKU6 dataset. In each box plot, the midline represents the median, and its lower and upper edges represent the first and third quartiles, and whiskers represent the 1.5 × interquartile range. *$P < 0.05$, **$P < 0.01$, two-tailed two-sample $t$-test. Source data are provided as a Source Data file.

the spatial inhomogeneity of cortex-wide integration/coordination (Methods). Specifically, we computed the mean cortical signals to capture the most prominent dynamics across the cortex, with peaks of high amplitude indicating a high degree of spatial homogeneity at that moment[47]. As depicted in Fig. 4a, the GS topology characterizes the degree of integration with global cortical dynamics for a given cortical region.

To temporally decompose distinct patterns of cortex-wide coordination, we analyzed 9600 cortical maps of time-resolved GS topology from 100 unrelated participants in HCP (4 runs for each person and 24 nonoverlapped windows each run; see "Methods"). We found that the hierarchical index can explain the distinct patterns of the GS topology: a higher index corresponds to a GS topology that spatially resembles the cortical hierarchy (Fig. 4b, $r = 0.55$, $P < 0.0001$); the relationship between the two variables reflects the two-cluster differentiation of GS topologies using K-means approach (Fig. 4b,

"Methods"). The GS topology difference between two states was highly analogous to the cortical hierarchy (Fig. 4e, $r = 0.89$, State 1: higher hierarchical index, and global signal 'integrated' with higher-order regions; State 2: opposite). Importantly, no individual was entirely assigned to state 1 or state 2, with the proportion of state 1 occupied 52.3 ± 18.8% across the 100 unrelated subjects (Supplementary Fig. 7a). Control analyses suggest that the clustering and result were primarily driven by changes in brain state rather than individual differences and can be replicated in independent subjects (Supplementary Figs. 7 and 8). As an alternative, a weighted strength approach was applied as a proxy for functional integration/coordination. The node-wise weighted strength was developed based on the graph-theoretical concept of degree, which aims to quantify the global functional connectivity to other regions. Following the same clustering pipeline, we found that the two approaches yielded a similar clustering solution ("Methods", Supplementary Fig. 9). To measure the functional diversity, we

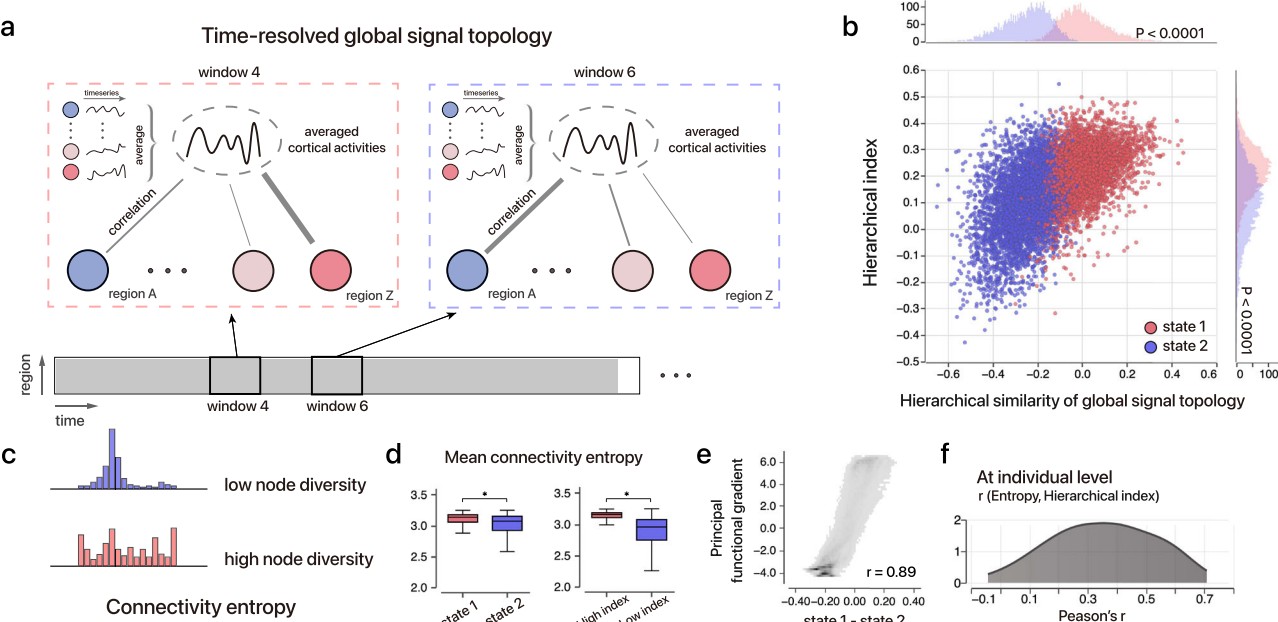

**Fig. 4 | Complex and dynamic brain states unveiled by global signal topology and the hierarchical index during rest. a** Simplified diagram for dynamic GS topology analysis. **b** two-cluster solution of the GS topology in 9600 time windows from 100 unrelated HCP individuals. Scatter and distribution plots of the hierarchical index; the hierarchical similarity with the GS topology is shown. Each point represents a 35 s fragment. State 1 has significantly larger hierarchical index ($P < 0.0001$, two-sided two-sample $t$-test) and hierarchical similarity with GS topology ($P < 0.0001$, two-sided two-sample $t$-test) than State 2, indicating a higher level of vigilance and more association regions contributing to global fluctuations; meanwhile, the two variables are moderately correlated ($r = 0.55, P < 1 \times 10^{-100}$, two-sided Spearman correlation). **c** For a particular brain region, its connectivity entropy is characterized by the diversity in the connectivity pattern. **d** Left: Higher overall connectivity entropy in State 1 than State 2 ($P = 1.4 \times 10^{-71}$, two-sided two-

sample $t$-test, $n_{\text{state 1}} = 4571, n_{\text{state 2}} = 5021$). Right: higher overall connectivity entropy in states with a higher hierarchical index (top 20% versus bottom 20%; $P < 1 \times 10^{-100}$, two-sided two-sample $t$-test, $n_{\text{high}} = 1920, n_{\text{low}} = 1920$). *$P < 0.0001$. In each box plot, the midline represents the median, and its lower and upper edges represent the first and third quartiles, and whiskers represent the $1.5 \times$ interquartile range. **e**, Difference in GS topology between State 1 and State 2 spatially recapitulates the principal functional gradient ($r = 0.89, P < 1 \times 10^{-100}$), indicating that the data-driven GS transition moves along the cortical hierarchy. **f** Distribution of Pearson's correlation between the hierarchical index and mean connectivity entropy across 96 overlapping windows (24 per run) across 100 individuals. In most individuals, the hierarchical index covaried with the diversity of the connectivity patterns (mean $r = 0.386$). Source data are provided as a Source Data file.

performed an entropy-based analysis deriving the connectome and temporal entropy for above time-resolved brain states (Fig. 4c, "Methods"). Both measures, especially connectome entropy, were considerably higher during the periods of State 1 compared with State 2, and associated with higher hierarchical index (Figs. 4d, 4f).

The results suggest that during resting states with a higher hierarchical index, cortical activity exhibits more complex cortex-wide coordination compared to the 'GS topology stereotype' dominated by sensory cortices[48]. This is associated with increased functional diversity in the connectome and temporal fluctuations, which may support cognitive demands during high vigilance.

**Relationship to the infra-slow cortex-wide propagation phenomenon**

Hierarchical heterogeneity may involve in the processing of infra-slow arousal regulation. Recent fMRI studies have consistently revealed an infra-slow global wave[26,27,49] that intrinsically propagates along the macroscale functional gradient and that is relevant to arousal and autonomic fluctuations. Based on a data-driven quasiperiodic pattern (QPP) analysis[26], the most primary recurring spatiotemporal pattern lasts approximately 20 s. However, it is unclear that how the infra-slow global wave propagates across different states of overall vigilance. Based on our results, we speculate that during higher states of vigilance, such a spatiotemporal wave would more favorably 'propagate' to the association cortex to modulate large-scale activities and complete an unabridged QPP cycle.

Therefore, we analyzed the resting-state fMRI data from the 100 participants in the HCP who showed the greatest reduction in the

hierarchical index across four runs (average Pearson's $r < -0.49$, "Methods") and truncated their initial and terminal 400 frames as coarse proxies for the high and low vigilance states, respectively. To determine the primary QPP event in the population, we downloaded a recently published QPP template[26] which was generated using an optimized, computationally expensive algorithm based on vertex-wise cortical data. As shown in Fig.5a, the primary QPP manifests as a dynamic cycle of activation and deactivation which spatially following the macroscale gradient, lasting approximately 21.6 s (30 volumes). To match the possible QPP events, we applied a sliding window approach to iteratively compare correlation between the template and each spatiotemporal flattened segment with a temporal step of 1 TR. Segments were identified as QPP events whose local maxima exceeding the threshold ($r = 0.4$), resulting hundreds of events detected for the 100 subjects (Fig. 5a, b; "Methods"; initial states: 785 events, terminal states: 993 events). We extracted the average time series from the top and bottom 20% cortical hierarchy regions with QPP events and found that almost all low-order fluctuations exhibited similar dynamics, resembling the typical trajectory in the group template (Fig. 5c, e). However, the high-order fluctuations bifurcated into distinct modes (Fig. 5d): typical or atypical trajectories (Fig. 5f). The great majority of trajectories were 'typical' in the initial 400 frames (i.e., higher vigilance). The proportion of 'atypical' trajectories significantly increased during the terminal 400 frames, suggesting that such waves were disregarded in the regulation of high-order networks. Meanwhile, the lower-order regions had larger fluctuations and thus dominated the global signal. With regard to functional connectivity within the QPP events, the initial periods exhibited more anticorrelation structures

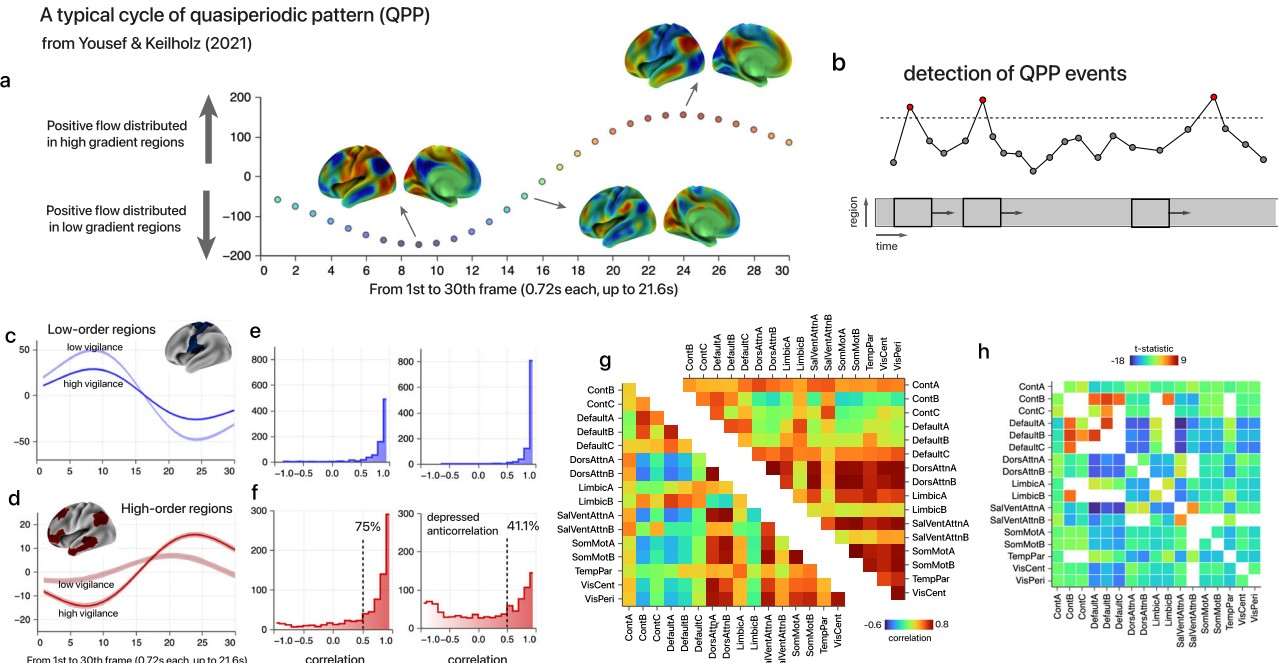

**Fig. 5 | fMRI quasiperiodic pattern manifested in different vigilance states. a** A cycle of spatiotemporal QPP reference from Yousef & Keilholz;[26] x-axis: HCP temporal frames (0.72 s each), y-axis: dot product of cortical BOLD values and principal functional gradient. Three representative frames were displayed: lower-order regions-dominated pattern (6.5 s), intermediate pattern (10.8 s) and associative regions-dominated pattern (17.3 s). **b** A schematic diagram to detect QPP events in fMRI. The sliding window approach was applied to select spatiotemporal fragments, which highly resemble the QPP reference. **c, d**, Group-averaged QPP events detected in different vigilance states (initial and terminal 400 frames, respectively). For this visualization, the time series of the bottom 20% (**c**, blue) and top 20% (**d**, red) of the hierarchy regions were averaged across 30 frames. Greater color saturation corresponds to the initial 400 frames with plausibly higher vigilance. Line of dashes: $r = 0.5$. **e, f**, Distribution of the temporal correlations between the averaged time series in the template and all the detected QPP events. Left: higher vigilance; right: lower vigilance. For the top 20% multimodal areas, an $r$ threshold of 0.5 was displayed to highlight the heterogeneity between the two states. **g** Mean correlation map of Yeo 17 networks across QPP events in different vigilance states. Left: higher vigilance; right: lower vigilance. **h** A thresholded t-statistic map of the Yeo 17 networks measures the difference in Fig. 5g (edges with uncorrected $P < .05$ are shown, two-sided two-sample $t$-test). Source data are provided as a Source Data file.

between the internal (default mode and control networks) and external systems (attention and sensory networks) and were greatly diminished by the end of the scanning session (Fig. 5g, h). This anticorrelation structure was previously considered to be a signature for the waking state in comparison with anesthesia[50] or disorders of consciousness[51]. The result showed a significant heterogeneity in the infra-slow wave primarily in the higher-order areas across different brain states.

## Hierarchical dynamics in macaque electrocorticography

BOLD signal fluctuation reflects localized changes in neural activity indirectly through a complex neurovascular coupling[52,53]. In this work, we ascribed the observed hierarchical signature to altered patterns of neural activities rather than to the physiological baseline. Previous experimental studies[53,54] suggested that local field potential gamma power could mediate the BOLD signals and localized coordination of neural activities. Furthermore, considering the cross-species conservation of the large-scale cortical hierarchical architecture in primates, here we hypothesize that the similar signature for global states of consciousness can be observed based on gamma band power fluctuations in macaque electrocorticography (ECoG).

We tested this idea using openly available data from the Neurotycho project[55,56] (Dataset 7), including large-scale, spatially resolved electrophysiological recordings in two densely sampled macaques across different states: awake (both eyes-opened and eyes closed), sleep, and anesthesia conditions. Following a similar approach to human fMRI data, we constructed a channel-channel functional network based on temporal fluctuations of gamma band-limited power

(BLP), and then calculated the cortical functional gradients using a diffusion embedding algorithm. Despite the sparse sampling of implanted electrodes in the two macaques and their considerable disparity ("Methods", Supplementary Fig. 10), the calculated principal gradients in each macaque revealed a clear unimodal-transmodal hierarchy across the neocortex (Fig. 6a, b), which is consistent with a recent study[27]. To statistically compare the cortical neural variability based on gamma BLP, we temporally segmented the ECoG recordings into multiple nonoverlapped windows (150 s, "Methods") and performed group comparisons of z-normalized maps between distinct states of consciousness. As a result, the patterns of neural variability varied significantly along the macroscale hierarchy during sleep and anesthesia, in comparison with the awake states (Fig. 6c-f, *Chibi*: $r_{\text{awake vs sleep}} = 0.61$, $r_{\text{recovery vs anesthesia}} = 0.58$; *George*: $r_{\text{awake vs sleep}} = 0.84$, $r_{\text{recovery vs anesthesia}} = 0.57$). Similarly, we calculated a low-dimensional hierarchical index based on gamma BLP and found the score can significantly distinguish different conditions (Fig. 6g, h). These findings suggest that the hierarchical shifting observed in electrophysiological and BOLD signals may underlie a shared neural mechanism of neural variability. Subsequently, we extended the QPP analysis to the macroscale, cortex-wide gamma BLP fluctuations, using a time window of 20 s to determine the recurrent spatiotemporal patterns. For each macaque, a propagating wave was revealed that was analogous to that found using human fMRI data (Fig. 6i, j, Supplementary Fig. 11a, b). Meanwhile, the average spectrogram of the gamma-BLP QPP events comprised the loss of mid-frequency activity as well as concurrent increases in low-frequency power and resembled a pattern of sequential spectral transitions (SSTs, as shown in Fig. 6k, l). SST events were identified by momentarily increases in delta power

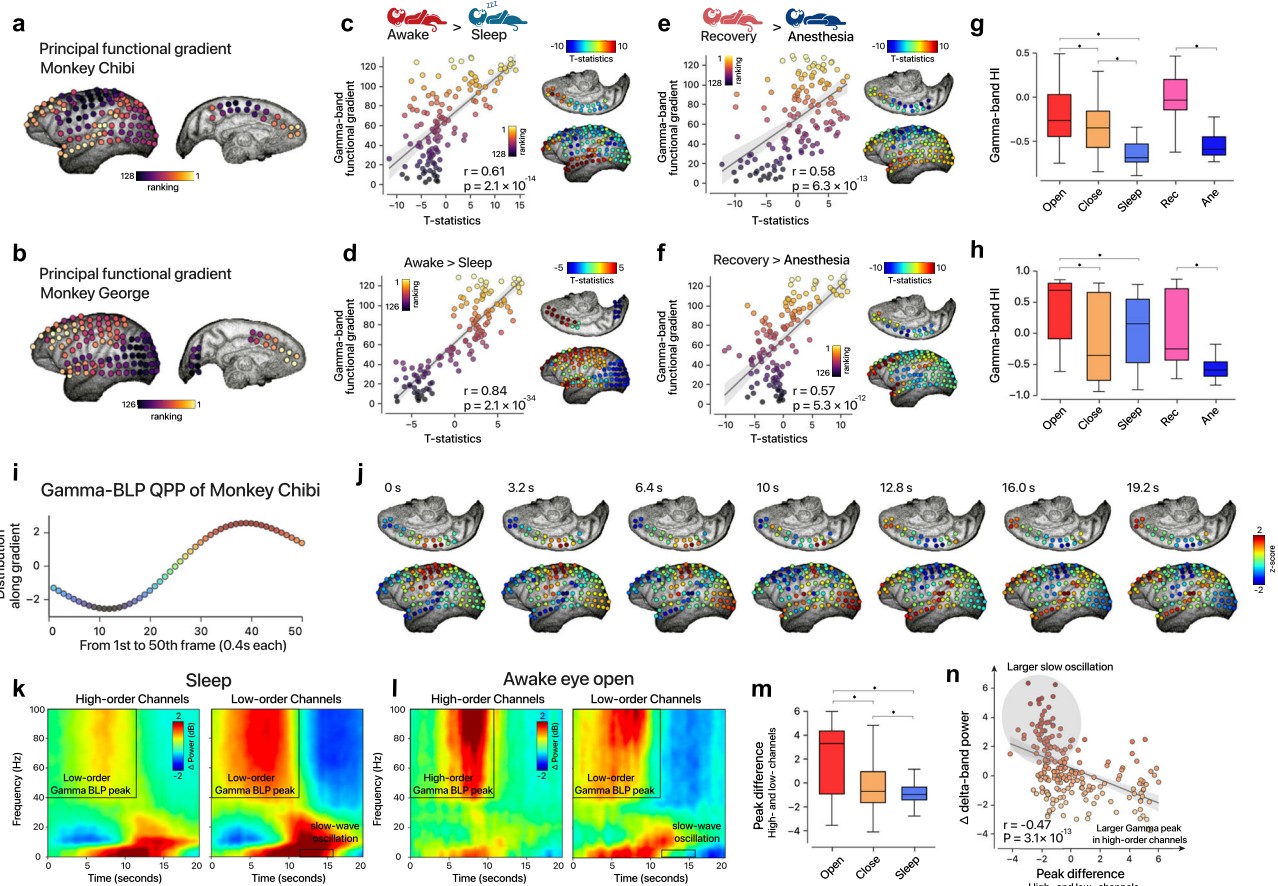

**Fig. 6 | Hierarchical dynamics in macaque electrocorticography. a, b** Principal embedding of gamma BLP connectome for Monkey *Chibi* and Monkey *George*. For this visualization, the original embedding value was transformed into a ranking index value for each macaque. **c, d** Cortex-wide unthresholded *t*-statistical map of the sleep effect for two monkeys. The principal functional gradient spatially associated with the sleep altered pattern (*Chibi*: n = 128 electrodes; *George*: n = 126 electrodes; Spearman rank correlation). Error band represents 95% confidence interval. **e, f** Cortex-wide unthresholded *t*-statistical map of anesthesia effect for two monkeys. Principal functional gradient correlated with anesthesia-induced pattern (*Chibi*: n = 128 electrodes; *George*: n = 126 electrodes; Spearman rank correlation). Error band represents 95% confidence interval. **g, h** The hierarchical index was computed for a 150-s recording fragment and can distinguish different conscious states (*P < 0.01, two-sided *t*-test). From left to right: eyes-open waking, eyes-closed waking, sleeping, recovering from anesthesia, and anesthetized states (*Chibi*: ns = 60, 55, 109, 30, 49 respectively; *George*: ns = 56, 56, 78, 40, 41,

respectively). **i** A typical cycle of gamma-BLP QPP in Monkey C; *x*-axis: temporal frames (0.4 s each), y-axis: dot product of gamma-BLP values and principal functional gradient. The box's midline represents the median, and its lower and upper edges represent the first and third quartiles, and whiskers represent the 1.5 × interquartile range. **j** Representative frames across 20 s. For better visualization, the mean value was subtracted in each frame across the typical gamma-BLP QPP template. **k, l**, Spectrogram averaged over high- and low-order electrodes (top 20%: left; bottom: right) in macaque C across several sleep recording (**k**) and awake eyes-open recording sessions. **m** Peak differences in gamma BLP between high- and low-order electrodes differentiate waking and sleeping conditions (*Chibi*, *P < 0.01; two-sided *t*-test; eye-opened: n = 213; eye-closed: n = 176; sleeping: n = 426). **n** The peak difference in gamma BLP (in the initial 12 s) predicts the later 4 s nonoverlapping part of the change in average delta power across the cortex-wide electrodes (Monkey *Chibi*: awake eye-closed condition, Pearson correlation). Error band represents 95% confidence interval for regression.

and are suggested to be involved in arousal modulation[57]. Specifically, the resemblance between the gamma-BLP QPP and the SST average patterns is more evident during eye-closed and sleeping conditions (Supplementary Fig. 11c, d) when the SST pattern was reported to be more stereotypical[57]. As a recent work suggesting the gamma component of SSTs to be a traveling wave[27], it seems that enhanced cortical excitability and the emergence of slow oscillations tend to couple together, resulting in the gamma and delta components in both the gamma-BLP QPP and SST. Unlike the anticorrelation in the fMRI QPP pattern[26], the average gamma-BLP signals of low- and high-order regions primarily increased in the first half (Fig. 6k, l). Using the peak difference between the low- and high-order regions, we observed that a relatively larger gamma BLP tended to emerge in the high-order cortex during higher levels of consciousness (Fig. 6m). Remarkably, this peak difference was predictive of subsequent global changes in delta power during eye-closed and sleeping conditions when SST-like patterns were evident ("Methods"; Fig. 6n). These results indicate a

heterogeneous regulation of the gamma-BLP wave in distinct states of consciousness: a higher gamma BLP propagating (from the unimodal) to the high-order cortex is associated with the subsequent emergence or suppression of cortical slow oscillations.

## Implication of histaminergic system
Previous animal studies have suggested that the ascending reticular activating system (ARAS) plays a crucial role in supporting consciousness[58,59]. Therefore, we hypothesize that specific ARAS neurotransmitter circuits (such as the histaminergic, cholinergic, and noradrenergic systems) may be preferentially involved in hierarchical heterogeneity across the cortex. To test the hypothesis, we performed a cross-modal analysis to search for genetic transcriptomes that were unimodally-transmodally distributed based on the Allen Human Brain Atlas (Methods). While this approach cannot reveal a causal link, it has the potential to yield plausible biological mechanism for macroscale imaging markers widely utilized by recent studies[60–63].

After identifying the top associated genes (Supplementary Tables 2 and 3), we found that the *HDC* (histidine decarboxylase, which is the unique enzyme catalyzing the decarboxylation of histidine to form histamine) gene was one of the most prominent genes (Fig. 7a-b-d, ranked 3rd). The leading position of the *HDC* gene was well replicated using another two independent analysis pipelines (Supplementary Tables 4 and 5): (i) It ranked 5th based on the pipeline by Anderson and colleagues[23]; (ii) it showed the 9th largest variance by Neurovault platform. Moreover, the spatial distributions of the expressions of the *HDC* and *HRH1* (histamine receptor H1, administration of histamine or H1 receptor agonists can induce wakefulness) genes are highly correlated (Fig. 7c, $P_{perm} < .0001$, see "Methods"). It is evident that histaminergic system help sustain wakefulness. The tuberomammillary nucleus (TMN) in the posterior hypothalamus is the major source of brain histamine and widely project histaminergic neurons to the cerebral cortex[64,65]. Therefore, the result implies that the molecular genetic basis of histaminergic function is spatially nonuniform along the cortical hierarchy, and we hypothesize that this relationship may extend to the TMN connectome architecture. To test this hypothesis, we applied a recently developed hypothalamus atlas[66] and high-resolution 7T resting-state fMRI data from the HCP (Dataset 8) to calculate group-level functional connectome pattern between the TMN and the cortical regions. As a result, we found that the TMN exhibited the most prominent spatial association with the principal functional gradient (Fig. 7e) across 13 hypothalamic regions, followed by the preoptic-anterior hypothalamus and dorsomedial hypothalamus, both of which play a role in sleep-wakefulness regulation[67,68]. These results, derived from both transcriptome and functional connectome data, provide preliminary evidence linking the histaminergic system to hierarchical dynamics across the neocortex.

## Discussion

In this study, we revealed a fundamental yet simple phenomenon by utilizing a variety of experimental paradigms (i.e., sleep, anesthesia, drowsiness, psychedelia, and psychiatric disorders), designs (intra- and inter-subject variability), timescales (changed over the course of several minutes or more than one year, i.e., the *MyConnectome* Project), imaging modalities, and species: the shifting of global states of consciousness is along a hierarchical continuum of cortical neural variability (Fig. 8a). Adhering to this principle, the multidimensional spatiotemporal patterns of cortical activity can be mapped to a low-dimensional signature (Fig. 8b), allowing for individual-level characterization of different states of consciousness. The signature exhibits significant elevations in potentially abnormal states of consciousness such as psychedelia and in individuals with psychiatric disorders (Fig. 8c). Subsequently, we show that the hierarchical signature aligns with complex patterns of functional coordination and diversity underpinning vivid wakefulness (Fig. 8d, left). Furthermore, we suggest that the hierarchical heterogeneity is modulated by spatiotemporal waves of cortical activities, as well as likely involvement of the histaminergic system (Fig. 8d, right). Combining behavioral, neuroimaging, electrophysiological, and transcriptomic evidence, our results suggest that global state of consciousness is supported by efficient hierarchical processing that can be constrained along a low-dimensional macroscale gradient.

Although most theories of consciousness primarily focus on conscious contents[69] (subjective experience), exploring global states can further our comprehension of how neural mechanisms support various conscious experiences in a general way[1], thereby providing new clues for the development of theoretical frameworks. With regard to the ongoing debate among theories of consciousness concerning the neuroanatomical location of consciousness[1,45,70] (i.e., whether it is situated in the frontal or posterior regions of the brain), our data indicates that, at least at the global state level, consciousness may not be dependent on a specific location in Euclidean space,

but rather associates with low-dimensional computational patterns in topological space. The observed cortical topology suggest that global states of consciousness probably rely on a global availability of hierarchical information processing, particularly in top-down modulation from its high-order extreme. We speculate that this hierarchical and top-down mechanism manifest a basic, general principle that is supported by different theories of consciousness from various perspectives. According to the higher-order theory[71] (HOT), conscious experience relies on meta-representations, where lower-order representations of perceptual signals can only be integrated into conscious perception when they are targeted by higher-order meta-representations. Therefore, effective hierarchical processing would play a crucial role in ensuring the integrity of meta-representational operations. Global neuronal workspace theory[72] (GNW) predicts the existence of a set of interconnected cortical neurons (workspace), which intrinsically situate in higher hierarchies to receive bottom-up information and can flexibly mobilize or suppress local processors through descending projections. The workspace neurons can select, amplify, and widely broadcast information to other processors, thus rendering it consciously accessible. Because the GNW does not specialize in localization but rather assumes a broadly distributed workspace, it is consistent with our results, suggesting that the workspace may be mapped continuously across the cortical processing hierarchy; the top-down feedback mechanism is also inferred as indispensable for a conscious system in *Integrated Information Theory*[43] (IIT), which defines consciousness as the quantity of irreducible integrated information generated from complex systems. On the other hand, the analysis of time-resolved connectome reconfiguration accords with the concept of the dynamic core hypothesis[44] (an early formulation of IIT), which highlights both neural integration and complexity for supporting consciousness.

We observed that the signature was significantly elevated in the presence of psychedelics and was associated with positive symptoms of psychosis, such as hallucinations and delusions. Building on previous research indicating that top-down mechanisms interact with sensory signals along a cortical hierarchy to shape perception[3], we postulate that this signature may serve as a marker of hierarchical top-down processing in healthy individuals, whose fluctuation would fall within a certain range. Deviations from this range may lead to aberrant conscious processing. For instance, prior studies have demonstrated that individuals with psychosis tend to rely more heavily on top-down processing from prior knowledge[3,73], while in contrast, top-down processing is impaired in disorders of consciousness[74]. Nevertheless, further research is necessary to validate this postulation, by utilizing a perceptual task paradigm and examining data from individuals with and recovering from disorders of consciousness.

Recent studies have unified spatiotemporally recurring waves[26,27,57] observed in BOLD and gamma BLP signals as an intrinsic physiological process associated with arousal. Extending previous work, we suggest that the infra-slow propagation wave as a heterogenous event relating to global states of consciousness. Specifically, during high vigilance, the waves were more spatiotemporally stereotypical. In contrast, even in the presence of larger fluctuations that started in the sensory cortices, the propagation appeared to be interrupted in the higher-order regions. We postulate that such slow, autonomic processing stabilizes wakefulness mainly by modulating the large-scale cortical activities of the higher-order association cortices. The failure of this modulation may facilitate the transition toward unconsciousness and promote slow oscillations relating to sleep and memory consolidation. Therefore, the global coordination activity would bifurcate into a typical mode (i.e., anticorrelation organization) and an atypical mode (i.e., increased delta power in SSTs) depending on the higher-order networks. This accords with several previous observations, such as larger sensory-dominant fluctuations

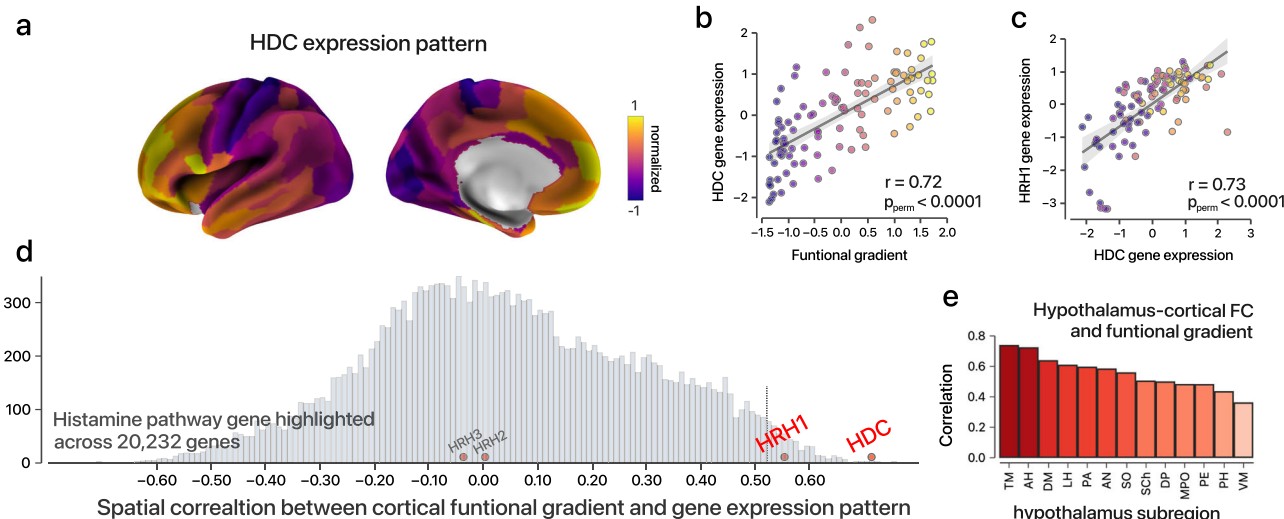

**Fig. 7 | Histaminergic system and hierarchical organization across the neo-cortex. a** Z-normalized map of the *HDC* transcriptional landscape based on the Allen Human Brain Atlas and the Human Brainnetome Atlas[109]. **b, c** Gene expression pattern of the *HDC* is highly correlated with functional hierarchy ($r = 0.72$, $P_{perm} < .0001$, spearman rank correlation) and the expression of the *HRH1* gene ($r = 0.73$, $P_{perm} < .0001$, spearman rank correlation). Error band shows 95% confidence interval for regression. Each region's color is defined by its average principal gradient, and a plasma colormap is used for visualization. **d** Distribution of Spearman's Rho values across the gene expression of 20232 genes and the

functional hierarchy. *HDC* gene and histaminergic receptors genes are highlighted. **e** Spatial association between hypothalamic subregions functional connection to cortical area and functional gradient across 210 regions defined by Human Brainnetome Atlas. The tuberomammillary nucleus showed one of the most outstanding correlations. From left to right: tuberomammillary nucleus (TM), anterior hypothalamic area (AH), dorsomedial hypothalamic nucleus (DM), lateral hypothalamus (LH), paraventricular nucleus (PA), arcuate nucleus (AN), suprachiasmatic nucleus (SCh), dorsal periventricular nucleus (DP), medial preoptic nucleus (MPO), periventricular nucleus (PE), posterior hypothalamus (PH), ventromedial nucleus (VM).

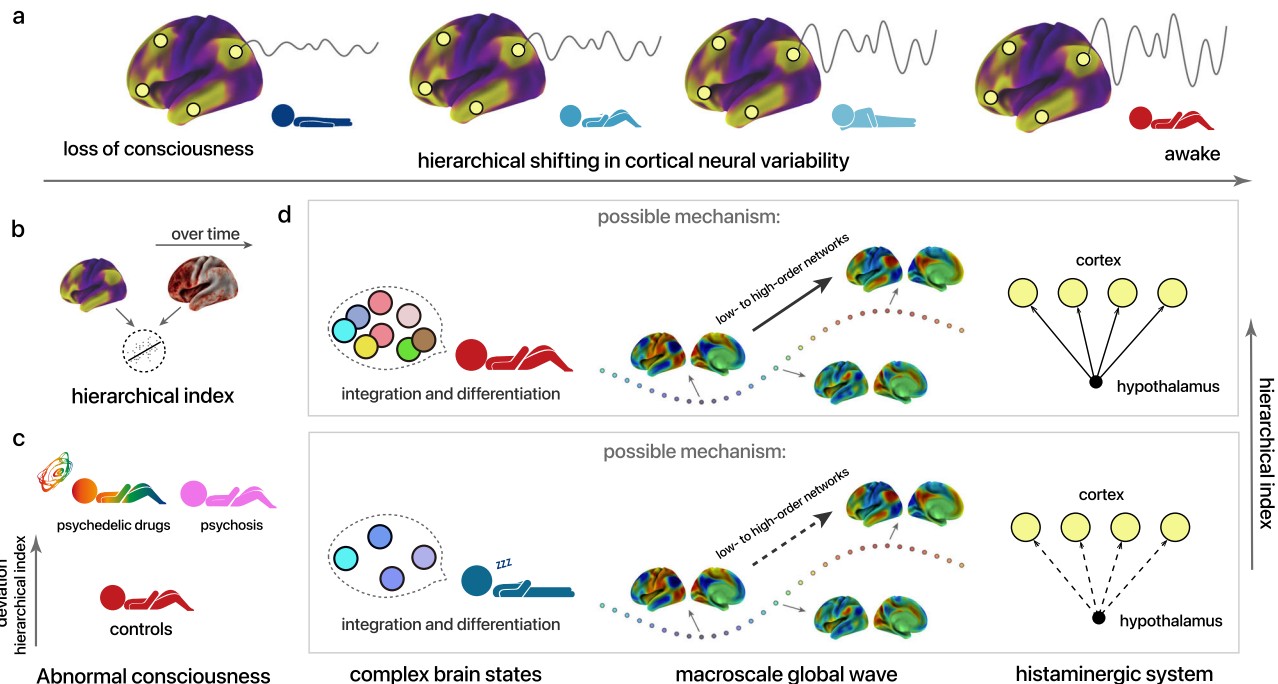

**Fig. 8 | A summary model of findings in this work. a** A schematic diagram of our observations based on a range of conditions: Altered global state of consciousness associates with the hierarchical shift in cortical neural variability. Principal gradients of functional connectome in the resting brain are shown for both species. Yellow versus violet represent high versus low loadings onto the low-dimensional gradient. **b** Spatiotemporal dynamics can be mapped to a low-dimensional hierarchical score linking to states of consciousness. **c** Abnormal states of

consciousness manifested by a disruption of cortical neural variability, which may indicate distorted hierarchical processing. **d** During vivid wakefulness, higher-order regions show disproportionately greater fluctuations, which are associated with more complex global patterns of functional integration/coordination and differentiation. Such hierarchical heterogeneity is potentially supported by spatiotemporal propagating waves and by the histaminergic system.

(dominating global signal) during drowsiness[75–77] and elevated activity of default mode areas preceding pupil dilation[78].

As an evolutionarily conserved system, histaminergic neurons play a prominent role in sustaining wakefulness[64] through their projections from hypothalamic TMN to a wide array of cortical and subcortical regions. A recent study found that histaminergic neurons can broadcast dual inhibitory-excitatory signals throughout the neocortex through GABA-histamine axons[79]. In this work, we observed a spatial correlation between the low-dimensional functional hierarchy, histaminergic molecular markers, and the TMN functional connectome, suggesting a potential role of the histaminergic system in modulating heterogeneous dynamics across the cortex. Further evidence is needed to establish direct causality, such as brain stimulation in animal models.

The principal gradient of dense functional connectome and T1w/T2w ratio maps were utilized as surrogates of cortical hierarchy. Both had a fine resolution and had previously been shown to be coupled with the anatomy of the hierarchy, such as defined by feedforward and feedback projections[34]. To better reflect hierarchical organization in the functional domain, the principal gradient map was chosen in our main analyses. A recent fMRI study characterized a hierarchical signature based on transfer entropy (measuring information flow), which yields a similar macroscale pattern[80]. Unlike the abovementioned anatomically or functionally defined hierarchies, a recent whole-brain computational modeling study[81] reported an altered diversity of 'intrinsic ignition' (capability of a region to propagate neuronal activity) induced by anesthesia as an indicator of hierarchical disruption. Previous task-based studies have supported that the hierarchical sensory processing is disrupted during loss of consciousness, as evidenced by the absence of higher-order cortical responses to auditory stimuli and a hierarchical attenuation of language processing[82]. Interestingly, after reviewing previous fMRI studies which tried to localize a consciousness-specific region, we found that most of the reported regions were distributed in areas with extreme gradients, regardless of their anatomical nonconvergence (Supplementary Table 6).

Several limitations should be noted. First, the level of consciousness is an ambiguous construct[83], and our study neglected consciousness as a multidimensional phenomenon. It is also an oversimplification to attempt to unify physiological sleep, pharmacological sedation, and psychedelic states into the same framework, and mechanisms that might distinguish them were not fully investigated. It is worth noting that the effect size of the vigilance map is relatively low. This could potentially be increased by providing additional information, such as using self-reported questionnaires to assess whether participants were prone to becoming sleepy while scanning. Second, our time-resolved state and recurring QPP analyses in human were restricted to HCP fMRI data, which might introduce biases into our hypothetical explanation of other conditions. Nevertheless, the analysis of the QPP has been extended to the ECoG data of macaques in different conditions. Third, we did not address how choices of different MRI scanners, head coils, or acquisition parameters could systematically influence the spatial distribution of temporal noise and the hierarchical index, which could limit the potential application of cross-scanner generalization.

Collectively, our work suggests that global states of consciousness can be conceptualized as a topological signature derived from spontaneous cortical activities. This is supported and validated through multiple lines of evidence, including cross-species generalization, abnormal conditions, functional integration and diversity, as well as potential mechanisms involving spatiotemporal waves and the histaminergic system.

## Methods
### Data and preprocessing
**Dataset 1: dexmedetomidine-induced sedation.** Twenty-one healthy male volunteers (age: $26.4 \pm 2.1$ years; right-handed; body mass index: $21.7 \pm 1.9$) were recruited from Capital Medical University, Beijing,

China. To ensure the safety of the experiment, all included volunteers were at an American Society of Anesthesiologists (ASA) physical status I or II. The exclusion criteria included: (1) the presence of metal implants in the body, (2) the presence of intracranial lesions or systemic comorbidities, (3) a history of general anesthesia, (4) a history of drug abuse or alcohol abuse, (5) an allergy to dexmedetomidine, (6) claustrophobia, and/or (7) left-handedness. The experiment protocol was approved by the Institutional Review Board of Beijing Tiantan Hospital, Capital Medical University, China. After being informed of the relevant details of the study, all subjects signed written informed consent to their participation. More details can be found in our registered clinical trial at clinicaltrials.gov (registration number: *NCT03343873*). We did not consider sex-specific effects in our study, as limited prior research has indicated significant sex differences in the global state of consciousness.

All volunteers were instructed to fast before the experiment (at least 6 h from solids and 2 h from liquids). Due to its minor effect on respiratory inhibition, dexmedetomidine was used as the anesthetic drug to induce the sedation states. Specifically, dexmedetomidine was administered through an intravenous catheter inserted into a vein in the right hand, initially with a bolus at 1 µg/kg/h over 15 min and subsequently by a 0.6 µg/kg/h continuous intravenous infusion to maintain sedation. Simultaneous, continuous monitoring, including heart rate, arterial pressure, pulse oxygen saturation, respiratory rate, and electrocardiography, was applied throughout all the experiments (Supplementary Table 7). The Observer's Assessment of Alertness/Sedation (OAA/S) scale and Ramsay sedation scale (RSS) were used to evaluate the sedation level. The subjects were judged to be awake or to be fully recovered if they responded readily to verbal commands or to their name being spoken in a normal tone (RSS score of 2; OAA/S scale score of 5). Subjects were judged to be under moderate sedation if they had lethargic responses to verbal commands and to their name being spoken in a normal tone or if they responded only after their name was called loudly (RSS score of 3–4; OAA/S scale score of 3–4). RSS and OAA/S scale verbal commands were repeated twice during each assessment. All the subjects wore headphones throughout the experiment and were thus spoken to through an MRI speaker. All communications occurred between MRI acquisitions, and the subjects were instructed to respond verbally. These operations were conducted by two certified anesthesiologists with complete resuscitation equipment available.

Resting-state fMRI data were recorded using an echo-planar imaging (EPI) sequence in a Siemens Medical Systems Prisma 3.0 T MRI system at Beijing Neurosurgical Institute. The scanning parameters were as follows: TR = 2000 ms; TE = 30 ms; field-of-view = 192 × 192 mm$^2$; acquisition matrix = 64 × 64; flip angle = 75°; slice thickness = 4 mm; voxel size = 3 × 3 × 4.4 mm$^3$. The subjects were scanned for 6 min 40 s across each of the three conscious states: normal wakefulness (RSS score, 2; OAA/S Scale score, 5), moderate sedation (RSS score, 3–4; OAA/S Scale score, 3–4), and recovery of consciousness (RSS score, 2; OAA/S Scale score, 5).

The fMRI data were preprocessed using an open-assess MATLAB toolbox BRANT v3.35[84]. The standardized pipeline included: (i) slice timing correction, fixing the temporal shifts of different slices; (ii) within-subject EPI image realignment, estimating and spatially correcting for head motions; (iii) spatial normalization to the Montreal Neurological Institute (MNI) standard space and resampling to a 3 × 3 × 3 mm$^3$ resolution; (iv) nuisance regression, regressors included linear trends, averaged signals, and their first-order temporal derivatives within the white matter and cerebrospinal fluid regions, as well as Friston's 24 head motion parameters[85] (3 rotation and 3 translation parameters, 6 parameters one time point before, and the 12 corresponding squared items); (v) bandpass filtering, the residuals of the regression models were bandpass-filtered (0.01–0.08 Hz) to further suppress low-frequency drifts and physiological noises such as breathing and heartbeat.

**Dataset 2: 3T resting-state data.** The resting-state fMRI data were acquired as part of the HCP S1200 release[86]. We analyzed functional imaging data acquired from a 3T Siemens Skyra scanner using multi-band EPI. Participants with 4 runs of resting-state data were included: 4 runs in REST1 and REST2 session with right-to-left (run 1 and run 3) and left-to-right (run 2 and run 4) phase encoding. Each run lasted 14 min and 33 s (TR = 720 ms; TE = 33.1 ms; field-of-view = 208 × 180 mm²; acquisition matrix = 104 × 90; flip angle = 52°; slice thickness = 2 mm; voxel size = 2 × 2 × 2 mm³). Within-scanner sleep during resting-state scanning is commonly reported. Based on a previous study that used combined EEG–fMRI trained classifiers during sleep stages, nearly one third of subjects were determined to have not maintained their wakefulness for over 3 min[29]. Thus, we speculate that subjects were predisposed to decreased alertness when close to the end of scanning. Although such speculation cannot be generalized to all individuals, such an effect could be overwhelming at the population level.

Resting-state fMRI data was processed based on the HCP-pipeline using the CIFTI grayordinate-based framework, and spatially structured physiological noise was corrected using the ICA + FIX method, which can be assessed in the HCP repository. Such an ICA-based approach[87] is effective for cleaning structured noise and provided a different denoising strategy beyond nuisance regression in this research. The signal was further cleaned with a Butterworth filter within 0.01–0.08 Hz. Finally, 982 individuals (462 male/520 females; mean age, 28.7 ± 3.7 years) with complete four resting-state runs without large motion (mean framewise distance <0.3 mm) were included in our study. To ensure that the results obtained from the surface domain and volume space were comparable, we also included additional ICA-FIX-denoised resting-state fMRI volumetric data of 100 unrelated subjects from the Extended HCP rfMRI dataset (Supplementary Fig. 12). The description of the behavioral phenotypes in HCP was detailed elsewhere[25]. Items belonging to alertness (Pittsburgh Sleep Questionnaire), in-scanner task performance, and cognition were selected for further analyses. To summarize the overall reaction time of the in-scanner tasks, the principal component analysis was conducted to items that tapped to reaction times (such as median reaction time during tasks; with more than 950 individuals available). Only the first PC was used because it explained 30.8% of the variance and positively correlated with all the items included, whereas the second PC only explained 17% of the variance.

**Dataset 3: Simultaneous EEG–fMRI during sleeping.** Young healthy volunteers (18–25 years) were enrolled via online advertisement. To ensure their safety, participants with a history of any psychiatric or neurological illness were excluded from the experiment; and participants with normal sleep quality were considered according to the Pittsburgh Sleep Quality Index (PSQI). Each subject provided written informed consent after a detailed explanation of the study protocol. No intake of alcohol or caffeine was allowed on the scanning day. The sleeping experiment started at approximately 01: 00 local time. Each participant was asked to lie down and fall asleep on the scanner bed. The experimenter checked the participant's sleeping condition through a microphone every 8 min. The response of the participant was recorded through an MRI-compatible multiple button-press box. Participants who failed to sleep did not continue with the experiment. The 2 h of sleeping scanning (nine 12.5 min runs) did not start until the participants were nonresponsive. All experiments were in accordance with the Declaration of Helsinki. The study protocol was approved by the Ethics Committee of Southwest University, China.

Sleep fMRI data were acquired using a 3T Siemens Trio scanner at the Sleep and Neuroimaging Center at Southwest University, Chongqing, China. Head movements were minimized by using a cushioned head fixation device. A T2-weighted gradient echo-planar imaging (EPI) sequence was applied, with the scanning parameters as follows: TR = 1500 ms; TE = 29 ms; field-of-view = 192 × 192 mm²;

acquisition matrix = 64 × 64; flip angle = 90°; slice thickness = 5 mm; voxel size = 3 × 3 × 5.5 mm³. To access sleep stages during fMRI scanning, simultaneous EEG–fMRI recordings were conducted by a 32-channel MRI-compatible Brain Products system (BrainAmp MR plus, Brain products, Munich, Germany). The channel position was in accordance with the international 10/20 system, and all impedances were below 10 kΩ. The FCz point was used as the reference for online signal collection, and the sampling rate was 5 kHz. To ensure the temporal stability of the EEG acquisition in relation to the switching of the gradients during the MR acquisition, we used a SyncBox (SyncBox MainUnit, Brain Products GmbH, Gilching, Germany) to synchronize the amplifier system with the MRI scanner's system. The EEG amplifier was a non-magnetic MRI-compatible EEG system and was charged by a rechargeable power pack placed outside the scanner bore. The amplified and digitized EEG signal was transmitted to the recording computer with fiber optic cables. The computer was placed outside the scanner room, and the adapter (BrainAmp USB-Adapter, Brain products, Gilching, Germany) converted the optical signal into an electric signal.

Sleeping fMRI data was preprocessed based on the BRANT standardized pipeline. The original EEG recording was processed using the Vision-Analyzer software (Version 2.0, Brain Products, Inc., Munich, Germany). Appling the average template subtraction procedure, we first removed the fMRI gradient and ballistocardiographic artifacts from the original EEG signal. Then the EEG data were downsampled to 250 Hz and digitally filtered within the 0.1–45 Hz band using a Chebyshev II-type filter. Furthermore, the residual fMRI gradient and ballistocardiographic artifacts, ocular artifacts, and the muscle activity artifact from the EEG data were eliminated through temporal independent component analysis (ICA). After removing severe artifacts, six EEG channels (O1, O2, F3, F4, C3, C4) were selected for manual sleep staging by two experts, with a time window of 30 s. Each segmented epoch was classified into awake, N1, N2, or slow-wave sleep condition according to the 2017 AASM manual[88]. Then, the 6 (out of 22; 1 male/5 females; mean age, 21.6 ± 1.3 years) volunteers with the most consistent sleeping trajectories, as labeled by two experts, were included in the further analyses.

To strengthen the reliability of the sleep-related results, we included a publicly available simultaneous EEG–fMRI dataset[36,89] from OpenNeuro platform for validation (ds003768). Briefly, the dataset has 33 healthy participants (17 males/16 females; mean age, 22.1 ± 3.2 years) collected at Pennsylvania State University. fMRI data was acquired by an EPI sequence: TR = 2100 ms, TE = 25 ms, slice thickness = 4 mm, slices = 35, FOV = 240 mm, in-plane resolution = 3 mm×3 mm. The fMRI data in sleep sessions was preprocessed using the BRANT standardized pipeline mentioned above. EEG data were collected using a 32-channel MR-compatible EEG system from Brain Products, Germany. The sleep staging was performed by a Registered Polysomnographic Technologist. The sleep staging was conducted by a Registered Polysomnographic Technologist. Similarly, the evaluation was based on six EEG channels (O1, O2, F3, F4, C3, C4) with each epoch lasting 30 s. Some epochs which yielded uncertain results were marked as "uncertain" and those with artifacts too large for reliable scoring were labeled as "unscorable". The individuals with more than 10 epochs labeled with "unscorable" or "uncertain" were excluded. In contrast to our dataset (collected after midnight), sleeping states in this dataset were mainly attributed to wakefulness, N1, and N2 (with only sporadic attributions of N3 stage), resulting in a more limited range of variation. As an illustration, sub-08 was predominantly assigned to wakefulness, with only sporadic occurrences of N1 stage. The sleep stage was encoded as ranked number (wakefulness: 0, N1 stage: -1, N2 stage: -2, slow-wave sleep stage: -3). To facilitate reliable analysis at the individual level, we selected 6 individuals with the largest variance in sleep stages for validation (subjects 1–6: sub-09, sub-10, sub-15, sub-24, sub-27, sub-28; Supplementary Fig. 6).

**Dataset 4: the MyConnectome project.** We downloaded the raw resting-state fMRI and behavioral data of the *MyConnectome* Project[37] from OpenNeuro database (ds000031), comprising a deeply sampled phenotyping of a single individual (a Caucasian male; 45 years old at study; right-handed) over a period of 18 months. Scan sessions 14–104 were included in the analysis (84 sessions according to a previous work[37]), which were from the original acquisition period of the study using a Siemens Skyra 3 T scanner at the University of Texas. In each session, 10 min of resting-state data were acquired using a multiband EPI sequence, with the following parameters: TR = 1160 ms; TE = 30 ms; field-of-view = 240 × 240 mm²; acquisition matrix = 96 × 96; flip angle = 63°; voxel size = 2.4× 2.4 × 2 mm³, 68 slices (64 slices after session 27). The fMRI data was preprocessed using the BRANT standardized pipeline but without the option of slice timing correction.

The *Myconnectome* Project provides a unique opportunity to study the longitudinal, intra-individual effects of physical and psychological factors on brain function. Scans were mostly performed on either Tuesdays or Thursdays at 0730 h. Before each Tuesday scan, the subject fasted and had no caffeine intake; on Thursdays, the subject was always fed and caffeinated. Thus, we inferred that the subject was in a relatively low state of energy both physically and psychologically during the Tuesday scans. To quantify the brain state linking to a neuroimaging session, we also assessed the self-reported Positive and Negative Affect Schedule (PANAS-X) questionnaires, which were administered after each scan. As referenced in a previous work[90], the arousal-related factors were categorized as those associated with fatigue (average scores: "drowsy": Q28; "sleepy": Q57; "sluggish": Q58; and "tired": Q62) or heightened attention (average scores: "attentive": Q11; "concentrating": Q18; and "lively": Q43) and were further compared across the different days.

**Dataset 5: psychedelic state.** The functional imaging data for the psychedelic state was collected and analyzed in a previous study[41] and was recently made available on the OpenNeuro database (ds003059). This resource included 15 healthy volunteers (11 male/4 females; mean age, 30.5 ± 8.0 years) who were all examined under conditions of both placebo and LSD administration. In these two conditions, each individual was scanned three times: the first and third runs were a common eyes-closed resting-state paradigm, and the second scan was accompanied by listening to music. More information about data acquisition and preprocessing can be found elsewhere[41]. Briefly, a gradient echo planer imaging sequence was used to acquire BOLD-weighted fMRI data (TR = 2000 ms, TE = 35 ms, field-of-view = 220 mm, acquisition matrix = 64 × 64, parallel acceleration factor = 2, flip angle = 90°). 35 oblique axial slices were acquired in an interleaved fashion, each 3.4 mm thick with zero slice gap (3.4 mm isotropic voxels). Each BOLD scan lasted 7 min. Because the released data did not include raw functional MRI images, we used the preprocessed data directly. Notably, this preprocessed pipeline rigidly controlled head motion using de-spiking, motion-related nuisance regression, and bandpass filtering (0.01–0.08 Hz) and has been demonstrated to have minimal motion effect on functional connectivity[41].

**Dataset 6: neuropsychiatric disorders.** We analyzed data from the UCLA Consortium for Neuropsychiatric Phenomics[42], which was obtained from the OpenNeuro database (ds000030). Specifically, 117 healthy subjects (62 males/55 females; mean age, 30.1 ± 8.6 years), 47 individuals with schizophrenia (35 males/12 females; mean age, 36.5 ± 8.8 years), 45 individuals with bipolar disorder (26 males/19 females; mean age, 35.0 ± 9.0 years), and 41 individuals with ADHD (21 males/20 females; mean age, 32.4 ± 10.5 years) were included in our study. All participants were asked to give written informed consent for their inclusion, following procedures approved by the Institutional Review Boards at UCLA and the Los Angeles County Department of Mental Health. All patients underwent behavioral and symptom assessments, including the Brief Psychiatric Rating Scale (BPRS). Resting-state data was scanned using an echo-planar imaging (EPI) sequence from a 3 T Siemens Syngo scanner at UCLA (TR = 2000 ms, TE = 30 ms, field-of-view = 192 mm, acquisition matrix = 64 × 64, flip angle = 90°, voxel size = 3× 3 × 4 mm³, 34 slices), lasting for 304 s. In addition, 92 individuals diagnosed with schizophrenia (57 males/35 females; mean age, 27.4 ± 6.7 years) and 98 healthy controls (53 males/ 45 females; mean age, 25.8 ± 5.3 years), whose data were collected from Peking University Sixth Hospital, China, were further analyzed for an independent validation. A consensus diagnosis of schizophrenia was made by two experienced senior psychiatrists according to the Diagnosis and Statistic Manual of Mental Disorders, fourth edition (DSM-IV) criteria for schizophrenia or schizophreniform disorder and finally diagnosed with schizophrenia after being followed up for at least six months. All patients had significant positive symptoms: more than four on at least three of seven positive items based on the Positive and Negative Syndrome Scale. The study protocol was approved by the Medical Research Ethics Committees of the local hospitals and written informed consent was obtained from all participants and/or their legal guardians. Detailed clinical information was provided in our previous work[91]. Eight minutes of resting-state fMRI data were acquired using an EPI sequence with the following parameters: TR = 2000 ms; TE = 30 ms; acquisition matrix = 64 × 64; flip angle = 90°; slice thickness = 4 mm; voxel size = 3.4375 × 3.4375 × 4.6 mm³. The fMRI preprocessing (UCLA and PKU6) is based on the BRANT platform and can be found elsewhere[91]. Subjects with high motion during the scanning were excluded (translation > 3 mm or rotation > 3°).

**Dataset 7: macaque EcoG recordings.** We downloaded the macaque electrophysiological recordings from the NeuroTycho resource. The acquisition details and experimental procedures can be found in previous publications[55,56]. Briefly, a chronically implanted customized 128-channel EcoG electrode array was employed to record neural activity in the left hemisphere. The sampling rate was 1 kHz using the Cerebus data acquisition system (Blackrock, UT, USA). We analyzed the EcoG data from two adult macaque monkeys (*Chibi: Macaca fuscata*; *George: Macaca mulatta*) acquired under different conditions: the awake, eyes-closed resting-state (*Chibi:* in 12 sessions, lasting 150 min; *George:* in 13 sessions, lasting 153 min), the awake, eyes-opened resting-state (*Chibi:* in 12 sessions, lasting 164 min; *George:* in 12 sessions, lasting 148 min), the sleeping state (*Chibi:* in 7 sessions, lasting 326 min; *George:* in 6 sessions, lasting 233 min), the anesthetized state (*Chibi:* in 8 sessions, lasting 132 min; *George:* in 9 sessions, lasting 117 min), as well as the eyes-opened, recovered state (*Chibi:* in 7 sessions, lasting 96 min; *George:* in 9 sessions, lasting 109 min). Some sessions were labeled with more than one condition, and each condition lasted more than 5 min. Under the sleep state, the experimental room was kept quiet and dark for up to 1.5 h, and the monkeys were conducted to sit calmly with an eye mask. Slow wave oscillation appeared intermittently during natural sleep. Immediately following the sleep condition, the eyes-closed condition was collected with the light turned on. For the eyes-open waking state, there was no eye mask. In the anesthesia experiment, several agents (ketamine, medetomidine, or propofol) were applied to induce loss of consciousness; and the anesthetized state was collected after a monkey no longer responded to manipulation of the monkey's hand or to touching the nostril or philtrum with a cotton swab.

The line noise was removed at 50 Hz using notch filtering. According to the previous work, two channels from monkey *George* were excluded due to stubborn artifacts[57]. The multitaper spectral estimation was applied to generate spectrograms for 1–100 Hz power in 1 Hz frequency bins. The window size was 1 s (0.2 s step size), and the number of tapers was set to 5. Next, we transformed the power spectrogram into decibels using a logarithmic function and normalized each frequency bin by removing the temporal mean of the power.

Band-limited power (BLP) was calculated by averaging the normalized spectrogram within defined frequency ranges: delta 1–4 Hz; alpha–beta 5–30 Hz; and gamma 40–100 Hz. BLP signals were further bandpass-filtered (0.01–0.08 Hz) using the Butterworth filter, and a forward-backward digital filter was implemented to eliminate any phase delay.

**Dataset 8: High-resolution 7T resting-state fMRI.** To finely characterize the functional connectivity of the hypothalamic system, we applied resting-state fMRI data, which were acquired from a 7T Siemens Magnetom scanner using a multiband sequence in HCP. 177 participants (70 male/107 females; mean age, 29.4 ± 3.3 years) with 4 full resting-state runs were included: 4 runs in REST1 and REST2 session with posterior-to-anterior (run 1 and run 3) and anterior-to-posterior (run 2 and run 4) phase encoding. Each run lasted 16 min (TR = 1000 ms; TE = 22.2 ms; field-of-view = 208 × 208 mm$^2$; acquisition matrix = 130 × 130; flip angle = 45°; slice thickness = 1.6 mm; voxel size = 1.6 × 1.6 × 1.6 mm$^3$). We started our analyses used on 7 T preprocessed fMRI data after ICA-FIX in the HCP repository, and volume-based images were used to cover the full hypothalamus. The preprocessed fMRI data retained a spatial resolution of 1.6 × 1.6 × 1.6 mm$^3$ and was a part of HCP new release in 2018, which had been fixed from the incorrect unwrapped direction in the fMRIVolume pipeline.

A recently developed hypothalamic atlas[66] was applied to delineate the segmentation of the hypothalamus, including 13 hypothalamic structures: anterior hypothalamic area, arcuate nucleus, dorsal periventricular nucleus, dorsomedial hypothalamic nucleus, lateral hypothalamus, medial preoptic nucleus, paraventricular nucleus, periventricular nucleus, posterior hypothalamus, suprachiasmatic nucleus, supraoptic nucleus, tuberomammillary nucleus, and ventromedial nucleus. Considering the potential spatial mismatch of small hypothalamic divisions, we performed a binary dilation for each subregion with a cube connectivity of 1 to moderately expand the shape, resulting in 75 voxels (from 18 voxels, tuberomammillary nucleus) with a spatial resolution of 1.6 × 1.6 × 1.6 mm$^3$. The time series within the dilated hypothalamic mask were averaged. For the cortical regions, we derived voxel-wise time series based on fMRI data smoothed with a 6 mm$^3$ FWHM kernel. All signals were temporally cleaned by a Butterworth filter within 0.01–0.08 Hz. Subsequently, the voxel-wise FC between the hypothalamic divisions and the cerebral cortex was calculated by Pearson's correlation for each run and averaged across all 176 individuals.

**BOLD variability**
It has recently been proposed that moment-to-moment brain signal variability is more informative than static metrics such as mean signal values[13]. We calculated BOLD variability (also termed low-frequency BOLD amplitude) as the standard deviation of temporally filtered time series (0.01–0.08 Hz), at the voxel-/vertex-wise level. The frequency band is a default parameter in the standardized BRANT pipeline. Based on Parseval's theorem, such a temporal metric is mathematically analogous to the amplitude of low-frequency fluctuations (ALFF) calculated in the frequency domain[92] (Eqs.1–2):

$$X(t) = \sum_{k=1}^{N}[a_k \cos(2\pi f_k t) + b_k \sin(2\pi f_k t)] \qquad (1)$$

$$\text{ALFF} = \sum_{k:f_k \in [0.01, 0.08]} \sqrt{\frac{a_k^2(f) + b_k^2(f)}{N}} \qquad (2)$$

We also calculated the fractional amplitude of low-frequency fluctuation[93] (fALFF), which measures the ratio of power in the low-frequency range to total power across the frequency range. The intra-

and inter-subject hierarchical index and spatial distribution based on fALFF are very similar to those based on BOLD variability (Supplementary Fig. 13).

**Consciousness-related cortical patterns**
We tested whether the consciousness-related pattern of the BOLD signal amplitude was consistent across different paradigms. Importantly, to characterize the spatial heterogeneity, rather than using an overall absolute value, the amplitude map was normalized into a z-score at the voxel/vertex level. Specifically, the HCP resting-state fMRI data was analyzed across all the cortical surface vertices in the CIFTI format, and the anesthesia and sleep fMRI data were analyzed across the cortical voxels in MNI space. In this way, three cortex-wide maps were generated and spatially compared using fMRI data collected under anesthesia, during sleep, and in the resting state.

**Dexmedetomidine-induced sedation.** The map in Fig. 1b was formed using the t-statistic values between the z-score BOLD amplitudes in sedation and wakefulness using a paired sample t-test (n = 21, FWHM = 6 mm$^3$). The pattern is highly consistent with that obtained using the contrast between states in sedation and recovery (Supplementary Fig. 1).

**Vigilance decreases.** The map in Fig. 1e was formed using the Spearman's rank r between the z-score BOLD amplitude and the time interval. Practically, each HCP run was divided into 24 nonoverlapping time windows of 50 frames each, and then the pairwise difference in amplitude was calculated. This approach resulted in up to 276 differed maps for each run, with the time interval ranging from 1 to 23 intervals. The larger interval was inferred to have a higher possibility of a decrease in vigilance. Collectively, for each cortical grayordinate, the Spearman's rank r was calculated using 1,084,128 (276 combinations × 4 runs × 982 individuals) points. We also identified 100 most representative subjects (exhibiting the greatest reduction of hierarchical index across four resting-state runs) to generate a new vigilance map (Supplementary Fig. 14), and the pattern was highly comparable to those obtained using all individuals (shown in Fig. 1e).

**Sleep.** The map in Fig. 1h was formed using the Spearman's rank correlation between the z-score BOLD amplitude and the sleep stages manually labeled by an expert (150 s interval; wakefulness: 0, N1 stage: −1, N2 stage: −2, slow-wave sleep stage: −3). Data from six volunteers was concatenated in the temporal resolution of 100 frames (150 s, FWHM = 6 mm$^3$), and the sleep score was calculated by simply averaging across the five labeled epochs of sleep stages (from 0, −1, −2, or −3).

**Statistical testing of spatial correlation**
To test the statistical significance of the spatial correlations, we controlled the effects of spatial autocorrelation using generated null models (a schematic diagram was presented at Supplementary Fig. 4). For analyses in Fig. 1c-f-i as well as Supplementary Fig. 3, we projected the vertex-wise principal functional gradient data onto the 32k_fs_LR sphere space and generated 10,000 spatially rotated null maps for calculating P value in permutation tests. Specifically, the Alexander-Bloch approach[94] was applied to generate spatially constrained null distributions by applying random rotations for spherical projection of the brain. This is a recommend method based on a recent study which evaluating of the effectiveness of 10 null models in mitigating the effects of spatial autocorrelation. For Figs. 7c and 7e, since sampling was not conducted directly in the surface space, we employed the Vázquez-Rodríguez method[95], an adaptation of the Alexander-Bloch approach for parcellated brain data, to generate 10,000 null models and present permutation-based P values.

## Hierarchy analysis

**Definition of cortical hierarchy/gradient.** Cortex-wide maps of the principal FC diffusion values and T1w/T2w ratios were used to characterize the macroscale hierarchical organization of the human brain. The principal FC diffusion, also called principal functional gradient, represents the main area of variance in functional connectivity and is spatially locating along the sensorimotor-to-transmodal axis. The functional gradient was calculated via diffusion map embedding, a non-linear manifold learning approach, based on the HCP S1200 group-average resting-state fMRI dense connectivity. The embedded space was constructed by a random walk process on a pairwise cosine similarity graph of dense coordinate's FC pattern. The algorithm is controlled by the parameter $\alpha = 0.5$, which can balance the global and local relationship between nodes constructed in the embedded space. The eigenvectors of the transition matrix on this graph were defined as diffusion coordinates. The detailed procedures were described in Margulies et al.[32]. In our work, we derived the top nine eigenvectors: the first eigenvector, which explains the most variance, was utilized for most of the analyses, while the 2nd to 9th eigenvectors were used for the control analysis in relation to three consciousness-related maps (Supplementary Fig. 5, the eigenvalues exhibited a sharp drop after the 9th component). The T1w/T2w ratio mapping, which was proposed as an in vivo measurement to index the gray-matter myelin content and anatomical hierarchy, was also downloaded from the HCP S1200 group-average data release.

**Diffusion map for macaque.** To achieve an individual-level principal gradient for the gamma-band connectome, we used the BrainSpace toolbox[96] (version 0.1.1) implemented in Python. Consistent with previous work[27], we analyzed fMRI data obtained during the awake, eye-closed resting state. First, a functional connectivity matrix was established via computing the pairwise Pearson's correlation between the time series of the gamma-band power across channels. A cosine kernel was then used to construct an affinity matrix for the averaged FC map across sessions for each macaque. The diffusion map was conducted using the pipeline described above based on BrainSpace (diffusion embedding, alpha = 0.5, sparsity = 0.9). The generated diffusion maps of the two macaques were visually analogous to those found in a previous work[27]. However, unlike in human fMRI acquisition, the implanted electrodes in the two macaques were sparsely sampled, and there was a considerable disparity between them. As illustrated in Supplementary Fig. 10, macaque Chibi and macaque George have a higher proportion of electrodes distributed in the medial motor and visual cortices, respectively, which may have augmented the internal functional connection ratios of the motor and visual networks to some extent, thus resulting in certain discrepancies in the gradient pattern in the motor and visual cortices. Nevertheless, it is essential to note that both the two cortical maps can reflect the large-scale unimodal-transmodal organization, and thus are highly correlated with the neural variability of gamma-BLP in different conscious states.

**Hierarchical index.** The hierarchical index was defined as the Spearman's r between the principal functional gradient and the normalized low-frequency BOLD amplitude across the entire cortex. This simple indicator describes how spontaneous fluctuations shift along the macroscale functional hierarchy across time. For the volume-based fMRI data (Dataset 1 and Datasets 3-6), the surface-based gradient in HCP fsLR32k was transformed into MNI volumetric space using the '-metric-to-volume-mapping' command in Connectome Workbench, using the ribbon constrained mapping algorithm option. The ribbon gradient map in MNI space was spatially equivalent to the image released in NeuroVault from Margulies et al.[32]. ($r > 0.99$, https://neurovault.org/images/24346/). For 3T HCP resting-state fMRI dataset, we used surface-based ICA-FIX denoised data based on CIFTI

grayordinate-based framework in the main analysis. We further examined whether the intra-subject temporal trajectory and the inter-individual difference of the hierarchical index were comparable between surface- and volume-based fMRI data. As illustrated in Supplementary Fig. 12, we found that the temporal trajectory of the surface- and volume-based hierarchical index was highly consistent across five randomly chosen subjects; likewise, the individual variation of the hierarchical index was observed to remain stable across 100 unrelated individuals.

## Complex brain state analysis

We hypothesized that the low-dimension flow of the hierarchical index is associated with distinct global brain states. In this part, resting-state fMRI data from the HCP dataset was analyzed while considering: (i) its high signal-to-noise ratio, temporal resolution, and large sample size; and (ii) that it can reveal rich dynamics in global conscious states, ranging from vivid waking states to natural light sleep[29]. Specifically, we divided the 1200 frames of each HCP run into 24 nonoverlapped 36 s windows across 100 unrelated individuals (up to 9600 fragments) to explore the tempo-spatial heterogeneity of global signal topology as well as brain complexity. For the investigation of the dynamic brain state, a window length between 30 and 60 s has been found to be reasonable and has been widely used in previous studies[97].

**Unsupervised learning of GS topology.** Here, we calculated the time-resolved GS topology in each nonoverlapped window. The global signal was defined as the average BOLD signal across all the cortical vertices. A GS topography map was calculated using the Pearson correlation between the global signal and time series for each grayordinate-based vertex across the cortex. Subsequently, the spatial correlation of the GS topology between each fragment was calculated as a 9600 × 9600 similarity matrix, and a data-driven K-means algorithm with Euclidean distance was applied. A two-cluster solution was applied, given its larger silhouette coefficient metric (from 2 to 10 clusters). Notably, the intention of the unsupervised learning was not to demonstrate that brain states are perfectly natural clusters but to decompose spatially heterogeneous patterns of GS topologies across time. To examine whether the clustering is primarily driven by individual difference, we computed the proportion of 96 sliding windows (24 nonoverlapped windows × 4 runs) of each subject that were assigned to state 1, leaving the remaining windows assigned to state 2. We found that no individual was entirely assigned to state 1 or state 2, with the proportion of state 1 occupied 52.3 ± 18.8% across the 100 unrelated subjects (Supplementary Fig. 8a). This preliminary evidence suggests that the k-means analysis was not mostly driven by differences between individuals. To further minimize the potential impact of individual differences, we repeated the subsequent analysis using a balanced subset of data: for each individual, the unmatched time windows would be removed randomly to ensure the proportion of state 1 and state 2 are equal. The results in Supplementary Fig. 8b, c were highly consistent with those shown in Fig. 4, despite the equal distribution of windows across the two states. Furthermore, we conducted a two-cluster k-means analysis of global signal topology at individual level, independently for 100 unrelated subjects, each with a sample size of 96 (24 nonoverlapping windows × 4 runs). To characterize the stability of the clustering results, we compared the individual-level clustering with the group-level clustering in terms of spatial similarity in comparison with 10,000 null models considering spatial autocorrelation. We found that the clustering based on the individual level was significantly correlated with the group-level spatial correlation (Supplementary Fig. 8d, state 1: averaged $r = 0.57$; state 2: averaged $r = 0.52$). Considering the limited sample size of the individual clustering, we randomly selected 5 individuals and repeated the same experiment 100 times, finding that the clustering results were highly correlated with the group-level analysis using 100 subjects

(Supplementary Fig. 8e, state 1: averaged $r = 0.88$; state 2: averaged $r = 0.81$).

To characterize functional integration/coordination, we also used a nodal strength approach to measure the pattern in which cortical regions are temporally connected. For each time window, we applied the 200 parcellation Schaefer atlas[98] and Pearson correlation to construct a cortical functional network ($200 \times 200$). Node strength is defined as the sum of weights of links connected to the node.

**Connectivity and temporal entropy.** As mentioned above, for each nonoverlapping window of an individual, we acquired a $200 \times 200$ function network defined by the Pearson's correlation between each pair of Schaefer ROIs. Connectivity entropy was then used to describe the distribution diversity of the functional connectivity. To quantify the amount of uncertainty in each brain state, we adopted Saenger et al.'s method[99] to transform the functional connectivity values for each node into discrete bins ($n = 10$) and calculated the normalized the Shannon entropy for each ROI's probability distribution. The connectivity entropy was calculated according to the following Eq. 3:

$$H = - \sum_{i=1}^{n} \frac{p_i \log(p_i)}{\log(n)} \qquad (3)$$

In addition, we used sample entropy to measure the regularity and complexity of the temporal fluctuation. Specifically, the time series was divided into chunks of m or m + 1 timepoints each. The chunks were then compared to find the Chebyshev distance between them. The parameter $r$ was set as the threshold to determine whether two chunks were similar. The sample entropy was defined as Eq.4:

$$H = - \log \frac{A}{B} \qquad (4)$$

in which A is the number of chunks of length m + 1 having a Chebyshev distance less than $r$ (B refers to the number of similar chunks of length m). We used $m = 2$ considering the length of time ($N$ less than 10 m), and $r = 0.5$ multiplied by the standard deviation of time series accounting for the amplitude variations of signals. Such parameters were previously used in resting-state fMRI studies[100,101].

## Global and physiological nuisance signals

The interpretation of the global signal is quite obscure, and its spatial contribution is structurally heterogeneous across the brain. The GS topography was shown to be aberrant in schizophrenia[47], bipolar disorder[102], and epilepsy[103], and can be significantly modulated by external stimuli[104]. Applying K-means clustering on time-resolved GS topography, we suggest that the reconfiguration of the GS topography reflects a large-scale, dynamic integration linking to vigilance states during rest. Thus, the GS regression strategy may reduce such dynamical topography across time, which probably distorts state-dependent analyses[105] but strengthens trait-like functional representations[106]. Additionally, we showed that the GS regression procedure has minimal effect on the hierarchical index (Supplementary Materials). Using HCP data that included simultaneous physiological recordings, we demonstrated that regressing out the respiratory and cardiac cycles did not influence our key results (Supplementary Materials). An additional analysis was also performed to evaluate the potential effects of head motion (Supplementary Materials). Critically, our results based on macaque electrocorticography data largely mitigated the concerns potentially introduced by motion artifacts and physiological nuisance data in human fMRI results.

## Recurring spatiotemporal pattern analysis

Spontaneous BOLD fluctuations manifest as one of a few recurring spatiotemporal patterns. Of these, one of the most investigated is a quasiperiodic pattern (QPP) described by Majeed et al.[107]. This QPP was identified by recursively matching and averaging similar segments of resting-state fMRI time series. The primary QPP involves a dynamic cycle of activation and deactivation that spatially follows the macroscale gradient (from the task positive network to the default mode network) and lasts approximately 20 s. For the human fMRI data, we directly applied a representative spatiotemporal template to match all possible QPP events. This template is publicly available from a recent study[26] that conducted an optimized, computationally expensive algorithm to detect QPP events based on HCP S900 subjects. Only cortical regions were included when generating this group template. Consistent with this previous work[26], we applied a sliding window approach to iteratively compare the correlation between the template (30 volumes, ~21.6 s) and each flattened 'segment' with a temporal step of 1 TR. To reduce computational complexity, we analyzed the ROI-based average time series to identify QPP events using the Schaefer 200 parcellation map. Spatiotemporal segments were identified as QPP events whose local maxima exceeded the threshold ($r = 0.4$). In this analysis, our purpose was not to demonstrate the cortex-wide propagation pattern or its role in arousal modulation, which have been revealed previously, but to study whether the previously reported QPP exhibits significant heterogeneous modes in different vigilance states. To achieve this, 100 HCP individuals with the most protruding drifts in hierarchical scores during 4 resting-state scans were included. For each of 400 scans (100 individuals × 4 runs), the first third and the last third of the time were categorized into two different vigilance states. In this way, we were able to perform the QPP analysis on 400 high-quality fMRI scans in two different global states while avoiding introducing individual difference. There are two major obstacles that prevented us from performing a human QPP analysis beyond the HCP data: (i) The QPP analysis is based on a recurring spatiotemporal pattern lasting approximately 20 s, which has been reliably identified in previous studies[26,27,107]. However, most of the human fMRI datasets we used have a repetition time of around 2 s, so only about 10 time points can be used for the calculation of spatiotemporal cycles. In contrast, the HCP dataset contains up to 30 time points (TR = 0.72 s), providing a more robust basis for analysis. (ii) In addition to the limited time points available in most human fMRI datasets, the HCP data have significantly more fMRI runs. This larger sample size is advantageous when attempting to compare QPP across different states.

We next conducted an exploratory QPP analysis for the filtered gamma-band (40–100 Hz) power from the macaque ECoG recordings. For each macaque, we established a representative spatiotemporal template during the awake, eyes-closed resting state, considering: (i) previous studies that suggested arousal shifts were linked to global fMRI signals in the eyes-closed condition[76], and (ii) that the template could be independently used to detect QPP events in other states (such as during sleeping and eyes-opened condition) for comparing QPP heterogeneity. We downsampled the temporal resolution into 0.4 s, and 50 timepoints (20 s) were then set as the segment duration. Based on the C-PAC package[108], a standard QPP algorithm was applied to reveal the recurrent spatiotemporal patterns. A threshold $r$ of 0.4 was used when building the template. Practically, the results were not sensitive to thresholds of r from 0.2–0.5. We found that the QPP templates could be reliably derived and were visually similar in the two monkeys. To match the QPP templates from the two monkeys, we applied a phase-adjusting procedure by shifting a few timepoints forward and backward. As shown in Fig. 6j and Supplementary Fig. 11, the QPP template resulted in the propagation of activity along the macroscale functional gradient. Across the 50 timepoints, the weighted signals along the principal functional embedding achieved a high temporal correlation ($r = 0.99$) between the two monkeys. The QPP template for each macaque was subsequently used to detect QPP events in other states. In each QPP event, we calculated the gamma BLP peak difference between the top and bottom 20 channels based on the cortical gradient to quantify the relative excitability of the high-order

cortex. The peak difference was calculated within the initial 12 s when the gamma BLP was most evident, and the delta signals in the subsequent 4 s were averaged across all channels.

## Imaging transcriptomic analysis

To examine which transcriptomic expression recapitulates the cortex-wide map of the functional hierarchy, we assessed the Allen Human Brain Atlas (AHBA)[28], a spatially comprehensive dataset of human transcriptional profiles. The gene expression data were acquired from post-mortem tissues in six donors without a history of psychiatric or neuropathological disorders. The donors were a 24-year-old African American male (H0351.2001), a 39-year-old African American male (H0351.2002), a 57-year-old European-ancestry male (H0351.1009), a 31-year-old European-ancestry male (H0351.1012), a 49-year-old Hispanic female (H0351.1015), and a 55-year-old European-ancestry male (H0351.1016). Further details are provided at http: //www.brain-map.org.

Instead of the Schaefer's parcellation, the Human Brainnetome Atlas[109] was used to spatially define the samples, because it was developed in volumetric space. Thus, it is more suitable for matching samples from the Allen Human Brain Atlas, whose anatomical location was provided in MNI space. We then aligned the gene expression data from the AHBA dataset into the 105 left cortical regions defined in the Human Brainnetome Atlas, using the standard workflow embedded in the abagen toolbox[110]. Specifically, the probe that exhibited the most consistent pattern of regional variation across donors was selected to index the gene expression. Using a scaled robust sigmoid normalization approach, expression values within each brain sample were normalized across genes and then normalized for each gene across samples for each donor. The anatomical information for the left cortex was used to match the hemisphere and tissue class designation provided by AHBA. Finally, the average normalized values of all samples encompassed within each parcel was defined in the left hemisphere Human Brainnetome Atlas, resulting in 105 expression values for each of 20,232 genes across the left cortex. The spatial association between the gene expression and the functional gradient across 105 brain regions was calculated by Spearman's rank correlation.

## A note on brain parcellations

In the section 'hierarchical cortex-wide fluctuations reflect ongoing states of consciousness', a network parcellation of Yeo's 17 networks was used to interpret the results, and the main analyses were performed in an atlas-free manner (e.g., in the sections 'characterizing hierarchical dynamics in single individuals' and 'hierarchical dynamics in psychedelic and psychotic brains'). In the section 'complex brain integration and differentiation', we computed temporal and connectivity entropy at the regional scale based on the Schaefer atlas and further compared the results with those from the Brainnetome atlas (Supplementary Fig. 15). In the section 'relationship to the infra-slow cortex-wide propagation phenomenon', we performed the QPP analysis using the time series derived from both the Schaefer parcellation (Fig. 6) and the Brainnetome atlas (Supplementary Fig. 16). The results were highly comparable. The analysis in the section 'hierarchical dynamics in macaque electrocorticography' was based entirely on the macaque electrodes. In the section 'implication of histaminergic system', we reported that three different pipelines all showed a highly spatially positive association between HDC gene expression and the main functional gradient: (i) using the Brainnetome atlas and the Abagen toolbox, ranked 3rd; (ii) using the surface pipeline of the Schaefer atlas provided by Anderson et al.[23], ranked 5th (Supplementary Table 5); (iii) the atlas-free pipeline on the Neurovault website, ranked 9th in terms of variance explained (Supplementary Table 6). We performed the analysis based on both the Schaefer parcellation (Supplementary Fig. 17) and the Brainnetome atlas (Fig. 7) for the analysis of hypothalamic functional connectivity.

## Reporting summary

Further information on research design is available in the Nature Portfolio Reporting Summary linked to this article.

## Data availability

The raw and processed 3T and 7T resting-state fMRI, and group-average dense functional connectivity data are available from the Human Connectome Project (https://www.humanconnectome.org/study/hcp-young-adult/document/1200-subjects-data-release). Simultaneous EEG–fMRI validation data during sleep, collected by Gu et al[89], can be downloaded at https://openneuro.org/datasets/ds003768/versions/1.0.9. The preprocessed *MyConnectome* Project data, originally analyzed here[37], is publicly available at https://openneuro.org/datasets/ds000031/versions/00001. The preprocessed fMRI data during placebo and LSD conditions, first reported by Carhart-Harris et al.[41], can be accessed at https://openneuro.org/datasets/ds003059. The resting-state fMRI of the Consortium for Neuropsychiatric Phenomics dataset is available at https://openneuro.org/datasets/ds000030/. The macaque ECoG data during anesthesia and sleep can be found at http://neurotycho.org/expdatalist/listview?task=78. The QPP template, generated in a previous work[26], can be downloaded at https://github.com/GT-EmoryMINDlab. The human gene expression was from the Allen Human Brain Atlas (https://human.brain-map.org/static/download) and analyzed with abagen toolbox (version 0.0.8, https://abagen.readthedocs.io/en/stable/). Other raw data are not publicly available due to data privacy laws and can be requested from the corresponding author. The data supporting the calculation of hierarchical index are available at Zenodo: https://doi.org/10.5281/zenodo.7855130 Source data are provided with this paper.

## Code availability

The BRANT toolbox (version 3.35) for fMRI data preprocessing is available at https://github.com/kbxu/brant. The documentation and code for generating spatially constrained null models are available at: https://github.com/netneurolab/markello_spatialnulls. QPP analysis was performed with code available here: https://github.com/FCP-INDI/C-PAC/blob/main/CPAC/qpp/qpp.py. The code used to compute the Shannon and Sample Entropy is available in the package NeuroKit2 (version 0.1.1, https://neuropsychology.github.io/NeuroKit/). The K-means clustering algorithm implemented in our study is based on the scikit-learn library (version 1.1.1, https://scikit-learn.org/stable/modules/clustering.html#k-means). Code to calculate hierarchical index at both volumetric and surface space in our work is available at Zenodo repository: https://doi.org/10.5281/zenodo.7855130.

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

## Acknowledgements

This work was supported by the Natural Science Foundation of China (82171543 to A.L., 82271284 to R.H. and 31971028 to X.L.), STI2030-Major Projects (2022ZD0211900 to A.L.), the Strategic Priority Research Pro-gram of the Chinese Academy of Sciences (XDB32020200 to A.L.), the Start-up Funding of Beijing Normal University (to B.L.), Basic Research Program Qinghai Province (2023-ZJ-753 to H.L.), Qinghai Plateau Renowned Doctors Program (to H.L.), Magnetic Resonance Union of Chinese Academy of Sciences (2021gz1003 to X.W.), the Collaborative Research Fund of the Chinese Institute for Brain Research, Beijing (2020-NKX-PT-02 to X.W.), and the New Cornerstone Science Foundation (to X.W.). We thank Prof. Yingchao Shi at Guangdong Institute of Intelli-gence Science and Technology for insightful discussions. We thank Drs. Rhoda E. and Edmund F. Perozzi for providing English and content editing assistance.

## Author contributions

B.L., and X.W., led the project. R.H., H.L., X.L., C.Z., and H.Y. contributed to the data collection. A.L., B.L. and X.W. designed the study and wrote the manuscript. A.L. analyzed the data. R.H., H.L., X.L., Y.H., Q.W., Y.Y., X.Z., X.T., and P.Y. interpreted the results. S.H., Y.H., and Q.W. provided critical comments. K.L., SZ.H., M.W., and Y.S. assisted in revising the manuscript. All authors contributed to editing and proofreading the manuscript.

## Competing interests

The authors declare no competing interests.
