## [Peer Review File · Nature Communications]

Hierarchical Fluctuation Shapes a Dynamic Flow Linked to States of ConsciousnessREVIEWER COMMENTS

Reviewer #1 (Remarks to the Author):

Li et al utilize different paradigms and multiple datasets (some of which are publicly available) collected across various species in order to provide a spatiotemporal characterization of the neural substrates underlying states of consciousness. The study tackles a very interesting scientific problem and the authors utilize sophisticated analytical methods and illustrate the results using nice figures. However, there are fundamental concerns and insufficient information within the presented results, which makes it difficult to evaluate the results and conclusions. The way the results are presented is complex and the manuscript is difficult to read. It is important that the authors underline and clearly explain how these results add to our understanding of the neural correlates of states of consciousness and contextualize their finding, taking into account the present literature on the subject.

Please find suggestions/comments below:

- The functional gradient presents only 25% of the variance (Page 33; Line 9). The manuscript would greatly benefit from a demonstration and analysis of the remaining variance/gradients.
- The results section also presents a T1w/T2w analysis. It would be helpful to clarify the motivation for this analysis in regards to the presented model of states of consciousness.
- It would help to clarify better, the motivation of the question under study, presented on Page 3, line number 8: What is the hidden link between the mental phenomena and the neural substrates of consciousness? While the next paragraph presents a motivation for looking at spatio-temporal pattern of activity in many different facets of brain function, the authors could perhaps motivate this better by elaborating why such a point of view is relevant for investigating the states of consciousness.
- For the non-specialist readers, it shall be helpful to explain the concept of the principle functional gradient in the main text, before such terminology is used.
- Figure 1d, it is hypothesized that there is a vigilance decrease during the resting state fMRI data acquisition. It would be more convincing, if appropriate evidence to support this claim could be presented. The authors cite a study (Tagliazucchi and Laufs, 2014), which reported that one-third of the subjects tested during resting state were unable to maintain wakefulness for over 3 minutes (Methods section: Page, 27, line 18). However, it would be important to find a suitable measure that supports the author's claim for the analyzed dataset. If the authors can identify the individuals exhibiting a reduction in alertness, they could focus their analysis on these subjects, which could help in strengthening their findings. Further, I suggest adding this information to the main text, where these results are presented.
- Page 5, Line 28, please clarify what are 'hierarchical disturbances'?
- Page 6, Line 1: While 'Multiple theories...' is stated, a single paper related to the integrated information theory has been cited. Please cite literature related to other theories, which are being referred to. Also, please clarify in the main text, how global signal topology is an appropriate method for characterizing and assessing global integration of information.
- Page 7, Line 28: please cite appropriate literature for this sentence.
- Page 32; Line 10-12:
Using comparative methods for different analysis shall be helpful. It is unclear why one analysis is done with the surface domain, while others are in the voxel space as HCP data is also available in the volume space.
Furthermore, please add information if the z-transformed bold amplitude correlation with the functional gradient was done utilizing the multiple comparison corrections (if applicable) (Figure 1).
- In the discussion section, authors mention the time scales ranging from several minutes to 'years', please clarify which experiments are being referred to in this regard.
- Page 19; Figure 6a-b:
The principle functional gradient across the brain of the two animals looks rather different. It would be essential to highlight, where are the differences.
- Page 9; Line 12-13:
Please provide appropriate citation implicating TMN as the wake-promoting center in the brain.

Please elaborate on the selection criteria for choosing the hypothalamic TMN region. For example, in Figure 7e, the correlation is similarly high with the anterior hypothalamic area. The preoptic area in the anterior hypothalamus has been implicated in sleep regulation (Rothhaas and Chung, 2021), and therefore also relevant to the study of states of consciousness. It would be helpful to discuss this in the context of the presented results.

- Page 14, Figure 1d: the correlation values are rather low (0.05 to - 0.05) for the resting-state vigilance decrease pattern. Please discuss.

- The yeo atlas (utilized in figure 1k-m, figure S3, etc) has a spatial overlap across the cortical brain regions. Therefore, the observed correlation encompasses topographic overlaps. Please discuss.

- With respect to the analysis related to gene expression, it would help to explain how it adds to the overall proposed model of consciousness. For example, how hierarchical heterogeneity is supported by the gene expression related to the histaminergic system.

Minor comments:

- Fig 2b: Human Connectome Project has been misspelled.

- Figure 5c, d: the word regions is misspelled.

- Figure 6 a, b: the word functional is misspelled.

- Figure 6 c, d, e and f: please add appropriate scale for the circular nodes displayed on the brain

- Page 4, Line 31: the word these is misspelled.

- Figure 7d: the words correlation and functional are misspelled

- Figure 7b: the word functional misspelled

Reviewer #2 (Remarks to the Author):

Thank you for giving me the opportunity to review this excellent manuscript by Li, Liu, Lei and colleagues. This is an impressive investigation across datasets, species, and methodologies. I am enthusiastic about this body of work, although I believe that some methodological aspects should be improved, primarily to show robustness. In general, my main comment is that the work adopts a lot of different methodologies, and while this is undoubtedly a strength, the methods themselves (terms, workflows) need to be more clearly explained in the text itself. Likewise, the Introduction and Discussion are rather dispersive, mentioning many theories and concepts without properly explaining them. I think this unnecessarily restricts the audience to consciousness specialists, whereas this paper should be made accessible to a much broader audience of neuroscientists. More methodological clarity and theoretical focus will achieve this, and give this paper the attention it deserves.

My detailed comments are provided below.

Abstract:

Line 11: "spatiotemporally propagation" \diamond spatiotemporally propagating.

Line 12 (and throughout the rest of the paper): "principle" should be "principal" (these words are often confused even by native speakers).

Line 15: "the" global state of consciousness.

An overall comment is that it would be preferable to have line numbers that are continuous, rather than restarting at each page

Introduction

Line 5: The consciousness entity: please rephrase.

Line 33: reference for the claim of drowsiness within minutes during resting-state?

In general, the Introduction mentions a lot of technical terms (criticality, scale-free dynamics, etc): it would be helpful for non-specialist readers to describe each new technical term a bit more.

Results

The Results are hard to understand without reading the Methods first (which is what I resorted to doing). I understand this order is the journal's policy, but I strongly suggest that in the Results the authors should introduce at least briefly each new analysis and its motivation before describing its outcomes, to orient the reader.

Page 4, Lines 20-21: this is very hard to follow: where are “millions of dense state pairings” coming from? How does this reflect vigilance?

Page 5, lines 22-23: the authors should distinguish between the psychedelic and psychotic state more carefully. Psychedelics can have “psychoto-mimetic” effects, but equating the two is an oversimplification that should be avoided, especially given the recent surge in psychedelic-assisted therapy aimed at treating mental disorders.

Page 5, line 25: higher hierarchical index seems in contrast with the results of Girn et al (2022, NeuroImage) on the same dataset, showing a “collapse” of the principal gradient. Could the authors explain this discrepancy?

Page 6, line 15: the authors should explain more clearly what “the graph theory approach” is, as the current explanation is very uninformative.

Page 7, lines 9-11: as above, this analysis is very unclear without having read the Methods.

Page 8, line 1: define BLP?

Page 9, lines 2-21: these results are impressive, but it does not seem that they show modulation by the histaminergic system, only correlation (albeit greater than other hypothalamic nuclei). The authors should relate these results to their previous observations pertaining to consciousness (for instance by considering the different levels of vigilance) if they want to make the kind of stronger claim that they presently do.

Discussion

The arousal-promoting system results are given great prominence, but as mentioned above, the claim of mediation is not substantiated by evidence: the authors only show a correlation in a separate dataset.

Page 10, line 8: “zombies” will only be familiar to readers with philosophy background: this terms should be clarified (or better, removed).

Line 15: reference for the dynamic core hypothesis?

Page 10, line 29: What is the evidence for modulation of neuronal excitability?

Page 11, line 30: what do the authors mean by “a conservative system”?

Page 11-12: the discussion of artificial neural networks, deep learning, and sleep is extremely speculative and seems to have very little to do with the actual manuscript’s results. Please remove.

The authors acknowledge that the QPP analysis is restricted to one dataset, but it is unclear why it cannot be also performed in the other datasets (especially the altered states such as sleep, anaesthesia and psychedelics), which would make the results much more robust.

Methods

Page 34: would the results of the k-means analysis also hold if performed separately for each individual? And relatedly: could the clusters be mostly driven by differences between individuals, rather than across time-points?

The authors should specify which analyses were performed with which parcellation: the Schaefer atlas is mentioned in some places, the Brainnetome in others: although the authors offer some justification for this, it would be very important to show that the results are not parcellation-specific.

The sliding windows used in this study are not consistent across analyses (30 frames, 50 frames); why not use either the same duration or the same number of frames?

Figure 3a: how is significance assessed at the individual-subject level? Three data-points per subject do not seem enough for this to be meaningful (and is this corrected for 15 comparisons)?

Figure 4a: the diagram is rather uninformative about how this was actually computed.

Figure 5c,d: “reions” should be “regions”; also define “saturability”?

Figure 7: it is now well-established that correlations between cortical maps can be spuriously inflated by spatial autocorrelation, and rigorous tests have been developed to control for this effect (e.g., Markello and Misic, 2021, NeuroImage). The authors should demonstrate that the correlation in Fig. 7b-c-e are statistically significant beyond the effects of spatial autocorrelation.

Also the authors do not explore the equally strongly anticorrelated genes at the other end of the distribution, which may hold equally interesting insights: why focus only on the positive correlations?

Figure 8 and Discussion: the focus on excitability seems misplaced, since this is not one of the aspects that the study's results actually support, as far as I can tell: why not focus on those instead?

Overall comment on the figures: I think the characterisation of the different states of consciousness with cartoons is truly excellent.

Reviewer #3 (Remarks to the Author):

The authors postulated low-frequency signal (BOLD) variability as a signature of consciousness across multiple brain states which include wakefulness, sleep, different forms of anesthesia, to be further evaluated in subjects under a psychedelic drug, psychotic patients, and others. The main result is an increase in variability which coincides with the spatial location of the first functional gradient, plus a series of analysis showing changes in integration consistent with previously studied metrics.

I think this is a fine paper from a methodological perspective, but unfortunately I also think that the questions addressed by it have been extensively studied in recent years, and thus that the results might lack novelty to be published in this journal. The scale-free properties of BOLD and EEG signals during sleep and drug-induced unconsciousness have been extensively investigated in the past (see Refs. at the end) showing changes in the slope of the power spectrum that are consistent with increased power of low frequency fluctuations. In turn, as correctly identified by the authors, this translates into enhanced signal variability. Moreover, these previous papers identified that these changes predominantly occurred in fronto-parietal DMN regions, which show a substantial overlap with the first functional gradient. Similar results hold to the other datasets studied by the authors (ECOG, LSD) as well as the analysis of integration-related metrics.

Also, in my opinion the interpretation of the results turns a bit awkward when maps are compared with the Allen Brain Atlas transcriptomic data and the authors attempt to adopt an interpretation based on the outcome (in this case, concerning the relevance of histamine). A discussion of the highly speculative use of these and other related methods for the exploratory analysis of fMRI results is beyond the scope of this review, but overall I think these analysis obstruct the main message of the manuscript and tend to be included since they became somewhat trendy.

Other limitation to be highlighted is the very limited number of subjects in the sleep dataset (n=8).

References:

Lei, X., Wang, Y., Yuan, H., & Chen, A. (2015). Brain scale-free properties in awake rest and NREM sleep: a simultaneous EEG/fMRI study. *Brain topography*, 28(2), 292-304.

Tagliazucchi, E., von Wegner, F., Morzelewski, A., Brodbeck, V., Jahnke, K., & Laufs, H. (2013). Breakdown of long-range temporal dependence in default mode and attention networks during deep sleep. *Proceedings of the National Academy of Sciences*, 110(38), 15419-15424.

Horowitz, S. G., Fukunaga, M., de Zwart, J. A., van Gelderen, P., Fulton, S. C., Balkin, T. J., & Duyn, J. H. (2008). Low frequency BOLD fluctuations during resting wakefulness and light sleep: A simultaneous EEG-fMRI study. *Human brain mapping*, 29(6), 671-682.

Zempel, J. M., Polite, D. G., Kelsey, M., Verner, R., Nolan, T. S., Babajani-Feremi, A., ... & Larson-Prior, L. J. (2012). Characterization of scale-free properties of human electrocorticography in awake and slow wave sleep states. *Frontiers in Neurology*, 3, 76.

He, B. J., Zempel, J. M., Snyder, A. Z., & Raichle, M. E. (2010). The temporal structures and functional significance of scale-free brain activity. *Neuron*, 66(3), 353-369.

REVIEWER COMMENTS

Reviewer #1 (Remarks to the Author):

Li et al utilize different paradigms and multiple datasets (some of which are publicly available) collected across various species in order to provide a spatiotemporal characterization of the neural substrates underlying states of consciousness. The study tackles a very interesting scientific problem and the authors utilize sophisticated analytical methods and illustrate the results using nice figures. However, there are fundamental concerns and insufficient information within the presented results, which makes it difficult to evaluate the results and conclusions. The way the results are presented is complex and the manuscript is difficult to read. It is important that the authors underline and clearly explain how these results add to our understanding of the neural correlates of states of consciousness and contextualize their finding, taking into account the present literature on the subject.

We are grateful for the reviewer's thoughtful comments and constructive suggestions. We have now carefully responded to the proposed suggestions and concerns as outlined below. For the reviewer's convenience, original comments are in *purple*, and our responses are in regular format. A list of the references mentioned throughout our response can be found at the end of this document.

To address the reviewer's important concern, we have now made substantial improvements to our revised Results section. Specifically, in the revised Results section 1, we updated the text to describe how group-level consciousness-related maps were calculated from HCP resting-state and sleep fMRI data, and added background and rationale for the use of functional gradient and T1w/T2w maps; in Section 2, we elucidated the rationale and methodology employed in computing the hierarchical index and provided a brief description of the *MyConnectome* dataset; in Section 3, we revised our rationale for examining participants under the influence of psychedelics and those with psychiatric disorders; in Section 4, we provided further details on using the global signal (GS) topology and node-wise weighted strength to characterize global integration/coordination; in Section 5, we revised the methodological details of the quasi-periodic pattern (QPP) analysis; in Section 6, we extended the methodology and motivation for using macaque electrocorticography data, also noting the difference of functional gradients between two macaques; in Section 7, we explained the rationale for the use of the Allen Human Brain Atlas, as well as the special emphasis on the tuberomammillary nucleus.

As suggested, we have substantially revised the Discussion to better contextualize our findings. In particular, we addressed that our work reveals a simple phenomenon: global states of consciousness are shifted along a hierarchical continuum of cortical neural variability, which is based on experiments using different paradigms, imaging modalities, and species. Adhering to this principle, the multidimensional spatiotemporal patterns of cortical activity can be mapped to a low-dimensional signature, allowing for individual-level characterization of different states of consciousness. We proposed that the global state of consciousness may not depend on a specific location in Euclidean space, but rather be associated with low-dimensional computational patterns in topological space. The observed topological signature may serve as a marker for the global availability of hierarchical top-down processing, whose fluctuation would fall within a certain range. Deviations from this range may associate with aberrant conscious processing, such as in psychedelia and in individuals with psychiatric disorders. The hierarchical signature aligns with global complex patterns of functional coordination and diversity underpinning vivid wakefulness. Furthermore, we suggested that the hierarchical signature observed is modulated by spatiotemporal waves of cortical activities, as well as likely involvement of the histaminergic arousal system. We discussed that how the hypothesis can be supported by multiple classical theories of consciousness. We have also added the relevant studies as references; more details can be found in our responses below and in the revised manuscript, with certain changes underlined. In an effort to make the manuscript more understandable, we also invited some colleagues in the field of neuroscience to read the manuscript and make adjustments based on their feedback.

Please find suggestions/comments below:

- The functional gradient presents only 25% of the variance (Page 33; Line 9). The manuscript would greatly benefit from a demonstration and analysis of the remaining variance/gradients.

We are grateful for the reviewer's suggestion. By our first exploratory analysis, we observed that the shifting of cortical neural variability across different states of consciousness (e.g., sleep, anesthesia, fatigue) appears to consistently present a continuous topological transition from lower-order brain regions to higher-order networks. We previously chosen the principal functional gradient to perform further spatial correlation analysis based on visual inspection and prior knowledge, since this gradient is able to account for the continuous change in the spatial distribution of large-scale networks from sensory/motor regions to the higher-order network, which is highly congruent with our primary observations and has been extensively studied in many prior works¹⁻⁵. To address the reviewer's suggestion, we have now conducted an additional analysis incorporating top 9 functional gradients in the revised version. As shown

in Fig. R1a, the 2nd gradient ranges from visual to somatosensory/motor cortex; the 3rd gradient ranges from default mode network to attention networks; 4th gradient ranges from visual to attention networks; 5th gradient ranges from visual, somatosensory/motor to auditory regions; the subsequent gradients become increasingly complex. The eigenvalue λ s exhibited a sharp drop after the 9th component (Fig. R1b). We then employed partial correlation to compare the spatial similarity between the top 9 functional gradients and the three consciousness-related patterns. As a result, the principal functional gradient was able to best explain the spatial patterning of neural variability relating to global states of consciousness (Fig. R1c-f), thus quantitatively confirming the plausibility of using the principal functional gradient.

Fig. R1 (added to the revision as Supplementary Fig. 5) Top nine functional gradients and spatial signature of cortex-wide BOLD amplitude relating to anesthetic effects, sleep, and vigilance. **a**, Visualization of 1st to 9th functional gradients of human functional connectome during resting-state. The principal gradient ranges from

unimodal to multimodal networks; the 2nd gradient ranges from visual to somatosensory/motor cortex; the 3rd gradient ranges from default mode network to attention networks; 4th gradient ranges from visual to attention networks; 5th gradient ranges from visual, somatosensory/motor to auditory regions; the subsequent gradients become increasingly complex. **b**, The top 300 eigenvalues from diffusion embedding of dense functional connectome. **c-f**, Partial correlations between nine cortical functional gradients and consciousness-related maps (top left, **c**: sedation pattern; bottom left, **e**: sleep pattern; right, **f**: vigilance decrease pattern). Higher association value corresponds to higher saturation of red.

Revisions to the manuscript:

- Main text from line 143: “*specificity was strengthened by controlling for other functional gradients and within individual Yeo's networks (Supplementary Fig. 3-5)*”.
- Methods from line 971: “*In our work, we derived the top nine eigenvectors: the first eigenvector, which explains the most variance, was utilized for most of the analyses, while the 2nd to 9th eigenvectors were used for the control analysis in relation to three consciousness-related maps (Supplementary Fig. 5, the eigenvalues exhibited a sharp drop after the 9th component).*”
- Added new figure as Supplementary Fig. 5.

- The results section also presents a T1w/T2w analysis. It would be helpful to clarify the motivation for this analysis in regards to the presented model of states of consciousness.

We appreciate the reviewer's useful suggestion. Beyond using the principal functional gradient as a proxy of cortical hierarchy, here we aimed to evaluate whether the same conclusion holds when using the MRI-derived T1w/T2w mapping, which provides a non-invasive correlate of the anatomical hierarchy⁶. Therefore, the relationship between the consciousness-related signature and the cortical hierarchical organization could be reliably established from either functional or structural aspects, supporting the proposed model of states of consciousness. To address the reviewer's remark, we have now clarified the motivation for this analysis: “*In addition, T1w/T2w mapping has been shown to provide a non-invasive correlate of the anatomical hierarchy based on the structural profiles. To quantitatively characterize our observations, we performed a cortex-wide spatial comparison between the consciousness-related patterns and the main functional gradient derived from the dense functional connectome data (Fig. 1c, f, i), as well as T1w/T2w mapping (Supplementary Fig. 3)*”.

- It would help to clarify better, the motivation of the question under study, presented on Page 3, line number 8: What is the hidden link between the mental phenomena and the neural substrates of consciousness? While the next paragraph presents a motivation for looking at spatio-temporal pattern of

activity in many different facets of brain function, the authors could perhaps motivate this better by elaborating why such a point of view is relevant for investigating the states of consciousness.

The reviewer's comments highlighted the need for us to more clearly articulate the motivation behind our work and how it contributes to our understanding of states of consciousness. We apologize for the lack of clarity in our previous writing. Taking this opportunity, we have revised the second and third paragraphs of the Introduction to provide a background of existing research and evidence to support our hypothesis: that, in contrast to isolated focus on neural variability (temporal flexibility) or anatomical region (spatial location), the integration of the two components, i.e. topographically organized neural variability patterns, may be a potential determinant of global states of consciousness.

In the second paragraph of the revised Introduction, we illuminated the significance of investigating the spatiotemporal dynamics of spontaneous activity from various perspectives, which motivates our utilization of task-free fMRI or electrocorticography data for studying consciousness. In response to the reviewer's inquiry, we previously sought to elucidate that by studying spatiotemporal dynamics, a “hidden link” may be revealed that could bridge neural features to the mental phenomena of consciousness. In the third paragraph, we explained that temporal variability is a key characteristic of spontaneous brain activity. This variability is thought to reflect the flexibility of neural systems, thus to support a broad repertoire of brain states, and have a specific spatial pattern across the cortex.

Revisions to the manuscript:

- Modified 2-3 paragraph of Introduction: *Revealing the complex but orchestrated brain organization underlying these global brain states (i.e., different levels of consciousness) is essential to understanding the neural mechanisms of consciousness⁷. The spatiotemporal organization properties of spontaneous brain activity, which is considered to constantly provide top-down predictive models for future interactions during perception^{8,9}, may offer essential insights into consciousness across different conditions (e.g., during anesthesia and sleep) and species. Resting-state fMRI (rsfMRI), as the widely used technique in humans to map the spatiotemporal patterns of spontaneous brain signals, has detected highly reproducible intrinsic functional brain networks¹⁰, which largely reflect anatomical organization, individual-specific¹¹ and task-evoked¹² information. Breakdown in consciousness are accompanied by intricate changes in various aspects of the intrinsic functional organization, such as long-distance interactions^{13–15}, anti-correlated structures¹⁶, and patterns of brain coordination^{17,18}. Therefore,*

uncovering the temporal and spatial aspects of spontaneous brain activity in different global brain states are crucial for understanding the unified brain functional organization of consciousness levels.

Temporal variability/flexibility, which quantifies the dynamic range of ongoing brain activities, is increasingly being recognized as a beneficial factor for the adaptability of neural systems¹⁹⁻²¹. For instance, greater variability of BOLD signals has been reliably observed in younger healthy individuals²² and associated with more efficient performance in cognitive tasks^{19,20,23}. According to computational models²⁴, the variability of spontaneous activity arises from the dynamic system's noise-induced transitions between multistable attractors²⁵ or fluctuation around a critical line proximal to instability²⁶. Generally, this variability reflects and supports the exploration of a broader brain dynamical repertoire. Conversely, deterministic task stimuli limits the flexibility and quenches signal variability^{27,28}. In the spatial domain, resting-state BOLD variability is not randomly distributed but recapitulates the relative expression of cell markers for input-modulating somatostatin and output-modulating parvalbumin interneurons²⁹, which plays an important role in mediating cortico-cortical communication³⁰. Taking these into account, we consider that neural variability organized in an intrinsic spatial arrangement, i.e., the integration of space and time, would be optimal for conscious processing and be sensitive to changes of brain states. Consequently, we hypothesize that a topographically organized neural variability pattern may orchestrate the rise and fall in global states of consciousness. Importantly, this hypothesis prompted us to focus on the search for a temporal-spatial nested signature rather than a specific neuroanatomical location as a determinant of consciousness.

- For the non-specialist readers, it shall be helpful to explain the concept of the principle functional gradient in the main text, before such terminology is used.

Following the reviewer's suggestion, we have now added the following clarification (from line 137) before introducing the principal functional gradient “*Previous studies^{1,31} have well-documented the existence of a principal functional gradient in the human brain. This gradient explains the greatest variance in the functional connectome and captures the cortical processing hierarchy, spanning from primary sensory to trans-modal areas.*”

- Figure 1d, it is hypothesized that there is a vigilance decrease during the resting state fMRI data acquisition. It would be more convincing, if appropriate evidence to support this claim could be

presented. The authors cite a study (Tagliazucchi and Laufs, 2014), which reported that one-third of the subjects tested during resting state were unable to maintain wakefulness for over 3 minutes (Methods section: Page, 27, line 18). However, it would be important to find a suitable measure that supports the author's claim for the analyzed dataset. If the authors can identify the individuals exhibiting a reduction in alertness, they could focus their analysis on these subjects, which could help in strengthening their findings. Further, I suggest adding this information to the main text, where these results are presented.

We thank the reviewer for raising this concern. This part of analysis is to establish a group averaged map for the decreasing vigilance. To achieve this, we divided every resting-state run from HCP data into non-overlapping time windows and considered that, at the population-level, the time windows closer to the end of scanning were predisposed to have lower alertness in comparison with earlier time windows. The decreasing vigilance during resting-state scanning has been demonstrated by many previous studies^{10,32-35}. For instance, Tagliazucchi et al reported that wake possibility is reliably inversely proportional to the scanning time from multisite fMRI data³⁴ (in their Figure 4a, see below). In our case, for each cortical grayordinate, the Spearman's rank r was calculated between the normalized low-frequency BOLD amplitude and the time interval across 1,084,128 (276×4 runs \times 982 individuals) pairs of states/time windows (illustrated in Fig. R2). Considering the large sample size, the calculated map could be stable and representative for overall trend of the population-level decreasing vigilance.

The reviewer suggested that we can utilize a subset of individuals who exhibiting an alertness reduction to strengthen our findings. In the latter part of our study, we developed the hierarchical index as a signature and found that it can capture fluctuations of wakefulness/vigilance during resting-state fMRI scanning. Across resting-state sessions in the *Myconnectome* dataset, the hierarchical index was able to explain daily fluctuations in self-reported fatigue and heightened attention (Figure e-f in the revised manuscript). Additionally, our analysis of temporal trends showed that, at the group level, within-individual hierarchical indices decreased over time. This supports our hypothesis of a population-level decrease in vigilance during resting-state scans. In response to the reviewer's remark, we identified 100 subjects who exhibited the greatest degree of reduction in alertness (through time trend analysis), and then repeated the analysis for this subset of individuals. As shown in Fig. R3 we found that the vigilance map was highly comparable to those obtained using all individuals ($r = 0.97$). However, such subset analysis was relatively empirical in threshold selection and may reduce the sample independence in the subsequent analysis. Therefore, here we kept previous results in the main text and added this result as a control analysis in the supplementary materials. We have also included a discussion of the effect size in our response below.

Figure 4a from Tagliazucchi et al. Probabilities of steady wakefulness and of finding an awake subject in the Frankfurt data set.

Fig. R2 (added to revised version as Supplementary Fig. 2). A schematic diagram illustrating the method used to generate the decrease in vigilance pattern from resting-state fMRI data. Four resting-state runs were performed for each individual, with each run consisting of 1200 frames divided into 24 non-overlapping windows. Within each run, every two windows were paired as a sample, with the greater the temporal distance between the two windows, the more likely a decrease in vigilance level was assumed; i.e., for each cortical vertex, the difference in temporal order and the difference in normalized BOLD amplitude were correlated across all pairs of windows and runs.

Fig. R3 (added to the revision as Supplementary Fig. 14). Vigilance decrease pattern based on 100 representative subjects (greatest degree of reduction in hierarchical index from time trend analysis). **a**, Cortex-wide unthresholded correlation map between time intervals and z-normalized BOLD amplitude; a negative correlation indicates that the amplitude became more larger along with scanning time and vice versa. **b**, Vigilance-related cortical maps are highly similar across the spatial distribution using all or 100 selected subjects. ($r = 0.97$, $P < .0001$, Spearman rank correlation).

The revisions to the manuscript:

- Main text line 122: *“Previous research has demonstrated that individuals would generally exhibit a decrease in alertness over the course of the scan at the population level^{32,34}. Therefore, we divided each resting-state run from HCP data into non-overlapping time windows (50 volumes, 24 windows), and two different time windows were paired to form 276 ($24 \times (24-1) / 2$) different combinations, with the time interval ranging from 1 to 23 intervals. The larger interval of two windows was inferred to have a higher possibility of a decrease in vigilance. To measure the group-level pattern of alertness dropping, Spearman's rank r was calculated between the normalized low-frequency BOLD variability and the time interval across 1,084,128 (276 combinations \times 4 runs \times 982 individuals) pairs of states/time windows (Fig. 1d-e, Methods, Supplementary Fig. 2).”*
- Methods line 937: *“We also identified 100 most representative subjects (exhibiting the greatest reduction of hierarchical index across four resting-state runs) to generate a new vigilance map (Supplementary Fig. 14), and the pattern was highly comparable to those obtained using all individuals (shown in Fig. 1e).”*
- Added a new Figure as Supplementary Fig. 14.

- Page 5, Line 28, please clarify what are 'hierarchical disturbances'?

We thank the reviewer for pointing this out. To avoid the misleading, we have now rephrased this ambiguous term from ‘hierarchical disturbances’ to ‘abnormally higher hierarchical index ’ (see line 187).

- Page 6, Line 1: While ‘Multiple theories...’ is stated, a single paper related to the integrated information theory has been cited. Please cite literature related to other theories, which are being referred to. Also, please clarify in the main text, how global signal topology is an appropriate method for characterizing and assessing global integration of information.

We are grateful to the reviewer for pointing out our oversight in citing the literature, and in the revised manuscript, we have supplemented the following references:

Tononi, G. & Edelman, G. M. Consciousness and complexity. *Science* vol. 282 (1998).

Dehaene, S., Changeux, J. P. & Naccache, L. The global neuronal workspace model of conscious access: from neuronal architectures to clinical applications. *Res. Perspect. Neurosci.* 18, 55–84 (2011).

Dehaene, S. & Changeux, J.-P. Experimental and theoretical approaches to conscious processing. *Neuron* 70, 200–227 (2011).

We agree with the reviewer's suggestion that we should provide a clearer explanation of how global signal topology can be used to assess global integration of information. In the revised version, we have now modified the Fig. R3 (Fig. 4a in the revised version) to more precisely depict how we calculated the time-resolved global signal topology. Briefly, we computed the averaged BOLD signals across the cortex and measured how this global signal coordinates cortical regional activity through Pearson correlation. A high degree of synchrony was considered to reflect its capacity for global integration. To address the reviewer’s remark, we have now added the following sentence (from line 197): *“Therefore, we employed a modified version of global signal (GS) topology to specifically quantify the spatial inhomogeneity of cortex-wide integration/coordination (Methods). Specifically, we computed the mean cortical signals to capture the most prominent dynamics across the cortex, with peaks of high amplitude indicating a high degree of spatial homogeneity at that moment³⁶. As depicted in Fig. 4a, the GS topology characterizes the degree of integration with global cortical dynamics for a given cortical region.”* To further elucidate cortical integration, an alternative approach was also included based on cortex-wide functional network, and the revised description reads as follows: *“Control analyses suggest that the clustering and result were*

primarily driven by changes in brain state rather than individual differences and can be replicated in independent subjects (Supplementary Fig. 7-8). As an alternative, a weighted strength approach was applied as a proxy for functional integration / coordination. The node-wise weighted strength was developed based on the graph-theoretical concept of degree, which aims to quantify the global functional connectivity to other regions. Following the same clustering pipeline, we found that the two approaches yielded a similar clustering solution (Methods, Supplementary Fig. 9).”

Fig. R4 (Fig. 4a in the revised manuscript). Simplified diagram for dynamic GS topology analysis.

- Page 7, Line 28: please cite appropriate literature for this sentence.

For the sentence ‘*BOLD signal fluctuation reflects localized changes in neural activity indirectly through a complex neurovascular coupling*’, the following two references were cited:

Hillman, E. M. C. Coupling mechanism and significance of the BOLD signal: A status report. Annual Review of Neuroscience vol. 37 (2014).

Mateo, C., Knutsen, P. M., Tsai, P. S., Shih, A. Y. & Kleinfeld, D. Entrainment of Arteriole Vasomotor Fluctuations by Neural Activity Is a Basis of Blood-Oxygenation-Level-Dependent “Resting-State” Connectivity. Neuron **96**, (2017).

- Page 32; Line 10-12: Using comparative methods for different analysis shall be helpful. It is unclear

why one analysis is done with the surface domain, while others are in the voxel space as HCP data is also available in the volume space. Furthermore, please add information if the z-transformed bold amplitude correlation with the functional gradient was done utilizing the multiple comparison corrections (if applicable) (Figure 1).

We are grateful to the reviewer for bringing these points to our attention. To ensure the robustness of the results, it is essential to ensure that the results obtained from the surface domain and volume space are comparable. As mentioned by the reviewer, we initially utilized HCP surface-based fMRI data in our main analysis, given that the initial release of HCP S1200 only included ICA-FIX denoised data of grayordinates timeseries (the HCP-recommended pipeline for general analysis of resting-state fMRI). To address the reviewer's important comment, in the revised version, we have now downloaded additional ICA-FIX denoised resting-state fMRI volumetric data of 100 unrelated subjects from Extended HCP rfMRI dataset (detailed information can be found here:

https://www.humanconnectome.org/storage/app/media/documentation/s1200/HCP_S1200_Release_Reference_Manual.pdf; page 99). Utilizing the new data, we investigated whether the intra-subject temporal trajectory and the inter-individual difference of the hierarchical index were comparable between surface- and volume-based fMRI data. As illustrated in Fig. R5 (incorporated into the revision as Supplementary Fig. 12), we found that the temporal trajectory of the surface- and volume-based hierarchical index was highly consistent across five randomly chosen subjects; likewise, the individual variation of the hierarchical index was observed to remain stable across 100 unrelated individuals.

Fig. R5 (added to the revised version as Supplementary Fig. 12). a, Intra-subject correlation: the dynamic trajectory of the hierarchical index based on surface- and volume-based ICA-FIX-denoised resting-state fMRI data over four 15-min resting state runs in five subjects (each point represents 150 TRs, up to 32 non-overlapping windows). b, Inter-subject correlation: the inter-individual difference in the hierarchical index based on surface- and

volume-based ICA-FIX-denoised resting-state fMRI data from 100 unrelated subjects. The hierarchical index was averaged across four runs for each subject.

The reviewer also inquired whether multiple comparison correction was applied to the spatial correlation analysis between the z-transformed bold amplitude correlation and functional gradient. In the revised version, we have now incorporated a multiple comparison correction for this part. In response to the reviewer's comments, we have also taken into consideration the suggestion of another reviewer, which cited a recent work by Markello et al³⁷. This work provides a comprehensive evaluation of the effectiveness of 10 null models in mitigating the influence of spatial autocorrelation when conducting statistical analyses of neuroimaging data. While the performance of the null models performed similarly, they recommend the Alexander-Bloch framework³⁸ as an optimal option for vertex-wise data (<https://markello-spatialnulls.netlify.app/recommendations.html>). As shown in Fig. R6, this approach can generate spatially constrained null distributions by applying random rotations for spherical projection of the brain. Therefore, following the Alexander-Bloch method, we projected the vertex-wise principal functional gradient data onto the 32k_fs_LR sphere space and generated 10,000 null maps for subsequent permutation tests.

Fig. R6 (added to the revised version as Supplementary Fig. 4) The generation of 10,000 null models accounting for the effects of spatial autocorrelation. Vertex-wise surface-based spatially permutation models on the Alexander-Bloch method³⁸.

Revisions to the manuscript:

- Main text from line 142: *“The statistical significance was assessed using a spatially permutation-based null model”*.
- Methods from line 731: *“To ensure that the results obtained from the surface domain and volume space were comparable, we also included additional ICA-FIX-denoised resting-state fMRI volumetric data of 100 unrelated subjects from the Extended HCP rfMRI dataset (Supplementary Fig. 12).”* and from line 1000: *“For 3T HCP resting-state fMRI dataset, we used surface-based*

ICA-FIX denoised data based on CIFTI grayordinate-based framework in the main analysis. We further examined whether the intra-subject temporal trajectory and the inter-individual difference of the hierarchical index were comparable between surface- and volume-based fMRI data. As illustrated in Supplementary Fig. 12, we found that the temporal trajectory of the surface- and volume-based hierarchical index was highly consistent across five randomly chosen subjects; likewise, the individual variation of the hierarchical index was observed to remain stable across 100 unrelated individuals.”

- *Methods from line 949: “To test the statistical significance of the spatial correlations, we controlled the effects of spatial autocorrelation using generated null models (a schematic diagram was presented at Supplementary Fig. 4). For analyses in Fig. 1c-f-i as well as Supplementary Fig. 3, we projected the vertex-wise principal functional gradient data onto the 32k_fs_LR sphere space and generated 10,000 spatially rotated null maps for calculating P value in permutation tests. Specifically, the Alexander-Bloch approach³⁸ was applied to generate spatially constrained null distributions by applying random rotations for spherical projection of the brain. This is a recommend method based on a recent study which evaluating of the effectiveness of 10 null models in mitigating the effects of spatial autocorrelation. For Fig. 7c and 7e, since sampling was not conducted directly in the surface space, we employed the Vázquez-Rodríguez method³⁹, an adaptation of the Alexander-Bloch approach for parcellated brain data, to generate 10,000 null models and present permutation-based P values”*
- *Added new figures as Supplementary Fig. 4, 12.*

- In the discussion section, authors mention the time scales ranging from several minutes to ‘years’, please clarify which experiments are being referred to in this regard.

We thank the reviewer for raising this question. We have now clarified this point in the revised version of our manuscript. In particular, the revised sentence at line 335 reads as follows: *integrating different ... timescales (changed over the course of several minutes or more than one year, i.e., the MyConnectome Project).*

- Page 19; Figure 6a-b: The principle functional gradient across the brain of the two animals looks rather different. It would be essential to highlight, where are the differences.

We thank the reviewer for the constructive suggestion to focus on the distinctions in the principal functional gradient between the two macaques. Here, we utilized EcoG data from the Neurotycho project⁴⁰, and although the 3D spatial arrangement of implanted electrodes is not available, we can roughly discern the corresponding anatomical location of the EcoG array through a 2D visualization map. For the reviewer's convenience, we have now depicted the T1w/Tw2 cortical maps and principal functional gradient of humans and macaques in Fig. R7 (added to the revision as Supplementary Fig 10). It is evident that the principal functional gradient is homologous between humans and macaques, spatially manifesting from unimodal to transmodal cortex, mirroring the T1w/Tw2 cortical map. Nevertheless, as the reviewer highlighted, the principal functional gradient of the two macaques appears to possess certain discrepancies.

We contend that such difference is mainly attributable to the disparity in the sampling distribution of the electrodes of the two macaques. In the context of the dimension reduction algorithm, the principal functional gradient mainly portrays the continuous variation pattern of functional connectivity across two axis extremes, i.e. the unimodal-transmodal axis across the cortex. However, unlike in human fMRI acquisition, the implanted electrodes in the two macaques were sparsely sampled, and there was a considerable disparity between them. As illustrated in Fig. R7, Macaque 1 and Macaque 2 have a higher proportion of electrodes distributed in the medial motor and visual cortices, respectively, which may have augmented the internal functional connection ratios of the motor and visual networks to some extent, thus resulting in certain discrepancies in the gradient pattern in the motor and visual cortices. Nevertheless, it is essential to note that both the two cortical maps are able to reflect the large-scale unimodal-transmodal organization, and are highly correlated with the neural variability of gamma-BLP in different conscious states (Fig. 6c-f in the revised version).

Fig. R7 (added to the revision as Supplementary Fig. 10). Human and macaque cortical hierarchical organization. **a**, Top: principal gradient of the functional connectome based on human resting-state fMRI data. Bottom: human cortical T1w/T2w contrast. **b**, Top: principal gradient of the functional connectome based on resting-state macaque ECoG data; two macaques. Bottom: macaque cortical T1w/T2w contrast. The T1w/T2w ratio maps shown here were from a previous study⁴¹.

Revisions to the manuscript:

- Main text from line 272: “Despite the sparse sampling of implanted electrodes in the two macaques and their considerable disparity (Methods, Supplementary Fig. 10), the calculated principal gradients in each macaque revealed a clear unimodal-transmodal hierarchy across the neocortex (Fig. 6a-b), which is consistent with a recent study⁴².”
- Methods from 984: “*However, unlike in human fMRI acquisition, the implanted electrodes in the two macaques were sparsely sampled, and there was a considerable disparity between them. As illustrated in Supplementary Fig. 10, macaque Chibi and macaque George have a higher proportion of electrodes distributed in the medial motor and visual cortices, respectively, which may have augmented the internal functional connection ratios of the motor and visual networks to some extent, thus resulting in certain discrepancies in the gradient pattern in the motor and visual cortices. Nevertheless, it is essential to note that both the two cortical maps can reflect the large-scale unimodal-transmodal organization, and thus are highly correlated with the neural variability of gamma-BLP in different conscious states.*”
- Added a new figure as Supplementary Fig. 10.

- Page 9; Line 12-13: Please provide appropriate citation implicating TMN as the wake-promoting center in the brain. Please elaborate on the selection criteria for choosing the hypothalamic TMN region. For example, in Figure 7e, the correlation is similarly high with the anterior hypothalamic area. The preoptic area in the anterior hypothalamus has been implicated in sleep regulation (Rothhaas and Chung, 2021), and therefore also relevant to the study of states of consciousness. It would be helpful to discuss this in the context of the presented results.

To address the reviewer's suggestion, in the revised version, we have now cited two literatures supporting TMN as a wake-promoting hub:

Yoshikawa, T., Nakamura, T. & Yanai, K. Histaminergic neurons in the tuberomammillary nucleus as a control centre for wakefulness. *British Journal of Pharmacology* vol. 178 (2021).

Thakkar, M. M. Histamine in the regulation of wakefulness. *Sleep Medicine Reviews* vol. 15 (2011).

Given that hypothalamic TMN region is the major source of histaminergic neurons and thus critical for promoting wakefulness⁴³⁻⁴⁵, we feel that the sentence alluded to by the reviewer '*TMN as the wake-promoting center in the brain*' was logically unclear. In the revised version, we have therefore modified the text (see line 318) making this claim and rephrased the full sentence as "*It is evident that histaminergic system help sustain wakefulness. The tuberomammillary nucleus (TMN) in the posterior hypothalamus is the major source of brain histamine and widely project histaminergic neurons to the cerebral cortex*^{43,44}".

The reviewer also questioned the selection criteria for choosing the TMN region in the hypothalamus. The hypothalamic connectome mapping based on high-resolution 7T fMRI data is a subsequent analysis which follows the transcriptome result showing the hierarchical expression patterns of HDC (histidine decarboxylase, which is the unique enzyme catalysing the decarboxylation of histidine to form histamine) gene as well as HRH1 (Histamine Receptor H1, administration of histamine or H1 receptor agonists can induce wakefulness) gene. To further support the hierarchical heterogeneity of histaminergic system, we tested whether functional connectivity of hypothalamic TMN would exhibit a similar preference with high-order regions across the cerebral cortex. As a result, we found that both functional connectome mapping of TMN and gene expression of histaminergic profiles recapitulated the cortex-wide unimodal-transmodal gradient. We agree with the reviewer that selection criteria of TMN should be elaborated, and additional discussion should be provided for other hypothalamic regions, such as the preoptic area. To address the reviewer's remark, we have now added the following the sentence (plus the references, from line 324): "*As*

a result, we found that the TMN exhibited the most prominent spatial association with the principal functional gradient (Fig. 7e) across 13 hypothalamic regions, followed by the preoptic-anterior hypothalamus and dorsomedial hypothalamus, both of which play a role in sleep-wakefulness regulation^{46,47}”

- Page 14, Figure 1d: the correlation values are rather low (0.05 to – 0.05) for the resting-state vigilance decrease pattern. Please discuss.

As outlined in our response above, this analysis is based on the group-level trend of decreasing wakefulness during resting-state scanning. Despite the relatively low effect size, the pattern is stable (see above, Fig. R3) and effective in representing a group-averaged map for decreased vigilance. However, we agree with the reviewer that the effect size merits further discussion. In the revised version, we have now included the following limitation (see line 418): *“It is worth noting that the effect size of the vigilance map is relatively low. This could potentially be increased by providing additional information, such as using self-reported questionnaires to assess whether participants were prone to becoming sleepy while scanning.”*

- The yeo atlas (utilized in figure 1k-m, figure S3, etc) has a spatial overlap across the cortical brain regions. Therefore, the observed correlation encompasses topographic overlaps. Please discuss.

We are grateful to the reviewer for bringing this to our attention. We have checked the parcellation data of the Yeo atlas⁴⁸ and can confirm that it consists of 17 spatially non-overlapping networks across the cortical regions. This is because the parcellation was developed using a clustering algorithm rather than techniques such as independent component analysis. For the reviewer's convenience, the atlas was visualized as follows:

Fig. R8 Surface visualization of Yeo's 17 canonical functional networks.

- With respect to the analysis related to gene expression, it would help to explain how it adds to the overall proposed model of consciousness. For example, how hierarchical heterogeneity is supported by the gene expression related to the histaminergic system.

We appreciate the reviewer's insight and agree that it would be helpful to discuss the significance and relevance of gene expression analysis to overall model of consciousness.

After establishing a gradient of neural variability as a signature for global states of consciousness, we propose that further investigation into its molecular mechanism could enhance our understanding of the pattern. Previous animal studies have suggested that the ascending reticular activating system (ARAS) plays a crucial role in supporting consciousness^{49,50}, leading us to hypothesize that ARAS neurotransmitter systems, such as the histaminergic, cholinergic, and noradrenergic systems, may be involved in hierarchical heterogeneity across the cortex. Due to the challenges associated with collecting human tissue samples across different states, we utilized an in-vivo "imaging transcriptomics" technique to examine the molecular correlates of the neuroimaging phenotype, linking spatial variations across the transcriptomic and imaging scales. While this approach is indirect, it has the potential to yield plausible biological and transcriptomic mechanism and has been widely used in various studies^{29,51-55}.

Interestingly, as addressed above, we have found that both gene expression of histaminergic system and functional connectome of hypothalamic TMN showed a prominent association with the principal functional gradient. Extending previous evidence showing TMN histaminergic neurons widely project to the cerebral cortex^{43,44} and promote wakefulness, our result suggests that histaminergic system is

nonhomogeneous and associated with cortical hierarchical organization, thus serving a plausible biological underpinning for our macroscopic fMRI observations.

Revisions to the manuscript:

- Modified main text from line 304: *“Previous animal studies have suggested that the ascending reticular activating system (ARAS) plays a crucial role in supporting consciousness^{49,50}. Therefore, we hypothesize that specific ARAS neurotransmitter circuits (such as the histaminergic, cholinergic, and noradrenergic systems) may be preferentially involved in hierarchical heterogeneity across the cortex. To test the hypothesis, we performed a cross-modal analysis to search for genetic transcriptomes that were unimodally-transmodally distributed based on the Allen Human Brain Atlas (Methods). While this approach cannot reveal a causal link, it has the potential to yield plausible biological mechanism for macroscale imaging markers widely utilized by recent studies^{51,52,55,56}. ”*

Minor comments:

- *Fig 2b: Human Connectome Project has been misspelled.*
- *Figure 5c, d: the word regions is misspelled.*
- *Figure 6 a, b: the word functional is misspelled.*
- *Figure 6 c, d, e and f: please add appropriate scale for the circular nodes displayed on the brain*
- *Page 4, Line 31: the word these is misspelled.*
- *Figure 7d: the words correlation and functional are misspelled*
- *Figure 7b: the word functional misspelled*

We have corrected these spelling errors.

Reviewer #2 (Remarks to the Author):

Thank you for giving me the opportunity to review this excellent manuscript by Li, Liu, Lei and colleagues. This is an impressive investigation across datasets, species, and methodologies. I am enthusiastic about this body of work, although I believe that some methodological aspects should be improved, primarily to show robustness. In general, my main comment is that the work adopts a lot of different methodologies, and while this is undoubtedly a strength, the methods themselves (terms, workflows) need to be more clearly explained in the text itself. Likewise, the Introduction and Discussion are rather dispersive, mentioning many theories and concepts without properly explaining them. I think this unnecessarily restricts the audience to consciousness specialists, whereas this paper should be made accessible to a much broader audience of neuroscientists.

We are appreciative of the reviewers' positive feedback and enthusiasm for our work. As the reviewer addressed, our work endeavors to present a comprehensive exploration across datasets, species, and methodologies to decode the neural mechanisms underlying global states of consciousness. Also, we fully agree with the reviewer's concerns and suggestions regarding the methodology, writing, and readability of our manuscript. These comments have substantially aid us in refining our work, and some of the key points are outlined as below. For the reviewer's convenience, original comments are in *purple*, and our responses are in regular format. A list of the references mentioned throughout our response can be found at the end of this document.

- **Methodological robustness.** As the reviewer suggested, we have now performed additional analyses to evaluate the statistical significance controlling the spatial autocorrelation, choice of different brain parcellations, as well as potential bias in clustering analysis (please see below). Based on other reviewers' suggestions, we also demonstrated the specificity of principal functional gradient, compared surface- and volume-based pipeline (please see above), and introduced a new dataset ensure the reproducibility for the sleeping condition (please see below).
- **Methodological clarity.** For each section, we have tried to first describe the methods and objectives, and then present the corresponding results in more detail.
- **Introduction and Discussion.** We revised the Introduction to make our motivations more understandable and clearer. In addition, the Discussion section has been substantially revised to

better contextualize our findings and to add more theoretical focus and discussion. A summary figure has also been updated to organize the overall findings.

- **Readability.** To be accessible to a broader audience, we have consulted with several experts in the field of neuroscience to review the revision and make any necessary changes.

My detailed comments are provided below.

Abstract:

Line 11: “spatiotemporally propagation” > spatiotemporally propagating.

Line 12 (and throughout the rest of the paper): “principle” should be “principal” (these words are often confused even by native speakers).

Line 15: “the” global state of consciousness.

According to the reviewer’s suggestions, we have now fixed above grammatical errors in our manuscript. We thank the reviewer’s reminding.

An overall comment is that it would be preferable to have line numbers that are continuous, rather than restarting at each page.

We have now have implemented continuous line numbers throughout the revised manuscript and response letter.

Introduction

Line 5: The consciousness entity: please rephrase.

We have now rephrased ‘the consciousness entity’ to ‘consciousness’.

Line 33: reference for the claim of drowsiness within minutes during resting-state?

Two reference paper has been added:

Tagliazucchi, Enzo, and Helmut Laufs. "Decoding wakefulness levels from typical fMRI resting-state data reveals reliable drifts between wakefulness and sleep." *Neuron* 82.3 (2014): 695-708.

Soehner, Adriane M., et al. "Unstable wakefulness during resting-state fMRI and its associations with network connectivity and affective psychopathology in young adults." *Journal of affective disorders* 258 (2019): 125-132.

- In general, the Introduction mentions a lot of technical terms (criticality, scale-free dynamics, etc): it would be helpful for non-specialist readers to describe each new technical term a bit more.

We are grateful for the reviewer's helpful suggestion. As previously mentioned, we have now enhanced the introduction section to make the content more understandable and transparent. To prevent potential reading difficulties, we invited peers with diverse research backgrounds in neuroscience to read the revised version and offer feedback for revisions. For instance, regarding the term *criticality*, we used a more accessible description (from line 82): “*According to computational models²⁴, the variability of spontaneous activity arises from the dynamic system's noise-induced transitions between multistable attractors²⁵ or fluctuation around a critical line proximal to instability²⁶.*”

Results

The Results are hard to understand without reading the Methods first (which is what I resorted to doing). I understand this order is the journal's policy, but I strongly suggest that in the Results the authors should introduce at least briefly each new analysis and its motivation before describing its outcomes, to orient the reader.

We agree that it would be beneficial to provide more context in the Results section to orient the reader. To address the reviewer's important suggestion, in the revised Results section we have modified each section to include additional descriptions of the motivations and introductions to the methodology, while adhering to the journal's word count requirements; some of the changes are outlined below:

- Section 1: we have updated the text to describe how group-level consciousness-related maps were calculated from HCP resting-state and sleep fMRI data; we also added background and rationale for the use of functional gradient and T1w/T2w maps.
- Section 2: we elucidated the rationale and methodology employed in computing the hierarchical index and provided a brief description of the Myconnectome dataset.
- Section 3: we revised our rationale for examining participants under the influence of psychedelics and those with psychiatric disorders.

- Section 4: we provide further details on using the global signal (GS) topology and node-wise weighted strength to characterize global integration/coordination.
- Section 5: we revised the methodological details of the quasi-periodic pattern (QPP) analysis.
- Section 6: we extended the methodology and motivation for using macaque electrocorticography data, also noting the difference of functional gradients between two macaques.
- Section 7: we explained the rationale for the use of the Allen Human Brain Atlas, as well as the special emphasis on the tuberomammillary nucleus.

Page 4, Lines 20-21: this is very hard to follow: where are “millions of dense state pairings” coming from? How does this reflect vigilance?

We thank the reviewer for raising these questions. This part of analysis is to establish a group averaged map for the decreasing vigilance. To achieve this, we divided every resting-state run from HCP data into non-overlapping time windows and considered that, at the population-level, the time windows closer to the end of scanning were predisposed to have lower alertness in comparison with earlier time windows. Such approach and hypothesis are underpinned by many previous studies^{10,32-35}. Practically, as shown in Fig. R9, each HCP run was divided into 24 non-overlapping time windows of 50 frames (1200 frames per run). For each run, two different time windows were paired to form 276 ($24 \times (24-1) / 2$) different combinations, with the time interval ranging from 1 to 23 intervals. The larger interval of two windows was inferred to have a higher possibility of a decrease in vigilance. Collectively, for each cortical grayordinate (Figure 1e in the revised manuscript), the Spearman's rank r was calculated between the normalized low-frequency BOLD amplitude and the time interval across 1,084,128 (276 combinations \times 4 runs \times 982 individuals) pairs of states/time windows.

In the latter part of our study, we developed the hierarchical index as a signature and found that it can capture fluctuations of wakefulness/vigilance during resting-state fMRI scanning. Across resting-state sessions in the *Myconnectome* dataset, the hierarchical index was able to explain daily fluctuations in self-reported fatigue and heightened attention (Figure 3e-f in the revised manuscript). Importantly, our time trend analysis revealed that within-individual hierarchical indices decreased across the resting-state scanning at the group level, which supporting our approach to consider scanning time as a population-level proxy of vigilance states.

Fig. R9 (added to revised version as Supplementary Fig. 2). A schematic diagram illustrating the method used to generate the decrease in vigilance pattern from resting-state fMRI data. Four resting-state runs were performed for each individual, with each run consisting of 1200 frames divided into 24 non-overlapping windows. Within each run, every two windows were paired as a sample, with the greater the temporal distance between the two windows, the more likely a decrease in vigilance level was assumed; i.e., for each cortical vertex, the difference in temporal order and the difference in normalized BOLD amplitude were correlated across all pairs of windows and runs.

Revisions to the manuscript:

- Main text from line 122: *“Previous research has demonstrated that individuals would generally exhibit a decrease in alertness over the course of the scan at the population level^{32,34}. Therefore, we divided each resting-state run from HCP data into non-overlapping time windows (50 volumes, 24 windows), and two different time windows were paired to form 276 ($24 \times (24-1) / 2$) different combinations, with the time interval ranging from 1 to 23 intervals. The larger interval of two windows was inferred to have a higher possibility of a decrease in vigilance. To measure the group-level pattern of alertness dropping, Spearman's rank r was calculated between the normalized low-frequency BOLD variability and the time interval across 1,084,128 (276 combinations \times 4 runs \times 982 individuals) pairs of states/time windows (Fig.1d-e, Methods, Supplementary Fig. 2).”*
- *Added new figure as Supplementary Fig. 2.*

Page 5, lines 22-23: the authors should distinguish between the psychedelic and psychotic state more carefully. Psychedelics can have “psychoto-mimetic” effects, but equating the two is an oversimplification that should be avoided, especially given the recent surge in psychedelic-assisted therapy aimed at treating mental disorders.

We appreciate the reviewer’s insight and agree that it would be more reasonable to distinguish psychedelic and psychotic state carefully. To address the reviewer’s important suggestion, we have now rewritten this part as follows: *“Having studied trajectories of the signature across wakefulness, anesthesia, and sleep, we next aimed to investigate whether it can be sensitive to other altered states of consciousness. Here, we focused on two related situations^{57–59}: i) psychotomimetic effects of drugs which inducing altered subclinical psychotic-like experiences; ii) individuals with psychiatric disorders whose conscious processing might be impaired.”.*

Page 5, line 25: higher hierarchical index seems in contrast with the results of Girn et al (2022, NeuroImage) on the same dataset, showing a “collapse” of the principal gradient. Could the authors explain this discrepancy?

We are delighted to provide an explanation of the relationship between our results and those of Girn et al that the reviewer has mentioned.

Although our work and that of Girn et al. are both related to the principal functional gradient, the computational models we applied are entirely different. Our work focused on the temporal variability of large-scale brain activity, which is linked to the system's dynamic flexibility. We found that the shifting of temporal variability is consistent across different paradigm in changing states of consciousness (e.g., sleep, anesthesia, fatigue), and presents a continuous topological transition from lower-order brain regions to higher-order networks. To characterize and quantify this pattern, we employed the principal functional gradient (also T1w/T2w map) as an empirical map for the cortical hierarchical organization at the populational level. In contrast, the work of Girn et al. is an extension of the work of Margulies et al¹. (which proposed the concept and computational framework of the principal gradients), aiming to construct a principal gradient analysis framework at the individual level. To achieve group comparison in gradient values, the individual-level gradients were usually derived by aligning the individualized embedding components to the group template via an iterative Procrustes rotation⁶⁰.

In addition to the methodological discrimination, we do not consider the conclusions of our work and Girn et al to be in disagreement. Our results indicate that a higher hierarchical index is linked to greater fluctuations and more complex integration/coordination patterns in higher-order brain regions (Fig. 4). In the work of Girn et al, they explain the functional gradient changes at the individual level as a movement along a continuous *shift in FC pattern (dis)similarity between two axis extremes*; that is, the more dominated the gradient, the more distinct patterns of functional connections between primary and higher-order brain regions; otherwise, the more "collapsed" the gradient, the *greater unimodal-transmodal integration* potentially occurs (the result in Girn et al).

Collectively, the unimodal-transmodal architecture is a ubiquitous feature of the cortex, and dimensionality reduction of fine-grained functional connectome is a viable approach to uncovering its hierarchical representations. In our study, we employed the group-level principal gradient to explain our findings, while Girn et al has utilized the individual-level gradient to explore the dynamical integration and segregation trend (*'unimodal-transmodal crosstalk'*). Therefore, we believe that our work differs in essence from the work of Girn et al by applying the principal functional gradient as a group template of cortical processing hierarchy, and our conclusions are not conflicting. In the revised version, we have now added the following sentence (as well as references, see line 149): *"Next, we aim to test whether the topologically altered spontaneous cortical activities can be compressed to capture the graded changes of the levels of consciousness, elucidating its reproducibility and power at the individual level. Therefore, we defined a univariate 'hierarchical index' as the rank correlation between the spatial distribution of cortical BOLD variability and the group-level principal functional gradient (Methods). This hierarchical index can provide a proxy for the topological shift of cortical neural variability, using the group-level gradient as an empirical map, which differs from studies focusing on the individual-level perturbation of functional gradients^{5,31}".*

Page 6, line 15: the authors should explain more clearly what "the graph theory approach" is, as the current explanation is very uninformative.

We appreciate the reviewer for raising this concern and we apologize for the previous uninformative description. In the revised manuscript, we have now provided a more detailed explanation as follows: *"As an alternative, a weighted strength approach was also applied as a proxy for functional integration / coordination. The node-wise weighted strength was developed based on the graph-theoretical concept of degree, which aims to quantify the global functional connectivity to other regions (Methods). Following the same clustering pipeline, we found that the two approach yielded a similar clustering solution (Methods,*

Supplementary Fig. 9)”. Correspondingly, in the revised Methods, we have now added: “To characterize functional integration/coordination, we also used a nodal strength approach to measure the pattern in which cortical regions are temporally connected. For each time window, we applied the 200 parcellation Schaefer atlas and Pearson correlation to construct a cortical functional network (200 × 200). Node strength is defined as the sum of weights of links connected to the node”.

Page 7, lines 9-11: as above, this analysis is very unclear without having read the Methods.

We apologize for the lack of clarity in our previous description and have now added the following description to the text (line 238): *“To determine the primary QPP event in the population, we downloaded a recently published QPP template⁶¹ which was generated using an optimized, computationally expensive algorithm based on vertex-wise cortical data. As shown in Fig. 5a, the primary QPP manifests as a dynamic cycle of activation and deactivation which spatially following the macroscale gradient, lasting approximately 21.6 seconds (30 volumes). To match the possible QPP events, we applied a sliding window approach to iteratively compare correlation between the template and each spatiotemporal flattened segment with a temporal step of 1 TR. Segments were identified as QPP events whose local maxima exceeding the threshold ($r = 0.4$), resulting hundreds of events detected for the 100 subjects (Fig 5a-b; Methods; initial states: 785 events, terminal states: 993 events).”*

Page 8, line 1: define BLP?

We have now defined ‘BLP’ as ‘band-limited power’ at line 271.

Page 9, lines 2-21: these results are impressive, but it does not seem that they show modulation by the histaminergic system, only correlation (albeit greater than other hypothalamic nuclei). The authors should relate these results to their previous observations pertaining to consciousness (for instance by considering the different levels of vigilance) if they want to make the kind of stronger claim that they presently do.

We are appreciative of the reviewer's positive remarks and thoughtful considerations. We agree that the previous claim is overly ambitious and that the current results do not support a strong conclusion that hierarchical dynamics are causally modulated by the histaminergic system.

Due to the challenges associated with collecting human tissue samples across different states, we utilized an in-vivo "imaging transcriptomics" technique to decode the molecular correlates of the neuroimaging phenotype, linking spatial variations across the transcriptomic and imaging domains. While this approach is somewhat indirect, it still has the potential to yield plausible biological and transcriptomic mechanism widely used in recent studies^{29,51-55}. In this way, we found that the gene expression of histaminergic system (a key component of the ascending reticular activating system) showed a prominent association with the principal functional gradient. Subsequently, the implication of histaminergic system is supported by the cortical functional connectome of hypothalamic TMN based on 7T fMRI. These results are in line with previous evidence showing that TMN histaminergic neurons widely project to the cerebral cortex^{43,44} and promote wakefulness. Therefore, our results suggest that the histaminergic system is nonhomogeneous and associated with cortical hierarchical organization, thus providing a plausible biological basis for our fMRI observations.

In response to reviewer's remark, in the revised version, we have now toned down the claim in relation to the histaminergic system, and highlighted that future research is necessary to determine its possible causal role to modulate hierarchical dynamics. Meanwhile, we acknowledge that spatial resolution of conventional fMRI datasets (e.g., sleep, anesthesia) fundamentally limits the further application of dynamic state analysis focusing on hypothalamic TMN, which has an average volume of around 50 mm³ (based on used hypothalamic atlas⁶²) and is less than 2 voxels at a resolution of 3 mm³. Therefore, we utilized the HCP 7T resting-state fMRI data to enable higher-resolution analysis.

Revisions to the manuscript:

- Revised results line 327: *"These results, derived from both transcriptome and functional connectome data, provide preliminary evidence linking the histaminergic system to hierarchical dynamics across the neocortex."*
- Modified version of related discussion: *"As an evolutionarily conserved system, histaminergic neurons play a prominent role in sustaining wakefulness⁴³ through their projections from hypothalamic TMN to a wide array of cortical and subcortical regions. A recent study found that histaminergic neurons can broadcast dual inhibitory-excitatory signals throughout the neocortex through GABA-histamine axons⁴⁵. In this work, we observed a spatial correlation between the low-dimensional functional hierarchy, histaminergic molecular markers, and the TMN functional connectome, suggesting a potential role of the histaminergic system in modulating heterogeneous*

dynamics across the cortex. Further evidence is needed to establish direct causality, such as brain stimulation in animal models.”

Discussion

The arousal-promoting system results are given great prominence, but as mentioned above, the claim of mediation is not substantiated by evidence: the authors only show a correlation in a separate dataset.

We agree with the reviewer that overdiscussing the histaminergic results is inappropriate as these results reflect a correlative rather than causal relationship. We have now moderated the discussion in our revised version, as addressed in our response above.

Page 10, line 8: “zombies” will only be familiar to readers with philosophy background: this terms should be clarified (or better, removed).

As suggested by the reviewer, we have now removed the term ‘zombies in the revised Discussion.

Line 15: reference for the dynamic core hypothesis?

To address the reviewer’s request, we have now cited the reference for the dynamic core hypothesis:

Tononi, G. & Edelman, G. M. Consciousness and complexity. Science vol. 282 (1998).

Page 10, line 29: What is the evidence for modulation of neuronal excitability?

We are appreciative of the reviewer for bringing this matter to our attention. We have eliminated the term ‘neuronal excitability’ from our discourse, as it was imprecise. Our research was mainly based on analyses of task-free fMRI and electrocorticography data, which captures macroscopic activity through alterations in metabolic and electrophysiological signals, which cannot infer the excitability of neurons. To address the reviewer's comment, we have amended the wording of key sentences to avert any potential misinterpretation. In our revised discussion, we have replaced the phrase ‘modulation of neuronal excitability’ with ‘modulating macroscopic brain activity’.

Page 11, line 30: what do the authors mean by “a conservative system”?

To avoid the ambiguity, we have now fixed this term by using “an evolutionarily conserved system” (line 393).

Page 11-12: the discussion of artificial neural networks, deep learning, and sleep is extremely speculative and seems to have very little to do with the actual manuscript's results. Please remove.

We understand and appreciate the reviewer's valid concern regarding the deviation of our discussion relating to artificial neural networks, deep learning, and sleep. The discussion builds on the standpoint that the dynamic range of BOLD signal fluctuations reflects system's dynamic flexibility, which is circuitual for both human fluid intelligence as well as artificial intelligence (AI). Considering these, the writing of this part was motivated and supervised by AI experts from our author list. However, we agree with the reviewer that such discussion could be unnecessary and disconnected to our main results in this work. Therefore, we have now removed the entire section of discussion in the revised version.

The authors acknowledge that the QPP analysis is restricted to one dataset, but it is unclear why it cannot be also performed in the other datasets (especially the altered states such as sleep, anaesthesia and psychedelics), which would make the results much more robust.

We appreciate the reviewer's concern and take this opportunity to clarify this limitation. In the revised Methods (from line 1103), we have now added the following sentences: *“There are two major obstacles that prevented us from performing a human QPP analysis beyond the HCP data: i) The QPP analysis is based on a recurring spatiotemporal pattern lasting approximately 20 seconds, which has been reliably identified in previous studies^{42,61,63}. However, most of the human fMRI datasets we used have a repetition time of around 2 seconds, so only about 10 time points can be used for the calculation of spatiotemporal cycles. In contrast, the HCP dataset contains up to 30 time points (TR = 0.72s), providing a more robust basis for analysis. ii) In addition to the limited time points available in most human fMRI datasets, the HCP data have significantly more fMRI runs. This larger sample size is advantageous when attempting to compare QPP across different states.”*

We also apologize for the lack of clarity in our previous writing; in addition to the QPP analysis on HCP data, we indeed conducted a QPP analysis on macaque electrocorticography (ECoG) data (in the section: *Hierarchical Dynamics in Macaque Electrocorticography*). As the first attempt to perform the QPP analysis on macaque ECoG recordings, we observed a recurrent spatiotemporal pattern analogous to those

in humans, which intrinsically propagates along the macroscale principal functional gradient, and the average spectrogram of such gamma-BLP QPP patterns resembled a typical pattern of sequential spectral transitions (SST) events. Extending to our human QPP analysis, the macaque result suggests that such global wave, in conjunction with SST event, is more likely to manifest a large-scale brain activity (i.e., larger gamma-BLP) of higher-order cortices during higher levels of vigilance (eye-opened > eye-closed > sleep). We have now clarified this point in the revised Discussion (line 422): “*Nevertheless, the analysis of the QPP has been extended to the ECoG data of macaques in different conditions*”.

Methods

Page 34: would the results of the k-means analysis also hold if performed separately for each individual? And relatedly: could the clusters be mostly driven by differences between individuals, rather than across time-points?

The reviewer raises a concern that the k-means clusters may be confounded by individual variations rather than time-dependent fluctuations. If the clusters are able to differentiate between distinct individuals rather than temporal segments of the same individual, this would be a major impediment to our analysis. To assess this possibility, we computed the proportion of 96 sliding windows (24 nonoverlapped windows \times 4 runs) of each subject that were assigned to state 1, leaving the remaining windows assigned to state 2. We found that no individual was entirely assigned to state 1 or state 2, with the proportion of state 1 occupied $52.3 \pm 18.8\%$ across the 100 unrelated subjects (Fig. R10a). This preliminary evidence suggests that the k-means analysis was not mostly driven by differences between individuals. To further minimize the potential impact of individual differences, we repeated the subsequent analysis using a balanced subset of data: for each individual, the unmatched time windows would be removed randomly to ensure the proportion of state 1 and state 2 are equal. The results in Fig. R10b-c were highly consistent with those shown in Fig. 4 in the main text, despite the equal distribution of windows across the two states. Furthermore, as the reviewer suggested, we conducted a two-cluster k-means analysis of global signal topology at individual level, independently for 100 unrelated subjects, each with a sample size of 96 (24 non-overlapping windows \times 4 runs). To characterize the stability of the clustering results, we compared the individual-level clustering with the group-level clustering in terms of spatial similarity in comparison with null models (spatial permutation, details can be found at our response below). We found that the clustering based on the individual level was significantly correlated with the group-level spatial correlation (Fig. R10d, state 1: averaged $r = 0.57$; state 2: averaged $r = 0.52$). Considering the limited sample size of the individual clustering, we randomly selected 5 individuals and

repeated the same experiment 100 times, finding that the clustering results were highly correlated with the group-level analysis in the main text (Fig. R10e, state 1: averaged $r = 0.88$; state 2: averaged $r = 0.81$). Collectively, the clustering primarily captures temporal dynamics rather than individual variations, and can yield stable results with a small sample size.

Fig. R10 (added to the revision as Supplementary Fig. 8) Supplementary analyses to evaluate the impact of inter-individual variations on the clustering of dynamic brain states. a, In 100 unrelated subjects, the K-means clustering method assigned a proportion of individuals to state 1, with each individual having 4×24 sample points; to mitigate the impact of inter-individual discrepancies, in subsequent analyses (b-c) we ensured that the sample points of the two states were balanced; for example, for a hypothetical sample, if state 1 represented 40% of the total, then we would eliminate 20% of state 2 to ensure that state 1 and state 2 were equally distributed. b, 2-cluster solution of the GS topology in 3840 balanced time windows from 100 unrelated HCP individuals. Scatter and distribution plots of the hierarchical index; the hierarchical similarity with the GS topology is shown. Each point represents a 35 s fragment. State 1 has significantly larger hierarchical index ($P < .0001$) and hierarchical similarity with GS topology ($P < .0001$) than State 2, indicating a higher level of vigilance and more association regions contributing to global fluctuations; meanwhile, the two variables are moderately correlated ($r = 0.47$, $P < .0001$). c, Top: Higher overall connectivity entropy in State 1 than State 2 ($P < .0001$). Bottom: Higher overall connectivity entropy in states with a higher hierarchical index (top 20% versus bottom 20%; $P < .0001$). d, The stability of the 2-solution K-means analysis based on a single subject (96 runs) was assessed by 100 independent experiments. The upper right panel shows the spatial correlation between the averaged cluster center and the group-level state 1 center; the gray area indicates the correlation based on the group center generated by the spatially permuted null model; the lower right panel

corresponds to state 2. The highest correlation was chosen to match individual and group states in each independent experiment. e, The stability of the 2-solution K-means analysis based on five randomly selected subjects (65×5 runs) was assessed by 100 independent experiments. Other parts are consistent with d.

Revisions to the manuscript:

- Main text from line 211: *“Control analyses suggest that the clustering and result were primarily driven by changes in brain state rather than individual differences”*.
- Modified Methods from line 1027: *“To examine whether the clustering is primarily driven by individual difference, we computed the proportion of 96 sliding windows (24 nonoverlapped windows \times 4 runs) of each subject that were assigned to state 1, leaving the remaining windows assigned to state 2. We found that no individual was entirely assigned to state 1 or state 2, with the proportion of state 1 occupied $52.3 \pm 18.8\%$ across the 100 unrelated subjects (Supplementary Fig. 8a). This preliminary evidence suggests that the k-means analysis was not mostly driven by differences between individuals. To further minimize the potential impact of individual differences, we repeated the subsequent analysis using a balanced subset of data: for each individual, the unmatched time windows would be removed randomly to ensure the proportion of state 1 and state 2 are equal. The results in Supplementary Fig. 8b-c were highly consistent with those shown in Fig. 4, despite the equal distribution of windows across the two states. Furthermore, we conducted a two-cluster k-means analysis of global signal topology at individual level, independently for 100 unrelated subjects, each with a sample size of 96 (24 non-overlapping windows \times 4 runs). To characterize the stability of the clustering results, we compared the individual-level clustering with the group-level clustering in terms of spatial similarity in comparison with 10,000 null models considering spatial autocorrelation. We found that the clustering based on the individual level was significantly correlated with the group-level spatial correlation (Supplementary Fig. 8d, state 1: averaged $r = 0.57$; state 2: averaged $r = 0.52$). Considering the limited sample size of the individual clustering, we randomly selected 5 individuals and repeated the same experiment 100 times, finding that the clustering results were highly correlated with the group-level analysis using 100 subjects (Supplementary Fig. 8e, state 1: averaged $r = 0.88$; state 2: averaged $r = 0.81$)”*

The authors should specify which analyses were performed with which parcellation: the Schaefer atlas is

mentioned in some places, the Brainnetome in others: although the authors offer some justification for this, it would be very important to show that the results are not parcellation-specific.

We thank the reviewer for raising this important concern. To address this issue, we have added a section to the revised Methods to explain which analysis was performed with which parcellation. Briefly, we utilized the Schaefer atlas in our main analyses using cortical parcellation; however, in Section 7, we chose to use the Brainnetome atlas because it was created in volumetric space, making it a better match for the samples in the Allen Human Brain Atlas, which provides anatomical locations in MNI space.

Furthermore, we have conducted supplementary analyses to assess the impact of parcellation factors on our findings. In Section 4, we initially computed regional-level temporal and connectivity entropy based on the Schaefer atlas; we have now compared the results with those obtained from the Brainnetome atlas, revealing a high degree of consistency between the two atlases (Fig. R11). In Section 5, we repeated the QPP analysis using the time series derived from Brainnetome atlas and found the results were highly comparable (Fig. R12). In Section 7, we reported that three different pipelines all showed a highly spatially positive association between HDC gene expression and principal functional gradient: i) using the Human Brainnetome Atlas and *abagen* toolbox⁵⁴, ranked 3rd; ii) using surface-pipeline of Schaefer atlas provided by Anderson et al, ranked 5th (Fig. R13a-b)²⁹; iii) atlas-free pipeline in *Neurovault* website, ranked 9th in terms of variance explained. For the analysis of hypothalamic functional connectivity, we supplemented the results based on the Schaefer parcellation (Fig. R13c). Collectively, these analyses demonstrate that our results are not parcellation-specific.

Fig. R11 (added to the revised version as Supplementary Fig. 15). The entropy measurements based on ROIs defined by the Brainnetome Atlas and Schaefer's parcellation are highly consistent across 9600 brain states (100 subjects \times 24 windows \times 4 runs). a, connectivity entropy. b, temporal entropy.

Quasiperiodic pattern (QPP) analysis based on Brainnetome atlas

Fig. R12 (added to the revised version as Supplementary Fig. 16). Group-averaged QPP events detected in different vigilance states (initial and terminal 400 frames, respectively) based on QPP analysis. For this visualization, the time series of the bottom 20% (a, blue) and top 20% (b, red) of the hierarchy regions were averaged across 30 frames. Greater saturability corresponds to the initial 400 frames with plausibly higher vigilance. c-d, Distribution of the temporal correlations between the averaged time series in the template and all the detected QPP events. Left: Higher vigilance; right: lower vigilance. For the top 20% multimodal areas, an r threshold of 0.5 was displayed to highlight the heterogeneity between the two states.

Fig. R13 (added to the revised version as Supplementary Fig. 17). a-b, Gene expression pattern of the HDC is highly correlated with functional hierarchy ($r = 0.72$) and the expression of the HRH1 gene ($r = 0.71$). c, Spatial association between hypothalamic subregions functional connection to cortical area and functional gradient. The tuberomammillary nucleus showed one of the most outstanding correlations. From left to right: tuberomammillary nucleus (TM), anterior hypothalamic area (AH), dorsomedial hypothalamic nucleus (DM), lateral hypothalamus (LH), paraventricular nucleus (PA), arcuate nucleus (AN), suprachiasmatic nucleus (SCh), dorsal periventricular nucleus

(DP), medial preoptic nucleus (MPO), periventricular nucleus (PE), posterior hypothalamus (PH), ventromedial nucleus (VM).

Revisions to the revision:

- Add to the Revised Methods: *“Instead of the Schaefer’s parcellation, the Human Brainnetome Atlas was used to spatially define the samples, because it was developed in volumetric space. Thus, it is more suitable for matching samples from the Allen Human Brain Atlas, whose anatomical location was provided in MNI space” and “A note on Brain Parcellations. In Sections 1-3, a network parcellation of Yeo’s 17 networks was used to interpret the results, and the main analyses were performed in an atlas-free manner. In Section 4, we computed temporal and connectivity entropy at the regional scale based on the Schaefer atlas and further compared the results with those from the Brainnetome atlas (Supplementary Fig. 15). In Section 5, we performed the QPP analysis using the time series derived from both the Schaefer parcellation (Fig. 6) and the Brainnetome atlas (Supplementary Fig. 16). The results were highly comparable. The analysis in Section 6 was based entirely on the macaque electrodes. In Section 7, we reported that three different pipelines all showed a highly spatially positive association between HDC gene expression and the main functional gradient: i) using the Brainnetome atlas and the Abagen toolbox, ranked 3rd; ii) using the surface pipeline of the Schaefer atlas provided by Anderson et al²⁹, ranked 5th (Supplementary Table 5); iii) the atlas-free pipeline on the Neurovault website, ranked 9th in terms of variance explained (Supplementary Table 6). We performed the analysis based on both the Schaefer parcellation (Supplementary Fig. 17) and the Brainnetome atlas (Fig. 7) for the analysis of hypothalamic functional connectivity.”*
- Added new figures as Supplementary Fig. 15-17.

The sliding windows used in this study are not consistent across analyses (30 frames, 50 frames); why not use either the same duration or the same number of frames?

We agree that providing a detailed description of the selection criteria would be beneficial. As suggested by the reviewer, in the revised version, we clarified how the sliding windows were selected. For the quasi-periodic pattern (QPP) analysis, the time window depends on the typical duration of primary recurring spatiotemporal pattern. As previous studies determined^{61,63}, the optimal human QPP lasts around 20 seconds. Correspondingly, in this work, we used a sophisticated QPP template recently developed based on HCP data, and the segment duration of this template was pre-set to 30 timepoints (TR=0.72 s). Thus,

the sliding window in our human QPP analysis was set to 30 frames. On the other hand, we applied 50 frames (lasts for 36 s) as the sliding window to investigate reconfigured brain states (e.g., connectivity analyses in global signal topology) across resting-state sessions. For such dynamical states analysis, a previous methodological study⁶⁴ specifically examined the impact of window length. As a result, they summarized that 30-60 s length of sliding window is widely deployed in previous studies and such choice is ‘reasonable’ based on their analysis; meanwhile, spurious fluctuations of dynamic functional connectivity would occur in shorter windows such as 20s. Considering these factors, QPP was limited to 20 seconds, and dynamic state analysis of this duration was deemed insufficient, thereby a sliding window of varying lengths was selected.

Revisions to the revision:

- Added to the main text line 240: *As shown in Fig.5, the primary QPP manifests as a dynamic cycle of activation and deactivation which spatially following the macroscale gradient, lasting approximately 21.6 seconds (30 volumes).*
- Added to the Methods line 1014: *“For the investigation of the dynamic brain state, a window length between 30 and 60 seconds has been found to be reasonable and has been widely used in previous studies⁶⁴”*

Figure 3a: how is significance assessed at the individual-subject level? Three data-points per subject do not seem enough for this to be meaningful (and is this corrected for 15 comparisons)?

The reviewer questions the statistical inference of psychedelic states. In the previous version of our manuscript (Figure 3a and its legend), we labelled asterisk for each subject who showed consistent higher hierarchical indexes across three runs during the LSD administration than in the placebo condition. We noticed that such asterisk could be confusing as we did not aim to represent any statistical inference at the individual level. To address the important concern, we have now used the triangle for those subjects (mentioned above) to avoid potential misunderstanding. We have also clarified that the significance was assessed at the group-level (15 subjects) as follows: *“we found that the hierarchical indices were significantly higher in the LSD condition than in the placebo condition across 15 subjects (Fig. 3a, $P < .01$, Two-way ANOVA)’.*

Figure 4a: the diagram is rather uninformative about how this was actually computed.

We thank the reviewer for raising this concern. After discussing with our colleagues, we have now updated the diagram (Fig. R14) with additional text and elements. We feel that the new diagram can better illustrate how the global signal topology was computed.

Fig. R14 (Fig. 4a in the revised manuscript). Simplified diagram for dynamic GS topology analysis.

Figure 5c,d: “reions” should be “regions”; also define “saturability”?

We have corrected the spelling and changed ‘saturability’ to ‘color saturation’.

Figure 7: it is now well-established that correlations between cortical maps can be spuriously inflated by spatial autocorrelation, and rigorous tests have been developed to control for this effect (e.g., Markello and Misic, 2021, NeuroImage). The authors should demonstrate that the correlation in Fig. 7b-c-e are statistically significant beyond the effects of spatial autocorrelation.

We are grateful for the reviewer's constructive suggestion. In the article³⁷ mentioned by the reviewer, Markello and Misic provide a comprehensive evaluation of the effectiveness of 10 null models in mitigating the effects of spatial autocorrelation when conducting statistical analyses of neuroimaging data. Furthermore, they provide excellent documentation and code (https://github.com/netneurolab/markello_spatialnulls) for implementing all null model frameworks.

While most of the null model results performed similarly, they recommend the Alexander-Bloch framework³⁸ as an optimal option for vertex-wise data (<https://markello-spatialnulls.netlify.app/recommendations.html>). This approach can generate spatially constrained null distributions by applying random rotations for spherical projection of the brain. Therefore, following the Alexander-Bloch method, we projected the vertex-wise principal functional gradient data onto the 32k_fs_LR sphere space and generated 10,000 spatially rotated null maps for calculating P value in permutation tests. In the revised version, we have now ensured statistical significance between consciousness-related maps and principal functional gradient in consideration of spatial autocorrelation (Fig 1c-f-i); then, by way of a parcellation derived from the null models of the principal gradient map, we demonstrate its significant spatial correlation with HDC gene expression (Fig 7b, $P < 0.0001$). On the other hand, for Fig. 7c and 7e, since sampling was not conducted directly in the surface space, we employed the Vázquez-Rodríguez method³⁹, an adaptation of the Alexander-Bloch approach for parcellated brain data, to generate 10,000 null models and present permutation-based P values. These additional analyses affirm the statistical significance of the spatial correlations in Fig. 7b-c-e after accounting for spatial autocorrelation. A schematic diagram was presented to elucidate the process of constructing the spatial permutation model:

Fig. R15 (added to the revision as Supplementary Fig. 2). The generation of 10,000 null models accounting for the effects of spatial autocorrelation. Top: vertex-wise surface-based spatially permutation models on the Alexander-Bloch method³⁸. Bottom: ROI-level surface-based spatially permutation models on the Vázquez-Rodríguez method³⁹.

- Main text from line 142: “*The statistical significance was assessed using a spatially permutation-based null model*”.
- Methods from line 949: “*To test the statistical significance of the spatial correlations, we controlled the effects of spatial autocorrelation using generated null models (a schematic diagram was presented at Supplementary Fig. 4). For analyses in Fig. 1c-f-i as well as Supplementary Fig. 3, we projected the vertex-wise principal functional gradient data onto the 32k_fs_LR sphere space and generated 10,000 spatially rotated null maps for calculating P value in permutation tests. Specifically, the Alexander-Bloch approach³⁸ was applied to generate spatially constrained null distributions by applying random rotations for spherical projection of the brain. This is a recommend method based on a recent study which evaluating of the effectiveness of 10 null models in mitigating the effects of spatial autocorrelation. For Fig. 7c and 7e, since sampling was not conducted directly in the surface space, we employed the Vázquez-Rodríguez method³⁹, an adaptation of the Alexander-Bloch approach for parcellated brain data, to generate 10,000 null models and present permutation-based P values.*”
- Added a new figure as Supplementary Fig. 2.

Also the authors do not explore the equally strongly anticorrelated genes at the other end of the distribution, which may hold equally interesting insights: why focus only on the positive correlations?

We hypothesis that the genetic expression patterns of molecular markers within specific ARAS modules (e.g., glutamatergic, cholinergic, histaminergic, and noradrenergic systems) would be associated with the observed hierarchical signature and may preferentially exhibit a spatial correlation with principal functional gradient.

In response to the reviewer's suggestion, we reordered the gene list based on absolute values and found that the top genes were predominantly positive, with only 11 negative genes in the top 100 (shown in Table. R1). Furthermore, HDC gene had a larger effect size than all negative genes. Additionally, we did not find any negative genes that are directly involved in the regulation of wakefulness or sleep among the top genes (listed in the Supplementary Table 3). Given the scope of our manuscript and our hypothesis, we did not provide further analysis on the other genes in this manuscript, but instead included them in the supplementary materials.

Gene symbol	Gene name	Correlation	P value
SHC1	SHC (Src homology 2 domain containing) transforming protein 1	-0.7011522	1.15E-16

NOL4	nucleolar protein 4	-0.6355382	4.25E-13
NR4A3	nuclear receptor subfamily 4, group A, member 3	-0.6247413	1.37E-12
ABR	active BCR-related	-0.6241972	1.45E-12
LOC100287347	similar to hCG1777462	-0.6171237	3.03E-12
PDE4C	phosphodiesterase 4C, cAMP-specific	-0.6137309	4.29E-12
FILIP1L	filamin A interacting protein 1-like	-0.6112344	5.53E-12
KCNA5	potassium voltage-gated channel, shaker-related subfamily, member 5	-0.6061666	9.19E-12
RAB36	RAB36, member RAS oncogene family	-0.6013123	1.48E-11
C2orf80	chromosome 2 open reading frame 80	-0.6006615	1.58E-11

Table R1 (added to the revision as Supplementary Table 3). The top 11 negatively correlated genes with the principal functional gradient, out of 100 genes with the strongest spatial association (based on absolute correlation value).

Figure 8 and Discussion: the focus on excitability seems misplaced, since this is not one of the aspects that the study's results actually support, as far as I can tell: why not focus on those instead?

We thank the reviewer for raising this important concern. As we responded above, we have revised the inaccuracy concerning the term *excitability*. Accordingly, the *excitability* part in Figure 8 has been removed, and we have updated it with a schematic figure to help organize the overall results of the manuscript. In response to the reviewer's suggestion, we have also extensively revised the discussion section, wherein the first paragraph summarizes and contextualizes our results, the second paragraph provides more theoretical discussion, and the third paragraph discusses aberrant states of consciousness while some parts, such as the discussion of the histaminergic system, have been attenuated. We feel that the revised version of the discussion is more closely aligned with our actual findings and better elucidates the insights of our discoveries.

Fig. R16 (Fig. 8 in the revised manuscript). A summary model of findings in this work. **A**, A schematic diagram of our observations based on a range of conditions: Altered global state of consciousness associates with the hierarchical shift in cortical neural variability. Principal gradients of functional connectome in the resting brain are shown for both species. Yellow versus violet represent high versus low loadings onto the low-dimensional gradient. **B**, Spatiotemporal dynamics can be mapped to a low-dimensional hierarchical score linking to states of consciousness. **C**, Abnormal states of consciousness manifested by a disruption of cortical neural variability, which may indicate distorted hierarchical processing. **D**, During vivid wakefulness, higher-order regions show disproportionately greater fluctuations, which are associated with more complex global patterns of functional integration/coordination and differentiation. Such hierarchical heterogeneity is potentially supported by spatiotemporal propagating waves and by the histaminergic system.

Overall comment on the figures: I think the characterisation of the different states of consciousness with cartoons is truly excellent.

As shown in Figure R16c, a novel cartoon illustration has been incorporated to represent abnormal states of consciousness.

Reviewer #3 (Remarks to the Author):

The authors postulated low-frequency signal (BOLD) variability as a signature of consciousness across multiple brain states which include wakefulness, sleep, different forms of anesthesia, to be further evaluated in subjects under a psychedelic drug, psychotic patients, and others. The main result is an increase in variability which coincides with the spatial location of the first functional gradient, plus a series of analysis showing changes in integration consistent with previously studied metrics.

I think this is a fine paper from a methodological perspective, but unfortunately I also think that the questions addressed by it have been extensively studied in recent years, and thus that the results might lack novelty to be published in this journal. The scale-free properties of BOLD and EEG signals during sleep and drug-induced unconsciousness have been extensively investigated in the past (see Refs. at the end) showing changes in the slope of the power spectrum that are consistent with increased power of low frequency fluctuations. In turn, as correctly identified by the authors, this translates into enhanced signal variability. Moreover, these previous papers identified that these changes predominantly occurred in fronto-parietal DMN regions, which show a substantial overlap with the first functional gradient. Similar results hold to the other datasets studied by the authors (ECOG, LSD) as well as the analysis of integration-related metrics.

References:

Lei, X., Wang, Y., Yuan, H., & Chen, A. (2015). Brain scale-free properties in awake rest and NREM sleep: a simultaneous EEG/fMRI study. Brain topography, 28(2), 292-304.

Tagliazucchi, E., von Wegner, F., Morzelewski, A., Brodbeck, V., Jahnke, K., & Laufs, H. (2013). Breakdown of long-range temporal dependence in default mode and attention networks during deep sleep. Proceedings of the National Academy of Sciences, 110(38), 15419-15424.

Horovitz, S. G., Fukunaga, M., de Zwart, J. A., van Gelderen, P., Fulton, S. C., Balkin, T. J., & Duyn, J. H. (2008). Low frequency BOLD fluctuations during resting wakefulness and light sleep: A simultaneous EEG-fMRI study. Human brain mapping, 29(6), 671-682.

Zempel, J. M., Polite, D. G., Kelsey, M., Verner, R., Nolan, T. S., Babajani-Feremi, A., ... & Larson-Prior, L. J. (2012). Characterization of scale-free properties of human electrocorticography in awake and slow wave sleep states. Frontiers in Neurology, 3, 76.

He, B. J., Zempel, J. M., Snyder, A. Z., & Raichle, M. E. (2010). *The temporal structures and functional significance of scale-free brain activity*. *Neuron*, 66(3), 353-369.

We are appreciative of the reviewer's feedback and have carefully considered these comments. We would like to respectfully clarify that some of the statements made were misinterpreted and therefore, not entirely accurate:

i) The reviewer has repeatedly highlighted that our work only manifests as an *increase/enhance* in certain *spatial locations*. However, our result is not simply unidirectional, but rather a combination of increases and decreases along a cortex-wide, continuous, low-dimensional axis.

ii) The reviewer has made an assumption that the slope of the power spectrum in the previous EEG/ECOG studies is comparable to our findings. However, the two cited EEG/ECOG studies (Zempel et al and He et al) did not characterize any spatial specificity when comparing different conscious states, which is in marked contrast to our work, which reported a topologically organized spatial pattern. Specifically, Zempel et al applied average signals of electrodes to compare the scale-free slopes between awake and slow-wave sleep states; He et al presented the scale-free properties of all or example electrodes across different arousal states and found that averaged power-law exponent and fMRI signal variance were moderately correlated across the total brain.

iii) The reviewer highlighted 5 previous studies (including our previous work) and emphasized previous studies found "*changes predominantly occurring in fronto-parietal DMN regions*". However, as we discussed in our paper, previous research has reported inconsistent findings regarding the specific brain regions involved in changes of conscious states. Among the 5 cited studies, Lei et al reported changes in both the visual network and the DMN network; Tagliazucchi et al observed changes across regions in the DMN and attention networks; however, Horowitz et al found that the DMN network did not vary significantly but instead reported changes across distributed regions such as the visual cortex, primary auditory cortex, and precuneus; Zempel et al and He et al did not discuss the locations of brain regions. This inconsistency in the findings of previous studies indicates the need for an explanation from a continuous topological perspective to elucidate the global states of consciousness. Furthermore, we demonstrated that the consciousness-related pattern was spatially associated with the hierarchical organization of the cortex within single networks, beyond the DMN network (Supplementary Fig. 3).

We have now taken this opportunity to emphasize the novelty and contexture of our findings in the revised Discussion: *“In this study, we revealed a fundamental yet simple phenomenon by utilizing a variety of experimental paradigms (i.e., sleep, anesthesia, drowsiness, psychedelia, and psychiatric disorders), designs (intra- and inter-subject variability), timescales (changed over the course of several minutes or more than one year, i.e., the MyConnectome Project), imaging modalities, and species: the shifting of global states of consciousness is along a hierarchical continuum of cortical neural variability (Fig. 8a). Adhering to this principle, the multidimensional spatiotemporal patterns of cortical activity can be mapped to a low-dimensional signature (Fig. 8b), allowing for individual-level characterization of different states of consciousness. The signature exhibits significant elevations in potentially abnormal states of consciousness such as psychedelia and in individuals with psychiatric disorders (Fig. 8c). Subsequently, we show that the hierarchical signature aligns with complex patterns of functional coordination and diversity underpinning vivid wakefulness (Fig. 8d, left). Furthermore, we suggest that the hierarchical heterogeneity is modulated by spatiotemporal waves of cortical activities, as well as likely involvement of the histaminergic arousal system (Fig. 8d, right). Combining behavioral, neuroimaging, electrophysiological, and transcriptomic evidence, our results suggest that global state of consciousness is supported by efficient hierarchical processing that can be constrained along a low-dimensional macroscale gradient.”*. Collectively, we believe that our results and insights are distinct from those reported in the cited literature.

Also, in my opinion the interpretation of the results turns a bit awkward when maps are compared with the Allen Brain Atlas transcriptomic data and the authors attempt to adopt an interpretation based on the outcome (in this case, concerning the relevance of histamine). A discussion of the highly speculative use of these and other related methods for the exploratory analysis of fMRI results is beyond the scope of this review, but overall I think these analysis obstruct the main message of the manuscript and tend to be included since they became somewhat trendy.

The motivation of imaging transcriptomics analysis is to provide molecular insights into the cortical gradient associated with global states of consciousness. Previous animal studies have suggested that the ascending reticular activating system (ARAS) plays a crucial role in supporting consciousness^{49,50}, leading us to hypothesize that ARAS neurotransmitter systems, such as the histaminergic, cholinergic, and noradrenergic systems, may be involved in hierarchical heterogeneity across the cortex. Due to the challenges associated with collecting human tissue samples across different states, we utilized an in-vivo "imaging transcriptomics" technique to decode the molecular correlates of the neuroimaging phenotype, linking spatial variations across the transcriptomic and imaging domains. While this approach is somewhat

indirect, it still has the potential to yield plausible biological and transcriptomic mechanism widely used in recent studies^{29,51–55}. Interestingly, we have found that both gene expression of histaminergic system and functional connectome of hypothalamic TMN showed a prominent association with the principal functional gradient. Such result is consistent with our hypothesis of the role of ascending reticular activating system and provides a plausible biological underpinning for our macroscopic fMRI observations. Extending previous evidence showing TMN histaminergic neurons widely project to the cerebral cortex and promote wakefulness, our result suggests that histaminergic system is nonhomogeneous and associated with cortical hierarchical organization, thus serving a plausible biological underpinning for our macroscopic fMRI observations. However, we know that such analysis cannot demonstrate a causal role of histaminergic system, we have now moderated the related description and discussion: “*As an evolutionarily conserved system, histaminergic neurons play a prominent role in sustaining wakefulness⁴³ through their projections from hypothalamic TMN to a wide array of cortical and subcortical regions. A recent study found that histaminergic neurons can broadcast dual inhibitory-excitatory signals throughout the neocortex through GABA-histamine axons⁴⁵. In this work, we observed a spatial correlation between the low-dimensional functional hierarchy, histaminergic molecular markers, and the TMN functional connectome, suggesting a potential role of the histaminergic system in modulating heterogeneous dynamics across the cortex. Further evidence is needed to establish direct causality, such as brain stimulation in animal models*”.

Other limitation to be highlighted is the very limited number of subjects in the sleep dataset (n=&).

To address the reviewer’s important concern, we have now included an additional sleep dataset in our revised version. As a result, we found that the individual-level correlation can be replicated in an independent simultaneous EEG-fMRI dataset. The details were described in the revised Methods: “*To strengthen the reliability of the sleep-related results, we included a publicly available simultaneous EEG-fMRI dataset^{65,66} from OpenNeuro for validation (ds003768). Briefly, the dataset has 33 healthy participants collected at Pennsylvania State University. fMRI data was acquired by an EPI sequence: TR=2100 ms, TE = 25 ms, slice thickness = 4mm, slices = 35, FOV = 240mm, in-plane resolution = 3mm×3mm. The fMRI data in sleep sessions was preprocessed using the BRANT standardized pipeline mentioned above. EEG data were collected using a 32-channel MR-compatible EEG system from Brain Products, Germany. The sleep staging was performed by a Registered Polysomnographic Technologist. The sleep staging was conducted by a Registered Polysomnographic Technologist. Similarly, the evaluation was based on six EEG channels (O1, O2, F3, F4, C3, C4) with each epoch lasting 30 seconds. Some epochs which yielded uncertain results were marked as “uncertain” and those with artifacts too large for reliable scoring were labelled as “unscorable”. The individuals with more than 10 epochs*

labelled with “unscorable” or “uncertain” were excluded. In contrast to our dataset (collected after midnight), sleeping states in this dataset were mainly attributed to wakefulness, N1, and N2 (with only sporadic attributions of N3 stage), resulting in a more limited range of variation. As an illustration, sub-08 was predominantly assigned to wakefulness, with only sporadic occurrences of N1 stage. The sleep stage was encoded as ranked number (wakefulness: 0, N1 stage: -1, N2 stage: -2, slow-wave sleep stage: -3). To facilitate reliable analysis at the individual level, we selected 6 individuals with the largest variance in sleep stages for validation (subjects 1-6: sub-09, sub-10, sub-15, sub-24, sub-27, sub-28; Supplementary Fig. 6).”

Fig. R17 (added to the revision as Supplementary Fig. 6). Hierarchical index captures the temporal variation in sleep stages in independent datasets (gray line: expert scores; blue line: hierarchical index; for subject 1: $r = 0.60$, $P < .01$; for subject 2: $r = 0.53$, $P < .01$; for subject 3: $r = 0.81$, $P < .01$; for subject 4: $r = 0.60$, $P < .01$; for subject 5: $r = 0.77$, $P < .01$; for subject 6: $r = 0.79$, $P < .01$; Pearson correlation). The vertical axis represents four sleep stages (wakefulness = 0, N1 = -1, N2 = -2, slow-wave sleep = -3), with time shown on the horizontal axis; for visualization, we normalized the hierarchical index over time and added the mean of the corresponding expert score.

1. Petrides, M. *et al.* Situating the default-mode network along a principal gradient of macroscale cortical organization. *Proc. Natl. Acad. Sci.* **113**, 12574–12579 (2016).
2. S.-J., H. *et al.* Atypical functional connectome hierarchy in autism. *Biol. Psychiatry* **83**, S269 (2018).
3. Meng, Y. *et al.* Systematically disrupted functional gradient of the cortical connectome in generalized epilepsy: Initial discovery and independent sample replication. *Neuroimage* **230**, (2021).
4. Holmes, A. J., Dong, H. M., Margulies, D. S. & Zuo, X. N. Shifting gradients of macroscale cortical organization mark the transition from childhood to adolescence. *Proc. Natl. Acad. Sci. U. S. A.* **118**, (2021).
5. Girn, M. *et al.* Serotonergic psychedelic drugs LSD and psilocybin reduce the hierarchical differentiation of unimodal and transmodal cortex. *Neuroimage* **256**, 119220 (2022).
6. Burt, J. B. *et al.* Hierarchy of transcriptomic specialization across human cortex captured by structural neuroimaging topography. *Nat Neurosci* **21**, 1251–1259 (2018).
7. Koch, C., Massimini, M., Boly, M. & Tononi, G. Neural correlates of consciousness: Progress and problems. *Nat. Rev. Neurosci.* **17**, 307–321 (2016).
8. Pezzulo, G., Zorzi, M. & Corbetta, M. The secret life of predictive brains: what’s spontaneous activity for? *Trends in Cognitive Sciences* vol. 25 (2021).
9. González-García, C. & He, B. J. A gradient of sharpening effects by perceptual prior across the human cortical hierarchy. *J. Neurosci.* **41**, (2021).
10. Fox, M. D. & Raichle, M. E. Spontaneous fluctuations in brain activity observed with functional magnetic resonance imaging. *Nature Reviews Neuroscience* vol. 8 (2007).
11. Finn, E. S. *et al.* Functional connectome fingerprinting: Identifying individuals using patterns of brain connectivity. *Nat. Neurosci.* **18**, 1664–1671 (2015).
12. Tavor, I. *et al.* Task-free MRI predicts individual differences in brain activity during task performance. *Science (80-.)*. **352**, (2016).
13. Maquet, P. *et al.* Default network connectivity reflects the level of consciousness in non-communicative brain-damaged patients. *Brain* **133**, 161–171 (2009).
14. Boveroux, P. *et al.* Breakdown of within- and between-network Resting State during Propofol-induced Loss of Consciousness. *Anesthesiology* (2010).
15. Tagliazucchi, E. *et al.* Breakdown of long-range temporal dependence in default mode and attention networks during deep sleep. *Proc. Natl. Acad. Sci. U. S. A.* **110**, 15419–15424 (2013).
16. Huang, Z., Zhang, J., Wu, J., Mashour, G. A. & Hudetz, A. G. Temporal circuit of macroscale dynamic brain activity supports human consciousness. *Sci. Adv.* (2020) doi:10.1126/sciadv.aaz0087.
17. Demertzi, A., Tagliazucchi, E., Dehaene, S., Deco, G. & Barttfeld, P. Human consciousness is supported by dynamic complex patterns of brain signal coordination. 1–12 (2019).
18. Luppi, A. I. *et al.* Consciousness-specific dynamic interactions of brain integration and functional diversity. *Nat. Commun.* (2019) doi:10.1038/s41467-019-12658-9.
19. Waschke, L., Kloosterman, N. A., Obleser, J. & Garrett, D. D. Behavior needs neural variability. *Neuron* (2021) doi:10.1016/j.neuron.2021.01.023.
20. Garrett, D. D., Kovacevic, N., McIntosh, A. R. & Grady, C. L. The importance of being variable. *J. Neurosci.* (2011) doi:10.1523/JNEUROSCI.5641-10.2011.

21. Uddin, L. Q. Bring the Noise: Reconceptualizing Spontaneous Neural Activity. *Trends Cogn. Sci.* **24**, 734–746 (2020).
22. Nomi, J. S., Bolt, T. S., Chiemeka Ezie, C. E., Uddin, L. Q. & Heller, A. S. Moment-to-moment BOLD signal variability reflects regional changes in neural flexibility across the lifespan. *J. Neurosci.* **37**, (2017).
23. Good, T. J., Villafuerte, J., Ryan, J. D., Grady, C. L. & Barense, M. D. Resting state bold variability of the posterior medial temporal lobe correlates with cognitive performance in older adults with and without risk for cognitive decline. *eNeuro* **7**, (2020).
24. Deco, G., Jirsa, V. K. & McIntosh, A. R. Emerging concepts for the dynamical organization of resting-state activity in the brain. *Nature Reviews Neuroscience* vol. 12 (2011).
25. Deco, G., Jirs, V., McIntosh, A. R., Sporns, O. & Kötter, R. Key role of coupling, delay, and noise in resting brain fluctuations. *Proc. Natl. Acad. Sci. U. S. A.* **106**, (2009).
26. Ghosh, A., Rho, Y., McIntosh, A. R., Kötter, R. & Jirsa, V. K. Noise during rest enables the exploration of the brain's dynamic repertoire. *PLoS Comput. Biol.* **4**, (2008).
27. Ito, T. *et al.* Task-evoked activity quenches neural correlations and variability across cortical areas. *PLoS Comput. Biol.* (2020) doi:10.1371/JOURNAL.PCBI.1007983.
28. Churchland, M. M. *et al.* Stimulus onset quenches neural variability: A widespread cortical phenomenon. *Nat. Neurosci.* (2010) doi:10.1038/nn.2501.
29. Anderson, K. M. *et al.* Transcriptional and imaging-genetic association of cortical interneurons, brain function, and schizophrenia risk. *Nat. Commun.* **11**, (2020).
30. Wang, X. J. Macroscopic gradients of synaptic excitation and inhibition in the neocortex. *Nat. Rev. Neurosci.* (2020) doi:10.1038/s41583-020-0262-x.
31. Hong, S. J. *et al.* Atypical functional connectome hierarchy in autism. *Nat. Commun.* (2019) doi:10.1038/s41467-019-08944-1.
32. Soehner, A. M. *et al.* Unstable wakefulness during resting-state fMRI and its associations with network connectivity and affective psychopathology in young adults. *J. Affect. Disord.* **258**, (2019).
33. Liu, T. T. & Falahpour, M. Vigilance Effects in Resting-State fMRI. *Frontiers in Neuroscience* vol. 14 (2020).
34. Tagliazucchi, E. & Laufs, H. Decoding Wakefulness Levels from Typical fMRI Resting-State Data Reveals Reliable Drifts between Wakefulness and Sleep. *Neuron* **82**, 695–708 (2014).
35. Fukunaga, M. *et al.* Large-amplitude, spatially correlated fluctuations in BOLD fMRI signals during extended rest and early sleep stages. *Magn. Reson. Imaging* (2006) doi:10.1016/j.mri.2006.04.018.
36. Yang, G. J. *et al.* Altered Global Signal Topography in Schizophrenia. *Cereb. Cortex* (2017) doi:10.1093/cercor/bhw297.
37. Markello, R. D. & Misic, B. Comparing spatial null models for brain maps. *Neuroimage* **236**, (2021).
38. Alexander-Bloch, A. F. *et al.* On testing for spatial correspondence between maps of human brain structure and function. *Neuroimage* **178**, (2018).
39. Vázquez-Rodríguez, B. *et al.* Gradients of structure–function tethering across neocortex. *Proc. Natl. Acad. Sci. U. S. A.* **116**, (2019).
40. Nagasaka, Y., Shimoda, K. & Fujii, N. Multidimensional recording (MDR) and data sharing:

- An ecological open research and educational platform for neuroscience. *PLoS One* **6**, (2011).
41. Mars, R. B. *et al.* Whole brain comparative anatomy using connectivity blueprints. *Elife* **7**, (2018).
 42. Raut, R. V. *et al.* Global waves synchronize the brain's functional systems with fluctuating arousal. *Sci. Adv.* **7**, (2021).
 43. Yoshikawa, T., Nakamura, T. & Yanai, K. Histaminergic neurons in the tuberomammillary nucleus as a control centre for wakefulness. *British Journal of Pharmacology* vol. 178 (2021).
 44. Thakkar, M. M. Histamine in the regulation of wakefulness. *Sleep Medicine Reviews* vol. 15 (2011).
 45. Yu, X. *et al.* Wakefulness Is Governed by GABA and Histamine Cotransmission. *Neuron* **87**, (2015).
 46. Rothhaas, R. & Chung, S. Role of the Preoptic Area in Sleep and Thermoregulation. *Frontiers in Neuroscience* vol. 15 (2021).
 47. Chou, T. C. *et al.* Critical Role of Dorsomedial Hypothalamic Nucleus in a Wide Range of Behavioral Circadian Rhythms. *J. Neurosci.* **23**, (2003).
 48. Thomas Yeo, B. T. *et al.* The organization of the human cerebral cortex estimated by intrinsic functional connectivity. *J. Neurophysiol.* **106**, (2011).
 49. Liu, D. & Dan, Y. A Motor Theory of Sleep-Wake Control: Arousal-Action Circuit. *Annual Review of Neuroscience* vol. 42 (2019).
 50. Magoun, H. W. An ascending reticular activating system in the brain stem. *Arch. Neurol. Psychiatry* **67**, (1952).
 51. Martins, D. *et al.* Imaging transcriptomics: Convergent cellular, transcriptomic, and molecular neuroimaging signatures in the healthy adult human brain. *Cell Rep.* **37**, (2021).
 52. Anderson, K. M. *et al.* Convergent molecular, cellular, and cortical neuroimaging signatures of major depressive disorder. *Proc. Natl. Acad. Sci. U. S. A.* **117**, (2020).
 53. Romme, I. A. C., de Reus, M. A., Ophoff, R. A., Kahn, R. S. & van den Heuvel, M. P. Connectome Disconnectivity and Cortical Gene Expression in Patients With Schizophrenia. *Biol. Psychiatry* **81**, 495–502 (2017).
 54. Markello, R. D. *et al.* Standardizing workflows in imaging transcriptomics with the Abagen toolbox. *Elife* **10**, (2021).
 55. Morgan, S. E. *et al.* Cortical patterning of abnormal morphometric similarity in psychosis is associated with brain expression of schizophrenia-related genes. *Proc. Natl. Acad. Sci. U. S. A.* (2019) doi:10.1073/pnas.1820754116.
 56. Romero-Garcia, R., Warrier, V., Bullmore, E. T., Baron-Cohen, S. & Bethlehem, R. A. I. Synaptic and transcriptionally downregulated genes are associated with cortical thickness differences in autism. *Mol Psychiatry* (2018) doi:10.1038/s41380-018-0023-7.
 57. Sass, L. A. & Parnas, J. Schizophrenia, Consciousness and the Self. *Schizophrenia Bulletin* vol. 29 (2003).
 58. Schartner, M. M., Carhart-Harris, R. L., Barrett, A. B., Seth, A. K. & Muthukumaraswamy, S. D. Increased spontaneous MEG signal diversity for psychoactive doses of ketamine, LSD and psilocybin. *Sci. Rep.* **7**, (2017).

59. Bayne, T. & Carter, O. Dimensions of consciousness and the psychedelic state. *Neurosci. Conscious.* **2018**, (2018).
60. Vos de Wael, R. *et al.* BrainSpace: a toolbox for the analysis of macroscale gradients in neuroimaging and connectomics datasets. *Commun. Biol.* **3**, (2020).
61. Yousefi, B. & Keilholz, S. Propagating patterns of intrinsic activity along macroscale gradients coordinate functional connections across the whole brain. *Neuroimage* **231**, (2021).
62. Neudorfer, C. *et al.* A high-resolution in vivo magnetic resonance imaging atlas of the human hypothalamic region. *Sci. Data* **7**, (2020).
63. Majeed, W. *et al.* Spatiotemporal dynamics of low frequency BOLD fluctuations in rats and humans. *Neuroimage* **54**, (2011).
64. Leonardi, N. & Van De Ville, D. On spurious and real fluctuations of dynamic functional connectivity during rest. *NeuroImage* vol. 104 (2015).
65. Liu, Y. G. and F. H. and L. E. S. and M. M. S. and X. L. Simultaneous EEG and fMRI signals during sleep from humans. *OpenNeuro* (2022) doi:10.18112/openneuro.ds003768.v1.0.9.
66. Gu, Y. *et al.* An orderly sequence of autonomic and neural events at transient arousal changes. *Neuroimage* **264**, 119720 (2022).

REVIEWERS' COMMENTS

Reviewer #1 (Remarks to the Author):

I thank the authors for addressing my questions and concerns. The revised manuscript has improved and represents an interesting contribution to the research on states of consciousness.

Reviewer #2 (Remarks to the Author):

I wish to thank the authors for their thorough work and their responsiveness to my own and the other reviewers' feedback. I am happy to endorse publication of this manuscript.

Reviewer #3 (Remarks to the Author):

Thank you for addressing my comments. Overall, I think the paper is methodologically sound, and even though I still think it has some overlap with the previous literature, the authors convinced me that there is enough novel content for this manuscript to be published.